# Evolution of the firn pack of Kaskawulsh Glacier, Yukon: meltwater effects, densification, and the development of a perennial firn aquifer

Naomi Ochwat[1,5], Shawn Marshall[1,2], Brian Moorman[1], Alison Criscitiello,[3] Luke Copland[4]

[1]Department of Geography, University of Calgary, Calgary, Alberta, T2N 1N4, Canada

[2]Environment and Climate Change Canada, Gatineau, Quebec, K1A 0H3, Canada

[3]Department of Earth and Atmospheric Sciences, University of Alberta, Edmonton, T6G 2R3, Canada

[4]Department of Geography, Environment and Geomatics, University of Ottawa, Ottawa, Ontario K1N 6N5, Canada

[5]Cooperative Institute for Research In Environmental Sciences, University of Colorado Boulder, Boulder, 80309, USA

*Correspondence to*: Naomi Ochwat (naomi.ochwat@ucalgary.ca)

**Abstract.** In spring 2018, two firn cores (21 m and 36 m in length) were extracted from the accumulation zone of Kaskawulsh Glacier, St. Elias Mountains, Yukon. The cores were analyzed for ice layer stratigraphy and density, and compared against historical measurements made in 1964 and 2006. Deep meltwater percolation and refreezing events were evident in the cores, with a total ice content of 2.33 ± 0.26 m in the 36-m core and liquid water discovered below a depth of 34.5 m. Together with the observed ice content, surface energy balance and firn modelling indicate that Kaskawulsh Glacier firn retained about 86% of its meltwater in the years 2005-2017. For an average surface ablation of 0.38 m w.e. yr$^{-1}$ over this period, an estimated 0.28 m w.e. yr$^{-1}$ refroze in the firn, 0.05 m w.e. yr$^{-1}$ was retained as liquid water, and 0.05 m w.e. yr$^{-1}$ drained or ran off. The refrozen meltwater is associated with a surface lowering of 0.73±0.23 m between 2005 and 2017 (i.e., surface drawdown that has no associated mass loss). The firn has become denser and more ice-rich since the 1960s, and contains a perennial firn aquifer (PFA), which may have developed over the past decade. This illustrates how firn may be evolving in response to climate change in the St. Elias Mountains, provides firn density information required for geodetic mass balance calculations, and is the first documented PFA in the Yukon-Alaska region.

## 1 Introduction

With the increasing effects of climate change and the need for understanding glacier and ice sheet melt rates, geodetic methods are useful for indirect measurements of mass balance (Cogley, 2009). Based on repeat altimetry, geodetic approaches to mass balance monitoring rely on several assumptions. Estimates must be made of the density of snow, firn, and ice at the sampling location, with the additional assumption that these densities remain unchanged between the two measurement dates. However, over multi-annual timescales in a warming climate this may not be true (Moholdt et al., 2010b). Meltwater percolation and refreezing can significantly change the firn density profile and mean density of the accumulation zone of a glacier (Gascon et al., 2013), and can introduce large uncertainties when using geodetic techniques to determine glacier mass balance if they are not properly accounted for. For example, Moholdt et al. (2010a) determined the geodetic mass balance of Svalbard glaciers to be $-4.3 \pm 1.4$ Gt yr$^{-1}$, based on ICESat laser altimetry, with the large uncertainty attributed to limited knowledge of the snow and firn density and their spatial and temporal variability. By altering the density and causing surface lowering, meltwater percolation, refreezing, and liquid water storage all complicate the interpretation of geodetic mass balance data.

Warming firn can result in increased meltwater production and altered firn densification processes. Initially, melt can round the snow grains and increase the snowpack density, and then percolate into the firn and refreeze as ice layers or lenses (Sommerfeld and LaChapelle, 1970; Cuffey and Paterson, 2010). On glaciers with medium to high surface melt, and high annual snow accumulation, meltwater that percolates below the winter cold layer often will not refreeze, and may thus form a perennial firn aquifer (PFA) if this water cannot effectively drain through crevasses or moulins (Kuipers Munneke et al, 2014). These internal accumulation processes can significantly increase the firn density, and once ice layers or PFAs form they affect how meltwater percolates through the firn pack (Gascon et al., 2013). Due to the spatial heterogeneity of meltwater retention, percolation, and refreezing processes, there are still many gaps in knowledge of how to model these processes and subsequently estimate firn density in areas where these processes occur (van As et al., 2016).

Meltwater retention in firn is also important for estimating glacial runoff contributions to sea level rise. Numerous recent studies have investigated meltwater refreezing processes in northern locations such as southern Greenland (Humphrey et al., 2012; Harper et al., 2012; De La Peña et al., 2015; MacFerrin et al., 2019; Vandecrux et al., 2020), Canadian Arctic Archipelago (Noël et al. 2018, Zdanowicz et al., 2012; Bezeau et al., 2013; Gascon et al., 2013), and Svalbard (Noël et al. 2020, Van Pelt et al., 2019, Christianson et al., 2015). In many locations with cold deep firn, short term increases in surface melt rates may not result in proportional increases in surface runoff due to percolation and refreezing of meltwater in the firn pack (e.g., Harper et al., 2012; Koenig et al., 2014; MacFerrin et al., 2019). However, in the long term this may lead to expansion of low-permeability ice layers, causing run-off to increase and expediting the movement of water from glaciers to the ocean (MacFerrin et al., 2019, Machguth et al., 2016). Current knowledge of these processes is limited for mountain

glaciers in other regions, although this information is required for improved estimates and models of glacier mass balance
and associated sea-level rise.

In this study two firn cores were retrieved in spring 2018 on Kaskawulsh Glacier, St. Elias Mountains, Yukon, and analyzed
for density and the effects of meltwater percolation and refreezing. We also use a firn model (Samimi et al., 2020) forced by
bias-adjusted ERA climate reanalyses to investigate the evolution of the firn through time at this location. Comparisons of
these measurements with firn density profiles collected at a nearby site in 1964 and 2006 enable us to: (i) quantify
contemporary firn characteristics and densification processes; (ii) determine how the physical properties of the firn pack
have changed over the past ~50 years; and (iii) assess the potential widespread presence of a PFA on the upper Kaskawulsh
Glacier.
**2 Study area**
The St. Elias Mountains are located in the southwest corner of Yukon Territory, Canada, and contain many peaks higher than
3000 m, including the highest mountain in Canada, Mount Logan, at 5959 m a.s.l. (Figure 1). The St. Elias is home to the
largest icefield outside of the polar regions, with an area of ~46,000 $km^2$ (Berthier et al., 2010). Measurements presented
here are focused on the upper accumulation zone of Kaskawulsh Glacier (Figure 1), which is part of an extensive (~63 $km^2$)
snowfield at an elevation of 2500-2700 m a.s.l. This plateau region has subtle topographic variations and includes the
drainage divide between the Kaskawulsh and Hubbard Glaciers.

Kaskawulsh Glacier is a large valley glacier located on the eastern side of the St. Elias Mountains within the Donjek Range,
and is approximately 70 km long and 3-4 km wide. Our 2018 drill site was located on the upper north arm of the glacier in
the accumulation zone (60.78°N, 139.63°W), at an elevation of 2640 m a.s.l. Based on satellite imagery, Foy et al. (2011)
estimated an average equilibrium line altitude (ELA) for the glacier of 1958 m a.s.l. for the period 1977-2007, while Young
et al. (2020) provided a mean ELA of 2261 ±151 m a.s.l. for the years 2013-2019. Our core site is thus well above the ELA,
and has remained within the main accumulation area of the glacier. Mean annual and summer (JJA) air temperatures from
1979-2019 were −10.7°C and −2.5°C, respectively, based on bias-adjusted ERA5 climate reanalyses (Hersbach et al., 2020).
The main melt season occurs from June through August. Over the period 1979-2016, Williamson et al. (2020) reported that
the St. Elias Icefield air temperature warmed at an average rate of 0.19°C decade$^{-1}$ at an elevation of 2000-2500 m a.s.l.,
rising to 0.28°C decade$^{-1}$ at an elevation of 5500-6000 m a.s.l.

Previous studies of Kaskawulsh Glacier have included an analysis of volume change over time based on comparisons of
satellite imagery and digital elevation models (Foy et al., 2011; Young et al., 2020). Several reports in the 1960s documented
various glaciological characteristics and processes occurring in the St. Elias Icefields, as part of the Icefield Ranges Research
Project (IRRP) (Wood, 1963; Grew and Mellor, 1966; Marcus and Ragle, 1970). Firn density and temperature measurements
to 15-m depth were made during this period at site IRRP A, near the Kaskawulsh-Hubbard divide and about 5 km from our
core site. Additional snow accumulation data are available from the 'Copland Camp' site on the upper Hubbard Glacier,
located ~12 km southwest of our drill site and at a similar elevation (Figure 1). A weather station located on a nunatak near
to Copland Camp has been in service since 2013 (60.70°N, 139.80°W, ~2600 m a.s.l.; Figure 1). Other relevant studies in the
region include ice cores collected from the Eclipse Icefield, located 12 km northwest of our drill site (Yalcin et al., 2006;
Zdanowicz et al., 2014) but at a higher elevation (3017 m a.s.l.).

We consider snow accumulation rates, weather conditions, and earlier firn core studies across several different locations
within this broad snowfield region that constitutes the upper accumulation areas of the Kaskawulsh and Hubbard Glaciers.
Some caution is needed in comparing different sites, but the region is relatively flat and uniform, with the exception of some
nunataks. Away from the nunataks there is negligible influence from topographic obstacles or valley walls, so we
hypothesize that the upper accumulation area will be exposed to similar climate conditions and snow accumulation rates over
long periods. The possibility of significant spatial variability cannot be ruled out, however, so we consider this further in the
data analysis.

**3 Methods**
**3.1 Ice core field collection**
Two 8-cm diameter cores were drilled between May 20[th] and 24[th], 2018, using an ECLIPSE ice drill (Icefield Instruments,
Whitehorse, Yukon). With a starting depth of 2 m below the snow surface, Core 1 was 34.6 m long and reached a depth of
36.6 m, and Core 2 was 19.6 m long and reached a depth of 21.6 m. The two cores were drilled 60 cm apart, and core
stratigraphy and density were recorded in the field. At a depth of 34.5 m below the snow surface, liquid water became
evident in Core 1; drilling was stopped at a depth of 36.6 m to avoid the risk of the drill freezing in the hole.

Once the cores were retrieved the presence of ice layers, ice lenses and "melt-affected" firn was logged and the stratigraphic
character (e.g., texture, opacity), depth, and thickness were recorded. Melt-affected firn refers to any firn that displays
physical characteristics indicating that there was the presence of liquid water at some point (Figure S5). This can result in ice
layers, ice lenses, or can be indicated by the lack of grain boundaries, the presence of air bubbles, and opacity. When an ice
horizon extended across the entire diameter of the core, it was labeled as an ice layer. If the ice horizon was of more limited
lateral extent, it was labeled an ice lens. Ice lenses were occasionally wedge shaped.

All of the density measurements for Core 1 were completed in the field. The Core 2 samples could not be measured for
density in the field due to lack of time and were flown to Kluane Lake Research Station frozen, where the measurements
were made within 24 hours of arrival. A random assortment of 125 out of the 196 Core 2 sample bags were damaged during
this transport and were not included in the measurements. This left a random reduced set of samples available to use for the
density analysis, so we were able to construct a density stratigraphy (Core 2 in Figure 2B), but uncertainties are higher for
Core 2 and most of our analysis therefore focuses on Core 1.

**3.2 Ice core density analysis**
Ice core density measurements were completed in the field. Each core was sawed into ~10-cm long sections and the diameter
of the sections measured at each end. The sections were then double bagged, weighed, and assessed for the quality of the
core sample and its cylindrical completeness, which we denote $f$. The average diameter was used to determine the volume of
the core section ($V$). Together with the mass of the core section, $m$, density was calculated following:

$$\rho = m/V, \text{ with } V = f\pi L(D/2)^2, \tag{1}$$


where $\rho$ is the density of the firn, $D$ is the average core section diameter, $L$ is the length of the section, and $f \in [0,1]$ is the
subjectively assessed fraction of completeness of the core section. For example, if visual inspection indicated that about 5%
of the core was missing (e.g., due to missing ice chips caused by the core dogs of the drill head), then $f$ would be 0.95.
Outliers were removed for the background firn density calculations if they were not physically possible (i.e., values >917 kg
m$^{-3}$ or <300 kg m$^{-3}$ at depths below the last summer surface). Outliers from 32-36 m depth had residual liquid water in them,
so these higher density values were retained.

In order to calculate the uncertainty in density, $d\rho$, random and systematic sources of error have to be taken into account in
the propagation of errors:

$$d\rho = \rho\sqrt{\left(\frac{dm}{m}\right)^2 + \left(\frac{dV}{V}\right)^2}. \tag{2}$$

The mass uncertainty was assumed to be 0.3 g, which is a conservative estimate given the scale's accuracy (±0.1 g), but
accounts for potential residual snow or water on the scale. The volume uncertainty is calculated by breaking down Eq. (1) for
sample volume, $V = fAL$, where cross-sectional area $A = \pi(D/2)^2$. There is uncertainty in the measured length of the core
section, $L$, the radius of the core section, $D/2$, and the assessment of the completeness of the core sample, $f$. Each of these
was calculated independently and the propagation of uncertainty was calculated from:

$$dV = V\sqrt{\left(\frac{df}{f}\right)^2 + \left(\frac{dA}{A}\right)^2 + \left(\frac{dL}{L}\right)^2}. \tag{3}$$

*dL* was assumed to be 0.25 cm because the tape measure had ticks at every mm so it could be measured with precision, but
core sections were often uneven, with crumbly edges caused by the drill cutters. The same uncertainty was assigned to the
measurement of core diameter. Given two independent measurements, the uncertainty in the diameter is $dD =$
$\frac{1}{2}\sqrt{(0.25)^2 + (0.25)^2} = 0.18$ cm. For the cross-sectional area, the uncertainty $dA = \pi\, D dD/2$.

Values of *f* were determined by assessing the shape of the core and deciding how complete a cylinder the core section
represented (e.g., accounting for missing volume due to chips from the core dogs along the edges). Three different people
performed this evaluation, so there was subjectivity in each of the *f* values and it is best to be conservative with this estimate.
We assigned this to be $df = 0.2$ for $f < 0.8$ and $df = 0.1$ for $f \geq 0.8$. The uncertainty of a higher *f* value is lower, because when
a core was of good quality it was obvious. Less complete cylinders were more difficult to assess, hence the greater
uncertainty when $f \leq 0.8$. The *f* value has the greatest effect on the overall uncertainty calculation for firn density. We did not
record *f* values for Core 2 in the field, so values are based on the measurements from Core 1. The minimum value recorded
in Core 1 was $f = 0.7$, with a maximum of 1 and an average of 0.96. We assume a value of $f = 0.96 \pm 0.1$ for all of Core 2.

The resulting uncertainty in the density was calculated from:

$$d\rho = \rho\sqrt{\left(\frac{dm}{m}\right)^2 + \left(\frac{dV}{V}\right)^2}\ . \tag{4}$$

For the average densities, $\bar{\rho}$, the uncertainty can be calculated from the standard error of the mean, $d\bar{\rho} = d\rho/\sqrt{N}$, for sample
size *N*. This can be estimated from the average value of $d\rho$, but we report the more precise uncertainty calculated from the
root-mean square value of all point values, $d\rho_k$: $d\bar{\rho} = \frac{1}{N}[\sum_N d\rho_k{}^2]^{1/2}$. Density can be expressed as water equivalence (w.e.)
for each core section from the conversion $w = L\rho/\rho_w$, where $\rho_w$ is the density of water. For the whole core, of length $L_c$, the
water equivalence is $w_c = L_c\bar{\rho}/\rho_w$, with units m w.e. We also include an estimate of the age of the cores, based on an
estimate of the average annual net accumulation rate, $\bar{a}$, with units m w.e. yr$^{-1}$. The age of the core is then $\tau_c = w_c/\bar{a}$.
Uncertainty is estimated by propagation of uncertainties in $w_c$ and $\bar{a}$. We use an uncertainty of $dL_c = 0.5$ m for the total
length of the core, $L_c$, which is based on measurements during retrieval of Core 1 of 35.05 m from the drill panel, 34.59 m
from the addition of core lengths, and 34.25 m from the sum of the ~10 cm samples. For Core 2 the length was 19.75 m from
the drill panel, 19.35 m from the addition of core lengths, and 19.63 m from the sum of the ~10 cm samples.

Ice fraction, $F_i \in [0,1]$, was calculated for each 10-cm section of the firn core. Here ice was defined based on its lack of air
bubbles and crystalline structure, as compared to the granular structure of firn. We refer to this as ice fraction, rather than
melt percent, as melt percent generally assumes that the meltwater remains within the net annual accumulation layer
(Koerner, 1977), which cannot be assumed here due to evidence that meltwater percolates beyond the annual accumulation
layer and refreezes into previous years' accumulation. The thickness of individual ice layers was summed within each 10-cm
core section. In core samples that had ice lenses, their diameter typically occupied about 50% of the core sample; therefore
their thickness was divided by two before being summed. For each core section, total ice content was divided by the length
of the section, $L$, to give $F_i$. These values were also summed to give the total ice core ice content.

To understand the firn densification process in the absence of refrozen meltwater, the 'background' firn density is of interest.
For each sample, we estimated this by subtracting the mass and volume of the ice to give the firn density in the absence of
ice content. We used a 30-cm moving average of total ice content and density in order to smooth out a possible error of ±10
cm in assigning the location of the ice features within the stratigraphy. Each sample had a measured bulk density, $\rho_b$, which
we assume resulted from a binary mixture of ice and firn, with densities $\rho_f$ and $\rho_i$. Ice and firn fractions, $F_i$ and $F_f$, were
defined with $F_i + F_f = 1$. The background firn density was then calculated following:
$$\rho_f = (\rho_b - \rho_i\, F_i)/F_f. \tag{5}$$

In cases where there was no ice fraction ($F_i = 0$), $\rho_f = \rho_b$. Ice layers and lenses were assumed to have a density of 874 ±35
kg m$^{-3}$, based on the average density of firn-core sections that were 100% ice in Greenland (873 kg m$^{-3}$) and Devon Ice Cap
(875 kg m$^{-3}$) (Bezeau et al., 2013; Machguth et al., 2016). This is different from the 917 kg m$^{-3}$ upper bound used in the
outlier analysis because that is the theoretical limit for pure ice, whereas 874 kg m$^{-3}$ is based on measured field data which
includes observed ice layers and lenses which have small bubbles and imperfections in them.

There is surface lowering associated with melting but without associated mass loss, due to subsurface refreezing. This
surface lowering is an 'apparent ablation' in airborne or satellite altimetry signals. We calculated this for each core section
using the background firn density, $\rho_f$, and length of the section, $L$. The 'thinning' or surface lowering of a given core
section, $\Delta L$, was estimated by reverting the ice to the density of the background firn, following:

$$\Delta L = L\left[\left(F_f + \frac{\rho_i\, F_i}{\rho_f}\right) - 1\right]. \tag{6}$$

Summed over the full firn column, this gives the total surface lowering associated with meltwater that percolates and
refreezes, with no actual loss of mass.



**3.3 Historical measurements**

As part of an expedition undertaken by the IRRP, Grew and Mellor (1966) measured snow density and temperature to a depth of 15 m at the Divide site on July 23, 1964 (Fig. 1). The first ~4 m were measured in a snow pit, while the remaining ~11 m were based on measurements of a core drilled with a Cold Regions Research and Engineering Laboratory (CRREL) coring auger. The original data is not available, so values were reconstructed based on digitization of the density plot provided in Figure 4 of Grew and Mellor (1966). This digitization was undertaken with WebPlot Digitizer 4.3 (Rohatgi, 2020), and has an estimated error of ±2 kg m$^{-3}$ for density and ±0.01 m for depth. Errors were calculated by clicking the same point 25 times and evaluating the variability of the points (i.e., the standard deviation).

From July 14-17, 2006, snow density and temperature measurements were recorded every 10 cm to a depth of 10.4 m at the Copland Camp as part of a University of Ottawa field class. Measurements from 0 to 5.4 m were recorded in a snow pit, while those from 5.5 to 10.4 m were based on a core recovered with a Kovacs Mark II coring system (Kovacs Enterprises, Oregon, USA). Density measurements in the snow pit were undertaken with a 250 cm$^3$ RIP 2 Cutter (Snowmetrics, Colorado, USA), and in the ice core by measuring and weighing core sections and using Eq. (1). Errors in the density measurements were determined from Eq. (4), and verified against density values recorded in a second snow pit dug to a depth of 4.0 m, approximately 2 m away from the first. All temperature measurements were undertaken with a Thermor PS100 digital stem thermometer with an accuracy of ±0.5°C.

Annual snow accumulation at the Copland Camp was measured between 2004-2011 with a Campbell Scientific SR50 Sonic Ranging Sensor mounted on a cross-arm on a vertical steel pole drilled into the firn. The SR50 was connected to a Campbell Scientific CR10X logger, and included a correction for the change in speed of sound with air temperature. The mounting pole was raised annually to keep it above the snow surface, and densities recorded in snow pits collected during annual University of Ottawa field classes (typically in early July) were used to convert the SR50 depth measurements into w.e. values.

**3.4 Energy balance and firn modelling**

ERA climate reanalyses were used to examine changes in climate and annual surface melting at the study site since the 1960s, coupled with a firn model to simulate the decadal evolution of firn temperature, hydrology, ice content, and density. Daily melt rates were calculated from 1965 to 2019 using a surface energy balance model (Ebrahimi and Marshall, 2016), coupled to a subsurface model of coupled thermal and hydrological evolution in the snow and firn (Samimi et al., 2020). The model calculates the surface energy budget and snow melt based on incoming shortwave and longwave radiation, temperature, relative humidity, wind speed, and air pressure, with internal parameterizations of surface albedo evolution and outgoing longwave radiation. Conductive heat flux to the snow surface and snow surface temperatures are simulated within the subsurface snow/firn model. Snow and firn densification are parameterized following Vionnet el al. (2012) for the firn

matrix ('background firn'), with bulk density including the additional mass of any ice or water content. Details of the model
are provided in the Supplementary Information.

Meteorological inputs for the surface energy balance model were derived from the ERA5 climate reanalysis for the period
1979 to 2019 (Hersbach et al., 2020), and extended back to 1965 using the ERA 20[th] century reanalysis (ERA20c; Poli et al.,
2016). ERA5 outputs are at a resolution of 0.25° latitude and longitude, and data for our analysis was averaged from ERA5
grid cells located at (60.75°N, 139.75°W) and (60.75°N, 139.5°W). ERA20c data are at 1° latitude and longitude resolution,
and we interpolated meteorological conditions to the upper Kaskawulsh Glacier from the four model grid cells at 60° to
61°N and 139° to 140°W. ERA20c fields were homogenized with ERA5 through bias adjustments for two years of overlap
in the reanalyses, 1979 and 1980, with ERA5 assumed to be the more accurate reconstruction. Monthly bias adjustments
based on this period of overlap were then applied to the ERA20c data from 1965 to 1978.

The reanalysis data represent the climatology over the region of the upper Kaskawulsh-Hubbard divide (i.e., a 0.25° grid
cell), and are not specific to our core site. The firn modelling is therefore taken to be generally applicable for this upper
plateau region. ERA air pressure and 2-m temperature and humidity fields were bias-adjusted to the specific elevation of our
core site, 2640 m (see the Supplementary Information). ERA5 temperature fields were evaluated against Copland weather
station data from 2014-2018, which indicate a small (0.6°C) cold bias in the ERA5 data for average summer (JJA)
temperatures over this period. ERA temperatures were further bias-adjusted by this amount. Our firn core site, the Copland
weather station, Copland Camp, and IRRP research sites all fall within the same ERA5 grid cell, and we make the
assumption that climate conditions are similar for similar elevations and glaciological settings within this region.

Surface energy balance and melt were calculated every 30 minutes, using mean daily meteorological forcing from ERA and
a parameterization of the diurnal cycles of temperature and incoming shortwave radiation (Ebrahimi and Marshall, 2016).
Snow accumulation is based on the ERA total precipitation, with a constant scaling factor of 1.6 in order to give
representative annual totals. With this scaling, mean annual precipitation at the site (± 1 standard deviation) was $1.83 \pm 0.32$
m w.e. yr$^{-1}$. Snow is updated monthly in the numerical simulation. We neglect rainfall, as we don't have a good constraint on
how much summer precipitation falls as rain and this will not be reliably predicted in the climate reanalysis. Summer
temperatures are cool (mean value of −2.4°C) and during our experience working at the Copland Camp in the month of July,
we have experienced numerous snow events but no rainfall. While rain must occur from time to time, we believe it to be rare
at the study site.

Subsurface temperatures were modelled for a 35-m firn column, with a simple model for meltwater percolation that accounts
for meltwater refreezing and the associated latent heat release where snow or firn is below 0°C (Samimi and Marshall, 2017;
Samimi et al., 2020). For the current study, we discretize the snow and firn into 58 layers from 0.1 to 1 m in thickness, with
higher resolution near the surface. The firn model is coupled with the surface energy balance model, solving for the firn
thermodynamic and hydrological evolution at 30-minute time steps for the period 1965 to 2019. The subsurface temperature
evolution includes vertical heat conduction and latent heat release from refreezing. Heat advection associated with snow
accumulation is neglected. When subsurface temperatures reach 0°C, liquid water is retained or percolates to depth,
following a Darcian parameterization for water flux: $q_w = -k_h \nabla \phi$, for hydraulic conductivity $k_h$ and hydraulic potential $\phi$
(Samimi and Marshall, 2017). For the numerical experiments in this study we set $k_h = 10^{-5}$ m s$^{-1}$ in snow and $10^{-6}$ m s$^{-1}$ for
snow and firn, respectively. Capillary water retention is calculated following Coléou and Lesaffre (1998). The default model
parameters are based on calibration at DYE-2, Greenland, in the percolation zone of the southern Greenland Ice Sheet
(Samimi et al., 2020). A broader range of model parameters is explored in sensitivity analyses presented in the
Supplementary Information.

The model is 'spun up' through a 30-year simulation with perpetual 1965 climate forcing (i.e., running through 30 annual
cycles with 1965 climate conditions). This provides the initial temperature, density, and ice-layer structure within the firn
column. Ideally, a spin-up simulation forced by the historical meteorological conditions (e.g., the years 1935-1965) would be
preferable to assuming a perpetual climatology from a single year. Mean annual and mean summer temperatures were
normal in 1965, so the model results are not strongly sensitive to this assumption (discussed in the results), but we explore a
range of numerical experiments to examine the model sensitivity to these initial conditions and the spin-up assumptions.

**4 Results**
**4.1 Ice core density**
The density data are plotted in Figure 2, fitted with a logarithmic curve to quantitatively compare our two cores. The first 4.2
m of both 2018 cores was dry and had an average density of 450 ±21 kg m$^{-3}$, with no ice content. At 4.2 m there was a
significant ice crust, with large crystal size, rounded grains and high impurity content, which was assumed to represent the
last summer surface (LSS) from 2017. The snow above this LSS layer was therefore classified as seasonal snow. In this
section we focus on the firn characteristics below the LSS, so our discussion is centered on the core recovered between 4.2
and 36.6 m below the surface for Core 1 (i.e., total firn length of 32.4 m), and between 4.2 and 21.6 m below the surface for
Core 2 (i.e., total firn length of 17.4 m). For consistency, we reference all depths to the seasonal snow surface throughout
this paper.

In the upper 10 m of firn (4.2 to 14.2 m below the surface; Table 1), Cores 1 and 2 had average densities of 588 ±8 kg m$^{-3}$
and 572 ±7 kg m$^{-3}$, respectively, giving an overall average density of 580 ±5 kg m$^{-3}$. Over the upper 17.4 m of firn in each

core (4.2 m to 21.6 m below the surface; the depth to the bottom of Core 2), Kaskawulsh firn had an average density of 632 ±4 kg m$^{-3}$. The full 32.4 m of firn at Core 1 (4.2 to 36.6 m below the surface) had an average density of 698 ±5 kg m$^{-3}$. Ice content generally increased with depth in the upper ~25 m of the core, but deeper sections were less icy (Table 1). The bottom 5 m of firn in Core 1 had an average density of 826 ±13 kg m$^{-3}$, but with no identified ice layers. Based on the high density and texture of this deep firn, along with the presence of liquid water in the deepest sections of the core, we believe that we drilled to near the base of the firn at the core site, but cannot confirm this as we halted drilling before reaching glacier ice.

Total ice content in the 32.2 m firn portion of Core 1 (4.2 to 36.6 m below the surface) was 2.33 ± 0.26 m of ice or 2.67 ± 0.24 m w.e. This is equivalent to 7.2% by volume and 11.9% by mass (Table 1). Using Eq. (5) and the values for ice content in Core 1, we estimate a background firn density of 676 ±6 kg m$^{-3}$ for the full column of firn, 3.2% less than the bulk density of the firn (Table 1). The two cores had very similar bulk and background densities over the upper 10 m of firn (4.2 to 14.2 m below the surface) and 17.4 m (4.2 to 21.6 m below the surface), where a direct comparison was possible. The total water equivalent of firn in Core 1 was calculated to be $w_c$ = 22.5 ±0.2 m w.e.

**4.2 Ice core stratigraphy**

The stratigraphy of the 2018 cores indicates numerous ice layers as well as melt-affected firn, distinguished by a lack of grain boundaries or opaque, bubbly firn. The first 4.2 m comprised the seasonal snowpack, with firn below. Within the first 6 m below the surface there were several small ice layers (<2.5 cm thick), interpreted as wind crusts (Figure 3). Several thick (>10 cm) ice layers were found between 6 and 26 m depth (1.8 to 21.8 m in the firn). The largest ice layer in Core 1 was 22 cm thick, found at 14.1 m (9.9 m in the firn). At 26.4 m (22.2 m in the firn) the ice layers and lenses disappeared. Below this the firn was almost entirely meltwater-affected, based on its appearance and texture, but without the quantity of ice lenses or ice layers that were present in the first 25 m. We interpret this section of the core as infiltration ice, consisting of water-saturated firn that has experienced refreezing. At 30 m depth (25.8 m in the firn), the meltwater effects were absent and there were two small ice layers and an ice lens. At 30.6 m depth the firn was melt-affected again. From 34.5 to 36.6 m (30.3 to 32.4 m in firn) the core sections expelled liquid water as they were extracted from the core barrel.

In Core 2 there were numerous ice layers starting at a depth of 3.8 m, and below 4.4 m (0.2 m in the firn) the core was meltwater-affected. There was a thick ice layer at 6.6 m (2.4 m in the firn) that was 30 cm lower than a similar ice layer in Core 1 at 6.3 m. There were numerous melt-affected layers between ice lenses much closer to the surface in Core 2 than Core 1. In Core 1 there were several ice layers at ~10 m depth (5.8 m in the firn), but these layers were not present in Core 2. At 14.4 m (10.2 m in the firn) another section of the firn had numerous ice layers (~20-30 cm deeper than recorded in Core 1), and at 14.6 m the thickest ice layer was encountered (12 cm), corresponding well with the thickest layer in Core 1.

Between 16 and 21.5 m (11.8 to 17.3 m in the firn) the core was melt-affected. We attribute differences between Core 1 and
Core 2 stratigraphy to uncertainty in the depth of features (as discussed in Section 3.2), and horizontal variability in
meltwater infiltration, which is known to occur at length scales less than 1 m (Parry et al., 2007; Harper et al., 2011). Spatial
heterogeneity in firn is common in areas with high surface melt due to differential melting and percolation that is complex
due to the presence of sastrugi and wind crusts, different permeability of the snow and firn, and vertical piping mechanisms
(Marchenko et al., 2017; Parry et al., 2007).

**4.3 Changes in firn characteristics over time**
The firn in the accumulation area of Kaskawulsh Glacier has become denser since 1964 (Figure 4a). The mean density of the
upper 7 m of firn was 516 kg m$^{-3}$ in 1964 (3.3 to 10.3 m below the surface), 590 kg m$^{-3}$ in 2006 (3.5 to 10.5 m below the
surface) and 549 kg m$^{-3}$ in 2018 (4.4 to 11.4 m below the surface). The difference between the average densities from the
upper 7 m of the 1964 and 2018 core is 33 kg m$^{-3}$, which is an increase of ~7%. It is difficult to assess whether firn
temperatures have changed over this time, as limited data are available from below the depth of the annual temperature wave
(~10 m for heat diffusion, and deeper than this with the effects of subsurface meltwater infiltration and latent heat release).
Borehole temperature records from Grew and Mellor (1966) indicate temperate (0°C) conditions at 15-m depth in the
summer of 1964, which suggests that deep temperate firn may have existed at this site in the 1960s. This supports the
assumption that Kaskawulsh Glacier is temperate (Foy et al., 2011), despite mean annual air temperatures of about –11°C on
the upper glacier.

Accumulation data from the IRRP A site, Copland Camp and our 2018 measurements do not show any evidence for a
significant change over time, although there can be high interannual variability. At IRRP A, Wagner (1969) reported values
between 1.3 m to 1.9 m w.e. yr$^{-1}$ for 1963. Marcus and Ragle (1970) measured a winter snow accumulation of 1.6 m w.e.
from 1964-1965. Holdsworth (1965) reported an estimated mean annual accumulation rate of 1.8 m w.e. yr$^{-1}$ in the early
1960s (year not specified) (Holdsworth, 1965). Yearly snow accumulation data from 2004-2011 collected with the SR50 at
Copland Camp indicate a mean annual accumulation rate of 1.77 m w.e. yr$^{-1}$, with variations between 1.3 and 2.4 m w.e. yr$^{-1}$.
The seasonal snowpack at our drill site was 4.2 m in May 2018, with an average snow density of 440 kg m$^{-3}$, giving a total
accumulation of 1.85 m w.e. for 2017-18.

Based on the above review, we adopt an estimate of $\bar{a}$ = 1.8 ±0.2 m w.e. yr$^{-1}$ for the net accumulation from 2005 to 2018.
Using this value, the firn layer of Core 1 represents 12.5 ±1.4 years of net accumulation (i.e., 2005-2017), or 13.2 ±1.4 years,
if the seasonal snowpack on top is counted. Over 12.5 years, the total measured ice content of 2.67 m w.e. in the firn equates
to an average meltwater refreezing rate of 0.22 m w.e. yr$^{-1}$.

## 4.4 Surface energy balance and firn modelling


Reconstructed air temperature, melt, and firn trends from 1965-2019 are shown in Figure 5. Summer air temperature from
the reanalysis (Figure 5A) shows a modest but statistically significant increase over the study period, with a trend of +0.07°C
decade$^{-1}$. Table 2 reports changes in meteorological, energy balance, and modelled firn conditions over this time. Specific
humidity and incoming longwave radiation increase markedly over the 55 years, with trends of +0.1 g kg$^{-1}$ decade$^{-1}$ and +3.5
W m$^{-2}$ decade$^{-1}$, respectively. This echoes the findings of Williamson et al. (2020), who report decadal-scale, high-elevation
warming in the St. Elias Mountains in association with increases in atmospheric water vapour and longwave radiation. These
trends augment the net energy available for melt, through increases in both the net radiation and latent heat flux. Modelled
annual melt averaged 230±210 mm w.e. yr$^{-1}$ from 1965 to 2019 and 380±310 mm w.e. yr$^{-1}$ from 2005 to 2017, 70% higher
than the long-term average. The latter interval represents the approximate period of record of the firn core. The trend in
surface melting is +62 mm w.e. yr$^{-1}$ decade$^{-1}$ from 1965 to 2019 (Figure 5B). The summer of 2013 was exceptional; it had
the warmest summer temperatures on record, $T_{JJA} = -0.7$°C, with 895 mm w.e. of meltwater (Table 2).

Within the model, 91% of the surface meltwater refreezes in the firn over the period 1965-2019, with 100% of it refreezing
in cool summers when meltwater generation is limited. Meltwater that does not refreeze percolates to depth in the firn
model. Figure 5B plots the annual melting minus refreezing, with positive values indicating deep percolation. If the firn is
temperate (0°C), meltwater can percolate through the entire depth of the firn column (35 m), where it is permitted to "drain"
through the lowest layer; this water leaves the system and is considered as runoff. Porewater in the firn can also refreeze in
the subsequent winter, to the depth of the winter cold wave (~8 m), accounting for the negative values in Figure 5B. This is
percolated meltwater that refreezes within the firn column in the following calendar year. Complete meltwater retention is
typical for most of the period from 1965 to the early 2010s, but there is a marked increase in modelled runoff over the last
decade (Figure 5B), indicating drainage through the full 35-m firn column. Only 72% of the surface melt refroze during the
period 2005-2017, with a five-fold increase in meltwater drainage, from an average of −20±120 mm w.e. yr$^{-1}$ from 1965-
2019 to −105±220 mm w.e. yr$^{-1}$ from 2005-2017. Meltwater that drains to the deep firn is portioned between porewater
storage (meltwater retention) and deep drainage (mass loss). This partitioning was almost equal in the firn model over the
period 2005-2017, with an average of 52 mm w.e. yr$^{-1}$ stored as liquid water in the deep firn and 54 mm w.e. yr$^{-1}$ of runoff:
water that drains out through the bottom layer of the firn, leaving the system. This equates to a total meltwater retention of
86% as either liquid water or refrozen ice, with a mass loss representing 14% of the summer melt.

Summers with high amounts of surface melt produce greater refreezing and warming of the snow and firn, eventually
overwhelming the content and enabling deep percolation and drainage. Figures 5C and 5D plot the modelled evolution of the
firn temperatures and the wetting and melting fronts, which closely coincide. Snow and firn temperatures in Figure 5C are
mean annual values at the snow surface (the upper 0.1 m), and at 10, 20, and 35 m depth. For a purely conductive

environment, ~10 m represents the depth of the annual temperature wave (Cuffey and Paterson, 2010), but latent heat release from meltwater refreezing warms the subsurface and causes a deeper influence of surface conditions, such that 10-m temperatures are highly variable (Table 2). The modelled wetting and melting fronts in Figure 5D suggest dramatic recent developments in firn thermal and hydrological structure at the Kaskawulsh drill site, with a regime shift in the firn structure over the period 2013-2017. This is consistent with the birth of a deep PFA at this time. Figure 6 plots the full subsurface temperature evolution over the period 1965-2019, showing the typical seasonal evolution of firn temperatures and the unusual nature of the hydrological breakthrough event that began in 2013 and persists through 2019. Figures 5E and 5F plot the modelled increases in average firn density and total firn ice content from 1965-2019. The average firn density in the model is 682 kg m$^{-3}$ in 2018, compared with 698$\pm$5 kg m$^{-3}$ measured in Core 1. The modelled firn densification since the 1960s roughly matches the observed density trend.

The model results in Figures 5 and 6 are for the 'reference' 1965-2019 ERA climatological forcing and firn model parameters. These are the direct ERA climate fields, bias-adjusted to represent the elevation of the core site and to give consistency with the regional Copland weather station data (2014-2018). The weather station has a similar elevation and topoclimatic environment and is about 11 km from the core site, falling within the same ERA5 grid cell. Firn model settings are based on calibrations against field data at DYE-2, Greenland, within the percolation zone of the southern Greenland Ice Sheet (Samimi et al., 2020), but we have no local field calibration of these model parameters. There are therefore uncertainties within both the climate forcing and the model parameters and assumptions. The Supplemental Information examines the sensitivity of model results to several important meteorological inputs and model parameters, as well as the strategy adopted for the model spin-up.

Selected results of the sensitivity tests are plotted in Figure 7, indicating the wide range of model behaviour that is possible with perturbations to the model inputs, parameter settings, and spin-up assumptions. An air temperature anomaly of $\pm1$°C applied to the reference ERA climatology gives very different firn evolutions from 1965-2019, with warmer temperatures driving a shift to temperate firn conditions in the late 1980s (Figures 7A and C). Warming of 2°C gives temperate firn for the entire period. In the other direction, a temperature anomaly of $-1$°C is sufficient to maintain polythermal conditions at the site, precluding the development of deep temperate firn or a PFA. Similar results are obtained with perturbations of $\pm10$ W m$^{-2}$ to the incoming longwave radiation (Supplemental Information). Increases in meltwater infiltration that are stimulated by lower values of the irreducible water content ($\theta_{wi} < 0.025$) have a similar effect to warming, promoting meltwater infiltration, firn warming, and the earlier development of temperate firn.

The simulations are also sensitive to the initial conditions (Figures 7B and D). Given evidence from Grew and Mellor (1966) that firn at 15-m depth was temperate in the mid-1960s near our core site, we introduce temperature anomalies from +0.5 to

+2°C to the spin-up climatology. A perturbation of +1.5°C creates temperate conditions to 12-m depth, and +2°C is
sufficient to create deep temperate firn which persists for several years (Figure 7D). Firn refreezes in the 1970s within the
model, and eventually follows a similar path to the reference simulation, but with a memory of warmer initial firn
temperatures. This permits a more rapid transition (or return) to deep temperate conditions spurred by the heavy melt season
in 2013. Overall, the model sensitivities in Figure 7 indicate that a wide range of model solutions is possible at this site,
indicating that Kaskawulsh Glacier firn is very close to the threshold for either temperate or polythermal conditions. We
discuss this further below.

The temperature forcing required to induce temperature firn the 1960s is relatively strong. The year 1965 that is used for the
model initialization is representative of the long-term mean climatology of the site, with mean summer and annual air
temperatures of −2.5°C and −10.8°C. This compares with averages of −2.4 ± 0.8°C and −10.7 ± 0.9°C for the period 1965-
2019 (Table 2).  Incoming solar and longwave radiation in summer 1965 averaged 304 and 240 $W\,m^{-2}$, compared with long-
term averages of 298 and 255 $W\,m^{-2}$. Net energy and melt were slightly lower than the long-term average, due to low
incoming longwave radiation, but overall, 1965 was a typical year. A warm anomaly of +2°C represents 2.5 standard
deviations above normal, giving a mean summer temperature of −0.5°C; this would represent the warmest summer on
record.

The initial firn density and ice content are relatively high when we force the model to produce temperate firn conditions in
the mid-1960s through the +2°C air temperature anomaly in the model spin-up. Values in 1965 are 679 $kg\,m^{-3}$ and 2.8 m,
compared with reference model values of 641 $kg\,m^{-3}$ and 0.7 m. Figure 8 plots the subsequent firn temperature and density
evolution if the +2°C temperature anomaly is maintained from 1965 to 2019 and in the case where the model forcing is
restored to the reference ERA climatology from 1965 to 2019. Subsurface temperature and density evolutions in the latter
case parallel that of the reference model after a transient adjustment period of about a decade, while the perpetual +2°C
anomaly maintains dense and temperate firn. The decadal adjustment of firn density (Figure 8B) is the 'over-turning' time of
the firn core, for downward advection of new snow and firn to 35 m depth. The temperature adjustment (Figures 8A,C) does
not follow this as it is governed by thermal diffusion time scales in the deep firn, giving a longer memory of the initial
conditions.

## 5 Discussion

### 5.1 Firn characteristics and changes over time

The accumulation area of Kaskawulsh Glacier currently has indications of widespread meltwater percolation and refreezing. Meltwater is stored within the firn as ice, as indicated by the presence of ice layers and infiltration ice, and there is liquid water at a depth of ~35 m below the surface. The density of the firn has increased by about 15% since 1964 in the first 7 m of firn, due to the increased presence of ice layers. However, the firn in 1964 was not without meltwater percolation and refreezing; Grew and Mellor (1966) note the presence of refrozen ice lenses and glands and report evidence for meltwater infiltration and refreezing at depths of ~5 m. Nevertheless, the quantity and thickness of ice layers and lenses have increased towards present day, as reflected in the changes in the stratigraphy and the density (Figure 4). The firn modelling also indicates decadal-scale increases in firn ice content and density (Table 2, Figure 5E). For the reference model parameter settings and ERA climate forcing, the model predicts a significant increase in melting (Figure 5B), driving increases in the depth of the melting and wetting fronts, meltwater percolation and runoff, and latent heat release associated with refreezing since the 1960s. This fundamentally changes the way the firn contributes to the mass balance of the glacier and englacial hydrological dynamics, as discussed further in section 5.3. There are significant decadal firn warming trends in the model (Figures 5 and 6), driven by the increases in melting and meltwater percolation. The modelling is not observationally constrained, however (Figure 7 and Supplementary Material), so the simulated firn warming is uncertain.

Increased firn meltwater and ice content, as well as potential firn warming in recent decades, will affect firn densification processes. Melting rounds snow grains and increases the rate of the first stage of densification. With enough melt to drive meltwater percolation through the snow and firn layer, meltwater can fill in air pockets and refreeze, further accelerating the transition from snow to ice (Cuffey and Paterson, 2010). The overall pattern of density measurements from 2018 resembles a logarithmic densification curve (Figure 2), as is typical for Sorge's Law of densification in dry snow (Sorge, 1935, Bader, 1954). However, with increasing meltwater percolation and refreezing effects, higher densities are common in the upper portions of the firn, as observed in our cores. Bezeau et al. (2013) report similar findings from Devon Ice Cap, where they found a depth-density reversal and suggest that Sorge's Law no longer holds in areas of significant warming. To account for this, firn densification models are being revised to address the effects of ice layers and warming temperatures on the rate of densification (Reeh, 2008; Ligtenberg et al., 2011), and other studies are revising mass balance estimates based on dynamic densification rates (e.g., Schaffer et al., 2020).

### 5.2 Perennial Firn Aquifer

We found unequivocal evidence for a deep perennial firn aquifer on the Upper Kaskawulsh Glacier, with excess water in the firn pore space below about 32 m depth. Some of this water drained during firn core acquisition (Supplemental material video 1 & 2, https://doi.org/10.5446/50918 and https://doi.org/10.5446/50919, respectively). We cannot tell whether this

PFA is a new feature at this site. Borehole temperature measurements from 1964 at a site close to our cores indicate temperate conditions at 15-m depth at this time (Grew and Mellor, 1966), and it is possible that firn has been temperate since that time, conducive to a PFA below the depth of the annual winter cold wave. There are no historical temperature measurements from greater firn depths at the site, and earlier coring efforts and radar surveys from the upper Kaskawulsh, Divide, or Eclipse sites make no comment or inference about the presence of liquid water, so we cannot attest to the age or origins of the PFA. It may well be a new feature.

The modelling results suggest that there are significant decadal increases in melting and refreezing since the 1960s at this site, driving firn warming, increased ice content, and densification (Table 2). The firn model predicts the development of wet, temperate conditions in the deep firn following the 2013 melt season, although it takes four years to fully develop (Figure 6). This was triggered by meltwater penetration to 11 m depth in 2013, which is below the depth of penetration of the winter cold wave. Temperate conditions propagated downwards in the following years and persisted to 2019, supported by several more years with above-average melting. Deep meltwater percolation during these years would support the development and recharge of a PFA or perched water table at the glacier ice-firn interface. This agrees with the stratigraphy found in the field. The presence of firn that has not been visibly affected by meltwater overlying the PFA implies that deep meltwater infiltration through vertical piping may be an important process here and may allow the PFA to be recharged in a heavy melt season. In the model, deep recharge does not occur every summer after the establishment of a temperate firn column; the summer melt still needs to break through the winter cold layer, which typically extends to 6-7 m depth (Figure 6). Also of interest in Figure 6 is a large melt event in 2007, which led to meltwater infiltration and warming to about 9-m depth. This was similar to the 2013 melt event, but the summers of 2008 to 2010 were relatively cool (average JJA temperatures of $-2.8°C$ and melting of 111 mm w.e.), leading to refreezing in the upper 9 m of firn. Thawing of the full 35-m firn column to shift it from polythermal to temperate conditions requires several years of sustained melt forcing in the model.

There are significant uncertainties in the modelling, associated with the climatological forcing, surface energy balance and firn model parameterizations, and initial conditions. The Supplemental Information explores these in detail, while Figure 7 provides an illustration of the range of simulated behaviour for different model settings. The 'reference model' results presented in Figures 5 and 6 should be seen as just one scenario, corresponding to our best estimate of the parameter settings. We lack local calibration and validation studies, so we cannot preclude different firn temperature and melt evolutions at this site, particularly given the inference of Grew and Mellor (1966) that firn at 15-m depth was temperate in the mid-1960s. The default model parameters and spin-up settings do not produce this; augmented warming or incoming radiation fluxes need to be introduced to the ERA climatology to produce temperate firn at this time. It is possible that strong melt seasons in the early 1960s created temporary temperate conditions in the upper firn column. Alternatively, the surface energy balance and firn hydrological models may underestimate the amount of melting and meltwater infiltration. The one firm conclusion is

that the climatological and glaciological conditions on the upper Kaskawulsh Glacier are very close to the tipping point
between polythermal and temperate conditions. A slight nudge to either side of the reference model settings can give either
persistently sub-zero or persistently thawed conditions in the deep firn at this site (Figures 7 and S1).

The presence of the deep PFA in 2018 indicates that it is currently temperate, despite mean annual air temperatures of about
−11°C. Meltwater refreezing releases enough latent heat to bring the firn to 0°C. All model simulations concur on this,
although the long-term evolution is uncertain. We don't know the fate of the water that drains through the firn, but the
reference model predicts a total drainage of 1.13 m w.e. over the 55-year simulation, most of this over the last decade. Some
of this is retained within the PFA, but some can be expected to run off. The water in the PFA on Kaskawulsh Glacier is
likely to be flowing, redistributing mass. The drill site was located high in the glacier's accumulation zone, with a gently
sloping surface (< 0.6°) resulting in a subtle hydraulic gradient. We likely drilled into the top of the water table of the PFA,
and with densities near the pore close-off density, it is likely the PFA does not extend much deeper. There may be
downslope flow along the firn-ice interface, as well as possible Darcian flow within the PFA itself (e.g., Christianson et al.,
554 2015).

The liquid-phase meltwater retention on Kaskawulsh is similar to the PFAs found in the high-accumulation areas of southern
Greenland and Svalbard (e.g., Miège et al., 2016; Christianson et al., 2015), and different than the water-saturated layers
commonly found on temperate glaciers. PFAs that have been studied on temperate mountain glaciers typically have a
saturated layer close to the surface (for example, 5 m below the surface at Storglaciären), have active discharge and recharge
processes (Fountain and Walder, 1998; Schneider, 1999; Glazyrin et al., 1977), and appear to experience seasonal drainage
over the winter months (Fountain, 1989, 1996; Jansson et al., 2003), likely due to high hydraulic gradients. Active water
flow in the firn has been observed in 19-m and 25-m pits at Abramov Glacier (Glazyrin et al., 1977), as well as Austfonna
ice cap in 1985 at 7 m depth where they also found sub-horizontal melt channels at 7, 15, and 30 m (Zagorodnov et al.,
2006). In 2012, "water-saturated" firn was found at 40-m depth in an ice core from Mt. Waddington, British Columbia (Neff
et al., 2012). However, they reported no significant alteration of chemistry from the melt above this layer and no additional
analysis of this layer was discussed (Neff et al., 2012). In 2015, a PFA was found on Holtedahlfonna icefield in Northwest
Svalbard (Christianson et al., 2015), and in 2019 a PFA was investigated at Lomonosovfonna ice cap approximately 100 km
to the southeast (Hawrylak and Nilsson, 2019).

According to Kuipers Munneke et al. (2014), PFA formation in Greenland is contingent upon a high annual snow
accumulation, which helps to insulate the underlying firn from the winter cold wave. Mean annual temperatures in
Greenland are well below 0°C and PFAs require latent heat release from meltwater refreezing, to warm the snow and firn to
0°C, along with meltwater penetration to depths of at least 10 m, to evade the winter cold wave (Kuipers Munneke et al.,
2014). The firn modelling suggests that meltwater penetration to depths of 10 m is rare at Kaskawulsh Glacier, but can occur
in strong melt seasons. Based on our measurements and earlier reports from the IRRP (Wood, 1963; Grew and Mellor,
1966), the estimated accumulation rate at our study site is 1.8 m w.e. yr$^{-1}$. This is similar to reported accumulation rates
where PFAs have been identified in southeastern Greenland (e.g., Miège et al., 2016). Melt rates at southeast Greenland PFA
locations are also comparable to those on the upper Kaskawulsh. Miège et al. (2016) report 0.73 m w.e. yr$^{-1}$ over the time
period 1979-2014, while Miller et al. (2020) estimated annual melt rates from 0.24-0.50 m w.e. yr$^{-1}$ in a PFA field study at
1700 m elevation in the Helheim Glacier catchment. Modelled melt rates on the upper Kaskawulsh Glacier are estimated at
$0.52\pm0.27$ m w.e. yr$^{-1}$ from 1965-2019 (Table 2). Recent (2005-2017) Kaskawulsh melt rates increased to $0.72 \pm 0.38$ m
w.e. yr$^{-1}$, similar to the long-term estimate of Miège et al. (2016) in southeast Greenland, and perhaps close to the threshold
for PFA formation and recharge.
**5.3 Implications for geodetic mass balance**
Liquid water is commonly found in the temperate firn of low- and mid- latitude mountain glaciers and plays an important
role in meltwater storage and glacier hydrology and mass balance (Fountain and Walder, 1989; Schneider, 1999). For
example, storage of meltwater in a PFA accounts for as much as 64% of internal accumulation for glaciers in Alaska and
Sweden (Trabant and Mayo, 1985; Schneider and Jansson, 2004). In general, melt can result in net surface lowering in four
main ways, but with differing impacts on mass balance: (i) melt which runs off results in direct mass loss; (ii) melt which
percolates and refreezes internally can result in surface lowering, with little to no mass loss; (iii) melt which makes it into a
PFA likely contributes to mass loss, but the storage of liquid water at the firn-ice interface delays runoff from hours to weeks
or longer (Jansson et al., 2003), and (iv) accelerated compaction of warming firn can result in accelerated surface lowering,
without any mass loss. These components can be interrelated and their relative importance depends on many factors
including spatial and temporal variations in melt, PFA thickness, the presence of ice lenses, and firn temperature, so their
effects are difficult to disentangle. We provide further information about these components below, including an assessment
of their relative importance for Kaskawulsh Glacier.

The climate reanalysis suggests that the effects of meltwater storage through refreezing or liquid retention are increasing as
the climate is warming. Geodetic mass balance measurements are compromised by climate change-induced densification
changes that are not accounted for when interpreting surface lowering of the accumulation zone (Reeh, 2008; Huss, 2013).
Mass balance studies in Greenland indicate that changing melt regimes, meltwater refreezing, and the unknown density and
pore capacity of snow and firn pose significant uncertainties when modelling the surface mass balance of ice sheets
(Lenaerts et al., 2019). Meltwater retention as porewater or refrozen ice will delay surface runoff, dependent on the water
storage characteristics of firn (e.g., pore space availability, water at interstitial grain boundaries) (Fountain and Walder,
1989; Schneider, 1999). If ice layers become too extensive or thick, they can form an 'ice slab,' a thick impermeable barrier
that leads to enhanced surface runoff (MacFerrin et al., 2019). The thickness of ice layers that prevents percolation is not
well understood. For example, in Greenland 12-cm thick ice layers were still permeable (Samimi et al., 2020) whereas Bell
et al. (2008) reported that a 1-2 cm ice layer prevented percolation at Devon Ice Cap, Canada. These phenomena and effects
are not limited to Greenland and the high Arctic. This study demonstrates that Kaskawulsh Glacier also experiences
meltwater storage in the form of ice layers and liquid water retention (as a PFA), with potentially significant recent changes
in firn structure and meltwater retention capacity. The increases in firn density and ice content found on Kaskawulsh Glacier
appear to be similar to other high-accumulation Arctic regions (Pohjola et al., 2002; De La Peña et al., 2015; Bezeau et al.,
613 2013).


The surface energy balance model is not observationally constrained at this site, so we don't have quantitative confidence in
the modelled mass balance and melt rates, but the reconstructed trends indicate a ~70% increase in summer meltwater
production at this site since the 1960s, leading to increased rates of refreezing and also the onset of meltwater runoff in
recent years. We neglected the potential influence of summer rainfall in this study, as we believe that rain events remain rare
at this site. However, they likely happen from time to time, and could become more prevalent in a warming climate. Rain
events would add sensible heat and liquid water to the snow and firn, further increasing snow and firn temperatures, water
infiltration into the firn, and meltwater runoff, where there is inadequate cold content or pore space to retain this water (as in
recent years at the site). In situ field studies are needed to confirm and constrain the meteorological and energy balance
conditions on the upper Kaskawulsh Glacier, to inform the mass balance processes for both the glacier and the PFA.

Melt totals are much less than the annual accumulation (~1.8 m w.e.), so the site remains within the accumulation area of the
glacier, with most of the meltwater refreezing. Increases in meltwater refreezing have driven decadal-scale firn warming
along with increases in ice content and firn density at this site. The modelling suggests a ~5% increase in firn density since
the 1960s and a doubling of the ice content, from 1.1 to 2.3 m over the full 35-m snow/firn column. Increases in summer
melting over the last decade are associated with meltwater infiltration and firn warming in the deep firn, with the likelihood
of meltwater runoff from the site in recent years. Within the model, 91% of surface melt refreezes over the 55-year
simulation, but this declines to 73% for the period 2005-2017. The remaining 27% drains to the deep firn through this
period, where it is either retained within the PFA or it may drain from the system. In the firn model, a total of ~0.7 m of
meltwater is stored as liquid water in the deep firn and ~0.7 m w.e. 'runs off' through the period 2005-2017, draining
through the bottom layer and leaving the system. In reality, this meltwater may drain through lateral transport in the PFA or
at the ice-firn interface.

The modelled 2018 firn core has an ice content of 2.6 m, compared to a total ice content of 2.33 ± 0.26 m measured in Core
1. The modelled ice content is a completely independent estimate but is in reasonable agreement with the firn core, giving
some confidence in the modelled refreezing. The model may slightly over-estimate the melt or the meltwater infiltration,
given that the modelled ice content is about 15% too high. However, that inference is not consistent with the apparent cold
bias in the model spin-up. Alternatively, firn in the model may be too cold through much of the simulation, causing an
overestimate of the modelled meltwater refreezing and retention capacity. If this is the case, runoff (summer mass losses)
from the site will be higher than our estimates, with negative implications for Kaskawulsh Glacier mass balance.

The accumulation zone of Kaskawulsh Glacier is estimated to have experienced a minimum of $0.73 \pm 0.23$ m of surface
lowering due to internal refreezing over the period represented by Core 1, which we estimate to be 12.5 years, or $0.06 \pm 0.02$
m yr$^{-1}$ from 2005-2017. This estimate of thinning is likely low, because neither the meltwater retention due to the infiltration
ice nor the presence of the PFA is included in this estimate. In previous measurements of surface elevation changes on
Kaskawulsh Glacier, Foy et al. (2011) found that the accumulation zone thinned by an average of 0.04-0.11 m yr$^{-1}$ from
1977-2007, with a total thinning of 1–3 m over this period. Larsen et al. (2015) reported mean elevation losses of 0-1 m yr$^{-1}$
towards the head of the glacier from 1995-2000. The thinning signal due to meltwater percolation and refreezing is within
the estimates of Foy et al. (2011) and Larsen et al. (2015), suggesting that some or all of the reported lowering could be due
to mass redistribution and not mass loss. The density of the firn has increased from 1964-2018 due to meltwater percolation
and refreezing. It is therefore likely that the surface has lowered since 1964 because of this increased densification.
**6 Conclusion**
The upper accumulation zone of Kaskawulsh Glacier firn has undergone significant changes since 1964, becoming denser
and more ice-rich. The mean density of the first 32 m of firn (4.2 to 36.2 m below the surface) was $698 \pm 5$ kg m$^{-3}$. Analysis
of historical density data indicate that the firn of Kaskawulsh Glacier has become up to 15% denser since the early 1960s,
due to the increased ice content and melt-affected firn. Increases in firn density due to meltwater refreezing over the 13-year
period represented by the firn core (2005-2018) are equivalent to a surface lowering of $0.73 \pm 0.23$ m ($0.06 \pm 0.02$ m yr$^{-1}$),
and this rate of surface lowering is likely increasing in association with the overall densification, connected to increases in
firn temperature as well as ice content. Though not observationally constrained, and therefore uncertain, the modelling
results suggest the likelihood of significant increases in melting and refreezing since the 1960s at this site, driving decadal-
scale increases in firn temperature, ice content, and density. The estimates of firn density and the evidence for densification
can help to inform geodetic mass balance measurements from this region.

Our study also illustrates a high-elevation accumulation area in the St. Elias Mountains that is undergoing a transformation
in response to climate change. The firn on upper Kaskawulsh Glacier now contains a PFA, which has likely developed over
the past decade. Firn modelling suggests that the PFA has developed in response to increased summer melting, meltwater
infiltration, and firn warming from the associated latent heat release. The Kaskawulsh Glacier PFA needs to be more widely
studied, as the spatial extent and depth of the aquifer are not yet known. Ground-penetrating radar measurements may
provide a method to investigate the spatial extent of the feature. Use of an electrothermal drill that can drill through water-
saturated firn may allow estimations of the depth of the firn aquifer, as well as subsequent studies on the potential flow of the
water within the aquifer. The firn modelling suggest that this site is experiencing meltwater runoff over the past decade, in
relation to the development of the PFA. A better understanding of this feature is needed to quantify the extent of meltwater
retention and the mass balance of Kaskawulsh Glacier. This region will likely continue to experience increasing amounts of
surface melt and refreezing within the snowpack and firn, extending to higher elevations, so there is some urgency to obtain
climate records from this region.

*Data availability*. Raw density data is available by contacting the corresponding author.

*Author contribution*. NO and AC collected field data. AC ran ion analyses, supervised the field campaign and helped with
figures. SM contributed to the design and funding of the study and was responsible for the firn modelling. BM and SM
provided supervision during the project. LC provided weather station data and contributed to the collection and interpretation
of data. NO analysed the data and wrote the manuscript, to which all co-authors contributed.

*Competing interests*. The authors declare no competing interests.

*Acknowledgements*. This work was part of a Polar Knowledge Canada Grant in support of Cryosphere-Climate Monitoring at
Kluane Lake Research Station, Yukon Territory. We acknowledge the Natural Sciences and Engineering Research Council
(NSERC) of Canada for additional financial support. We thank Parks Canada for permission to conduct this research in
Kluane National Park, under research and collection permit KLU-2018-28117. We are grateful for the field crew Étienne
Gros and Peter Moraal, Icefield Instruments Inc., for assistance with coring, and members of the University of Ottawa
Glaciology and Northern Field Research classes, particularly Jean Bjornson, for assistance with the snow pit measurements.
The Arctic Institute of North America, Kluane Lake Research Station, and Icefield Discovery supported fieldwork logistics.
We thank Kristina Miller, University of Calgary, for field support and countless glaciological discussions, Shad O'Neel and
Louis Sass of the U.S. Geological Survey for sharing firn density data from Alaska, Christian Zdanowicz, Uppsala
University, for sharing his 2004-2011 snow depth data from the upper Hubbard Glacier, and Eduard Khachatrian for
translating the Glazyrin et al. (1977) paper.

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

**Table 1:** Total ice content, ice fraction ($F_i$), bulk density ($\rho_b$), background density ($\rho_f$), and total water equivalent ($w$), for the
portion of each core below the last summer surface. Depths are reported from the May 2018 snow surface and the firn
portion of the core started at the 2017 summer surface, at 4.2 m depth.

|  | Depth below surface (m) | Total Ice content (m) | $F_i$ (% vol) | $F_i$ (% mass) | $\rho_b$ (kg m$^{-3}$) | $\rho_f$ (kg m$^{-3}$) | $w$ w.e. (m) |
|---|---|---|---|---|---|---|---|
| Core 1 | 4.2-14.2 | 0.67 ± 0.07 | 6.7 ± 0.7 | 13.0 ± 1.3 | 588 ± 8 | 565 ± 9 | 5.88 ± 0.08 |
|  | 4.2-21.6 | 1.51 ± 0.15 | 8.7 ± 0.9 | 15.6 ± 1.6 | 640 ± 6 | 613 ± 7 | 11.08 ± 0.11 |
|  | 4.2-36.6 | 2.33 ± 0.26 | 7.2 ± 0.7 | 11.9 ± 1.2 | 698 ± 5 | 676 ± 6 | 22.49 ± 0.15 |
| Core 2 | 4.2-14.2 | 0.42 ± 0.04 | 4.2 ± 0.4 | 8.4 ± 0.8 | 572 ± 7 | 556 ± 7 | 5.72 ± 0.07 |
|  | 4.2-21.6 | 0.81 ± 0.08 | 4.7 ± 0.5 | 8.5 ± 0.9 | 624 ± 5 | 609 ± 6 | 10.85 ± 0.09 |
| Average | 4.2-14.2 | 1.18 | 4.0 | 564 | 580 ± 5 | 560 ± 5 | 5.80 ± 0.05 |
|  | 4.2-21.6 |  |  |  | 632 ± 4 | 611 ± 4 | 10.97 ± 0.07 |




**Table 2:** Climate, surface energy balance, and firn conditions, 1965 to 2019, based on the ERA meteorological forcing at the core site. Decadal trends are reported from linear fits to the data. The period 1965-1975 represents the historical baseline period, when much of the work of the IRRP was completed. 2005-2017 represents the period of record of Core 1, and 2013 was an exceptional year, which potentially marked the initial development of the firn aquifer at this site. Melt and refreeze refer to the total annual melting and refreezing in the 35-m snow and firn column, 'drainage' is the total annual melt minus refreezing, and 'net melt' is the surface mass loss/drawdown associated with summer melting. Freeze-thaw cycles in the surface layer of the snow mean that some fraction of the net energy that is available for melt is directed to refrozen (i.e., recycled) meltwater. As a result, the net melt – meltwater that is available to percolate into the deeper snow- and firn-pack – is less than the total summer melt. Not that rainfall is neglected in this study, and is assumed to be negligible.

878

|  | 1965-2019 Mean ($\pm 1\ \sigma$) | Trend (decade$^{-1}$) | 1965-1975 Mean ($\pm 1\ \sigma$) | 2005-2017 Mean ($\pm 1\ \sigma$) | 2013 |
|---|---|---|---|---|---|
| *Meteorological Conditions* | | | | | |
| $T_{ann}$ (°C) | $-10.7 \pm 0.9$ | $+0.16$ | $-11.2 \pm 0.7$ | $-10.4 \pm 1.0$ | $-9.6$ |
| $T_{JJA}$ (°C) | $-2.4 \pm 0.8$ | $+0.07$ | $-2.2 \pm 0.9$ | $-2.1 \pm 0.7$ | $-0.7$ |
| $T_{SJJA}$ (°C) | $-2.3 \pm 0.8$ | $+0.29$ | $-2.9 \pm 0.9$ | $-1.8 \pm 0.6$ | $-0.8$ |
| $PDD$ (°C d) | $54 \pm 23$ | $+3.6$ | $49 \pm 16$ | $69 \pm 31$ | $123$ |
| $q_v$ (g kg$^{-1}$) | $3.7 \pm 0.2$ | $+0.10$ | $3.5 \pm 0.2$ | $3.9 \pm 0.2$ | $4.2$ |
| | | | | | |
| *Surface Energy Balance (JJA values)* | | | | | |
| $Q^*$ (W m$^{-2}$) | $18 \pm 11$ | $+3.7$ | $8 \pm 3$ | $26 \pm 13$ | $45$ |
| $Q_N$ (W m$^{-2}$) | $10 \pm 9$ | $+2.6$ | $4 \pm 3$ | $16 \pm 13$ | $37$ |
| net melt (mm w.e. yr$^{-1}$) | $230 \pm 210$ | $+62$ | $100 \pm 80$ | $380 \pm 310$ | $895$ |
| melt (mm w.e. yr$^{-1}$) | $520 \pm 270$ | $+81$ | $360 \pm 130$ | $720 \pm 375$ | $1360$ |
| refreeze (mm w.e. yr$^{-1}$) | $500 \pm 195$ | $+48$ | $360 \pm 130$ | $615 \pm 205$ | $1100$ |
| drainage (mm w.e. yr$^{-1}$) | $20 \pm 120$ | $+32$ | $0 \pm 0$ | $105 \pm 215$ | $260$ |
| | | | | | |
| *Firn Conditions* | | | | | |
| $T_1$ (°C) | $-12.8 \pm 0.9$ | $+0.2$ | $-13.3 \pm 0.8$ | $-12.4 \pm 0.9$ | $-11.5$ |
| $T_{10}$ (°C) | $-7.3 \pm 3.4$ | $+1.8$ | $-11.3 \pm 0.8$ | $-2.9 \pm 2.4$ | $-3.0$ |
| $T_{20}$ (°C) | $-7.2 \pm 3.6$ | $+2.1$ | $-12.2 \pm 0.5$ | $-3.7 \pm 2.5$ | $-4.5$ |
| $T_{35}$ (°C) | $-8.0 \pm 3.5$ | $+2.1$ | $-12.7 \pm 0.4$ | $-4.8 \pm 1.9$ | $-5.2$ |
| $z_{thaw}$ (m) | $6.8 \pm 9.4$ | $+3.6$ | $1.2 \pm 1.0$ | $13.1 \pm 12.5$ | $18.0$ |
| $E_{lat}$ (MJ m$^{-2}$) | $126 \pm 41$ | $+9.3$ | $98 \pm 30$ | $147 \pm 43$ | $258$ |
| $\rho_b$ (kg m$^{-3}$) | $655 \pm 10$ | $+4.6$ | $645 \pm 3$ | $663 \pm 12$ | $671$ |
| ice content (m) | $2.0 \pm 0.6$ | $+0.2$ | $1.1 \pm 0.3$ | $2.3 \pm 0.4$ | $2.6$ |


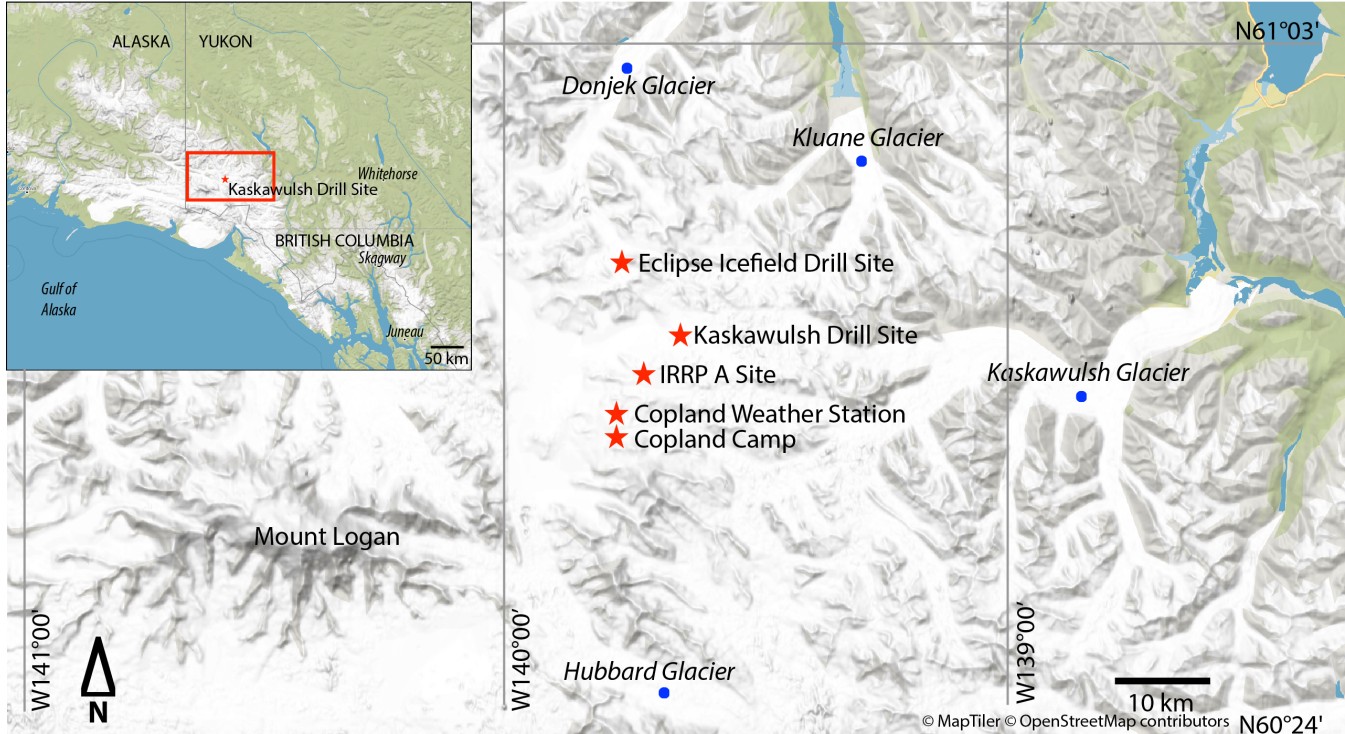

**Figure 1:** Field locations in the St. Elias Icefield, Yukon. IRRP A site is the site of the 1964 firn core that is referenced in
our study (Grew and Mellor, 1966). Base map from http://openmaptiles.org/.

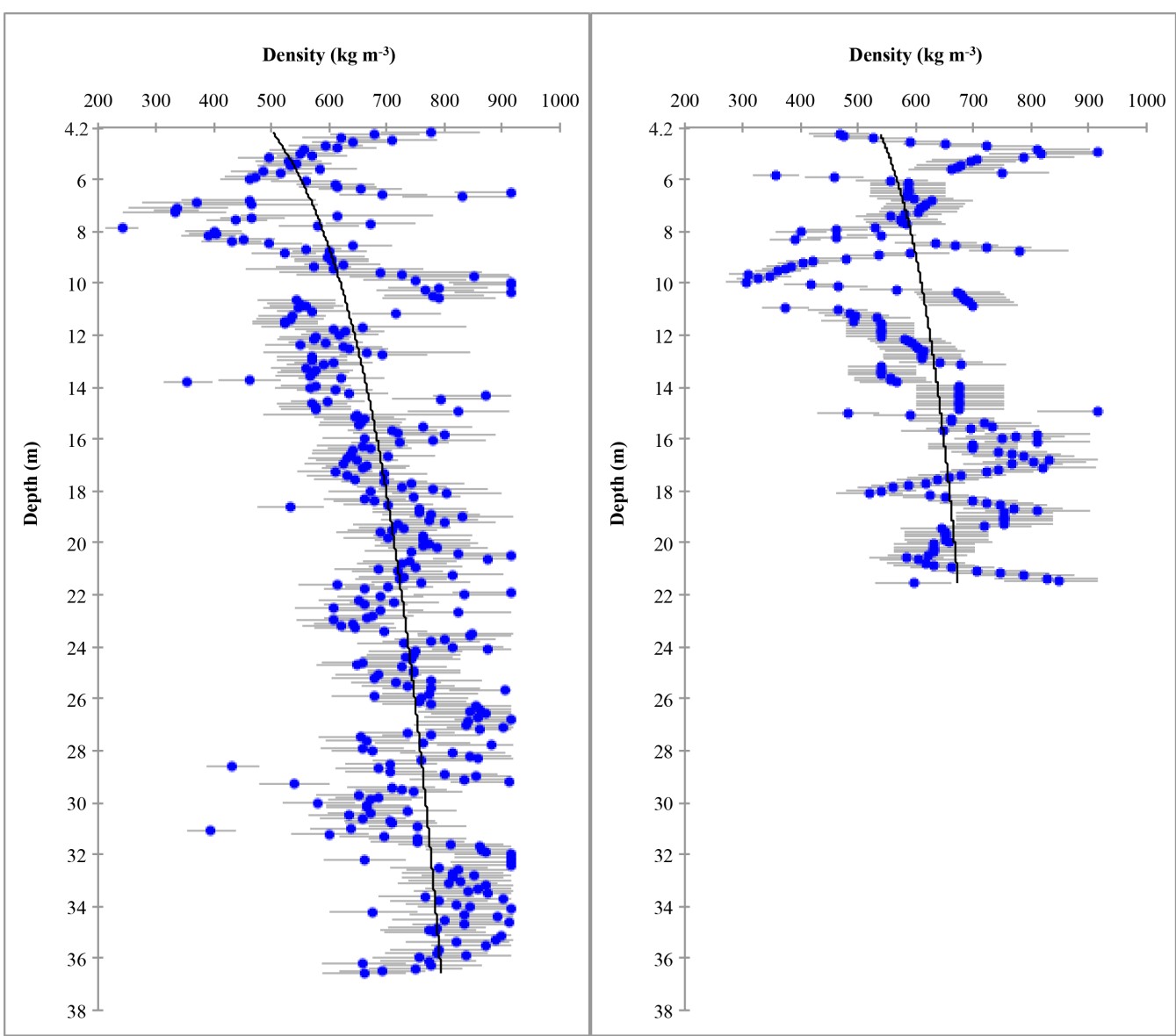

**Figure 2:** Measured firn densities of: (A) Core 1, and (B) Core 2 (May 20-24[th] 2018), with uncertainties and best-fit logarithmic curves (black line). The depth scales are truncated at the location of the last summer surface at 4.2 m depth, as the profile consisted of seasonal snow above this**.**


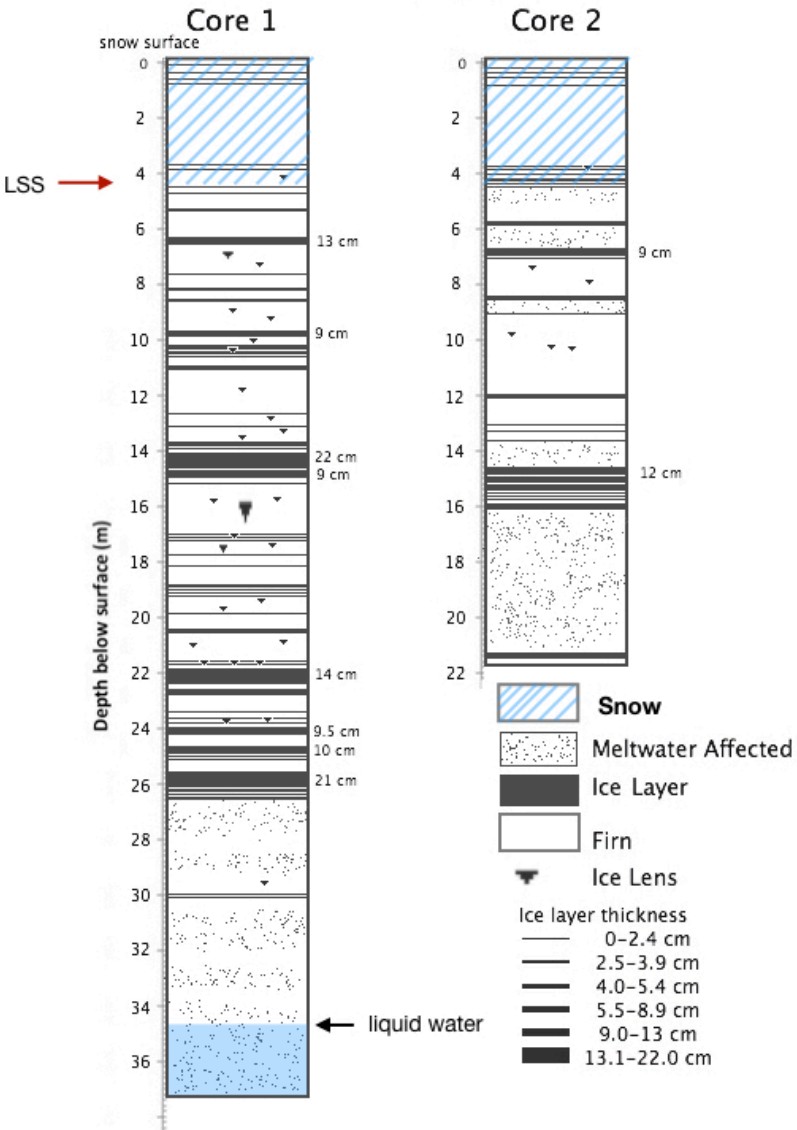


**Figure 3:** Stratigraphy of the cores collected in May 2018. LSS is the last summer surface at 4.2 m, the boundary between
seasonal snow above and firn below. Ice layer thicknesses were classified in the legend by thickness distribution. Note that
the ice layers in the first several meters of the core are interpreted as wind crusts.




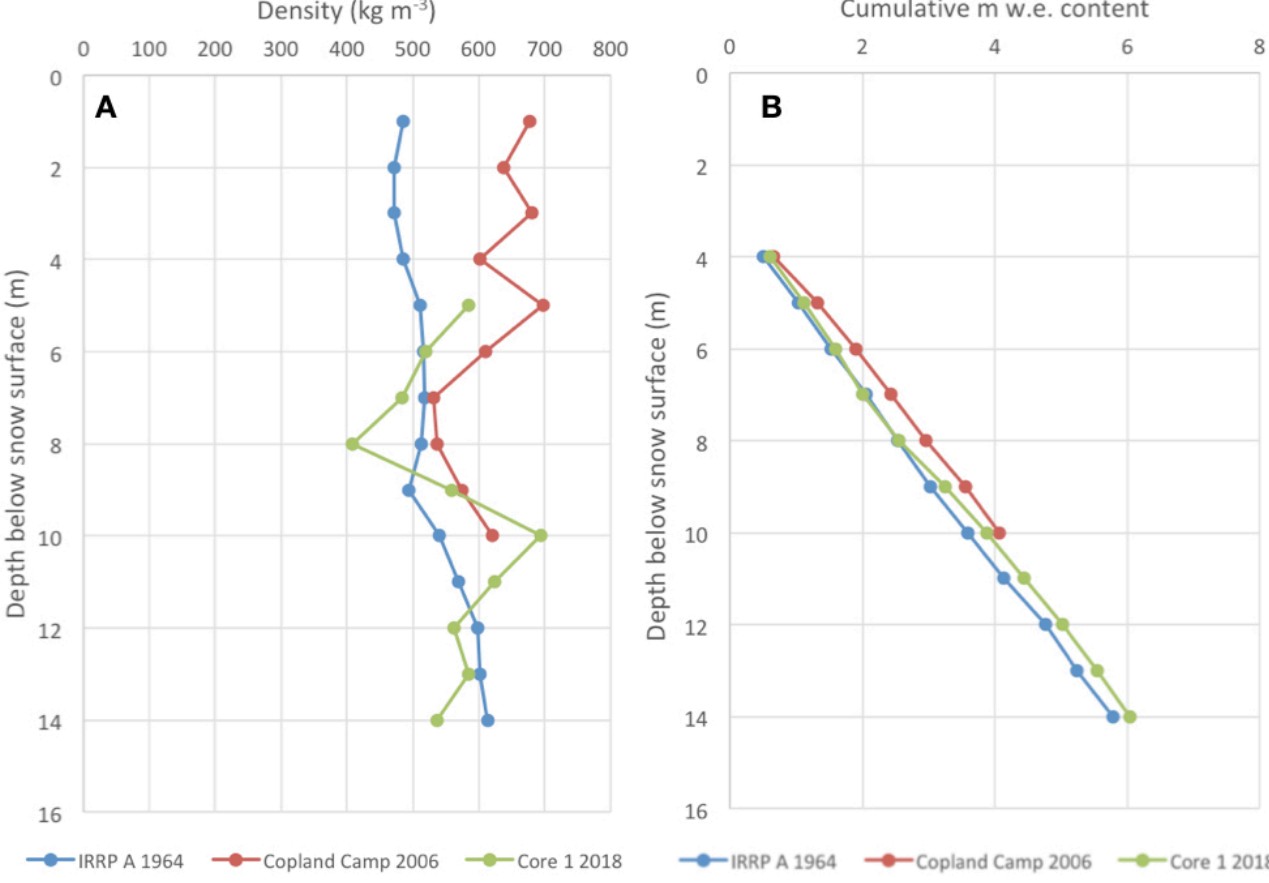



**Figure 4:** A) Comparison of densities averaged over 1 m segments at IRRP A on July 23, 1964 (Grew and Mellor, 1966; blue); at Copland Camp on July 14-17, 2006 (red); at Core 1 on May 20-24, 2018 (green). Depth of LSS (i.e., boundary between seasonal snow above and firn below) was 3.28 m in 1964, 3.50 m in 2006, and 4.22 m in 2018; the density data for 2018 begins at the LSS due to the difference in time of year of the measurements compared to the others; B) Comparison between cumulative w.e. content in the 1964, 2006 and 2018 profiles, starting at the LSS.

936

937

938

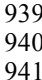

**Figure 5:** Modelled meteorological, surface mass balance, and firn conditions from 1965 to 2019: (A) Summer (JJA) air and snow-surface temperatures, °C; (B) Annual melting and 'drainage' (melting minus refreezing), mm w.e. yr$^{-1}$; (C) Annual mean snow and firn temperature at the surface (0.1 m) and at depths of 10, 20, and 35 m, °C; (D) Modelled maximum depths of the summer wetting and thawing fronts, m; (E) Average firn density for the full firn column and in the upper 20 m, kg m$^{-3}$; (F) total firn ice content, m.


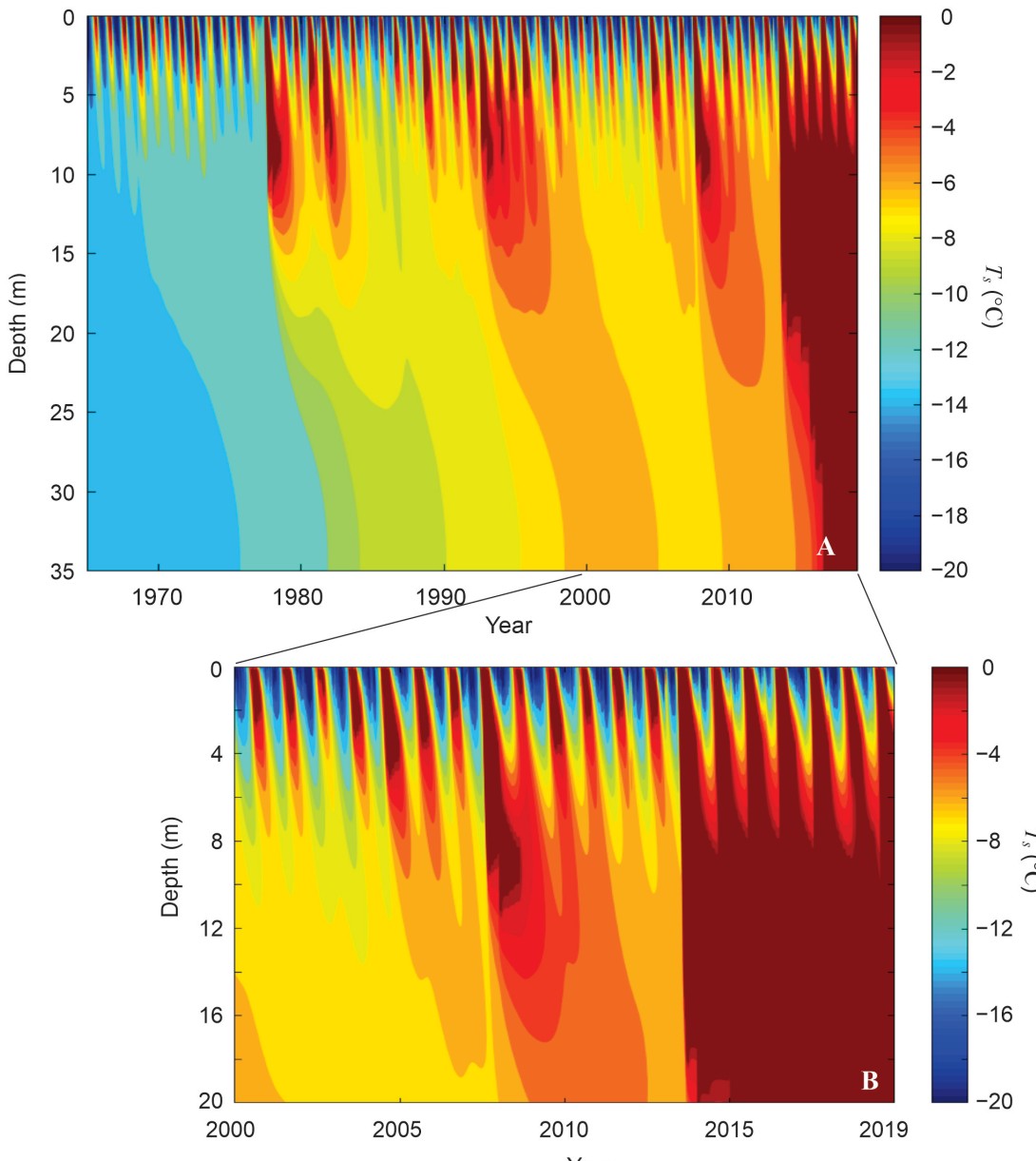

**Figure 6:** Modelled subsurface temperature evolution for the reference model climatology and parameter settings. (A) 1965-
2019, full 35-m firn column; (B) 2000-2019, upper 20 m. Deep temperate conditions conducive to a firn aquifer developed
from 2013 to 2017, in response to several subsequent summers of high melting and deep meltwater infiltration.

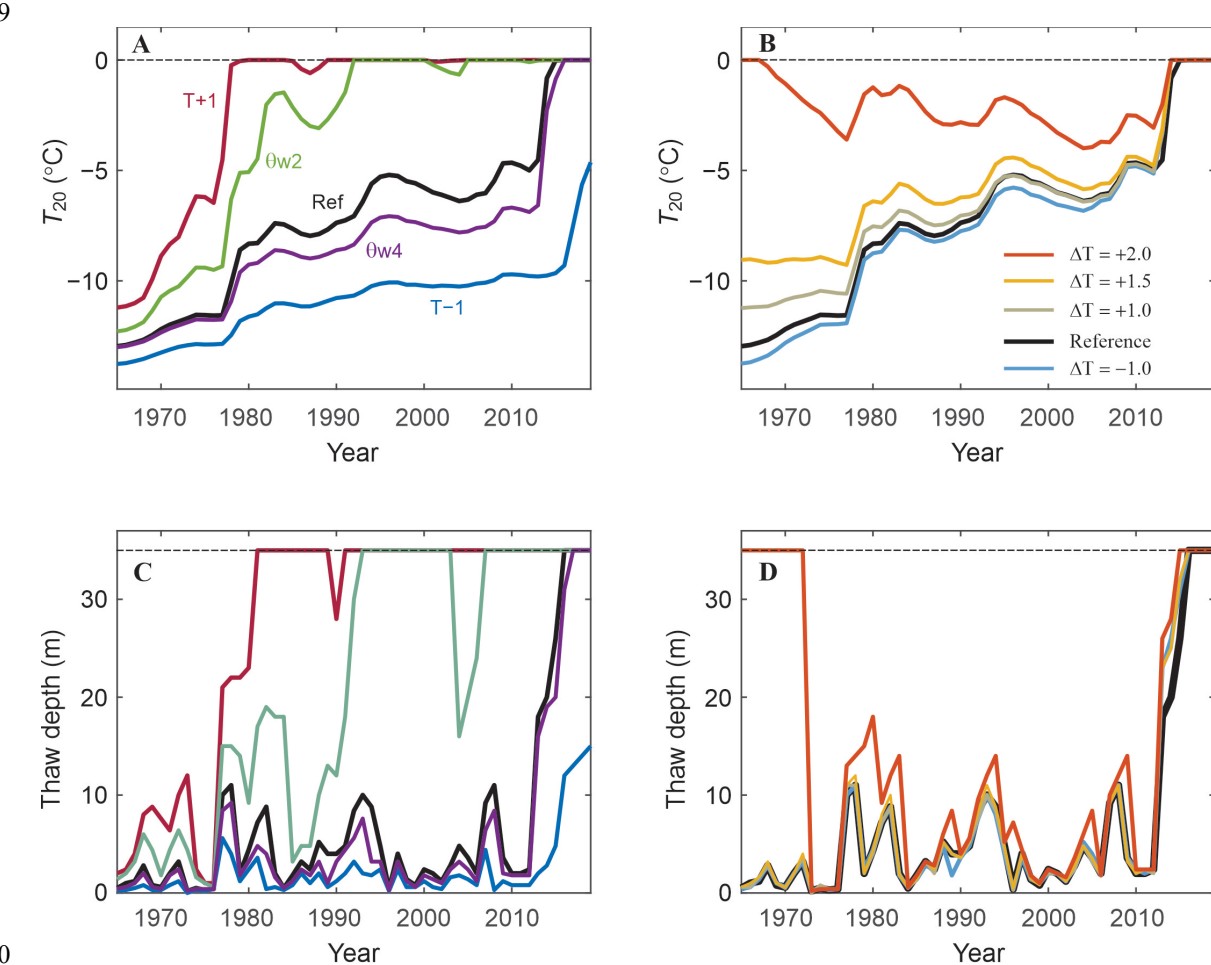

**Figure 7:** Sensitivity of the model simulations to (A, C) meteorological forcing and firn model parameters, and (B, D) initial conditions, through different model spin-up settings. (A) Mean annual 20-m temperatures and (C) seasonal thaw depths from 1965-2019 for the reference model and for sensitivity experiments with ±1°C and for irreducible water contents of 0.02 (θw2) and 0.04 (θw4). The line colours in (A) also apply to (C). An extended set of sensitivity tests is presented in the supplementary material. (B) 20-m temperatures and (D) thaw depths from 1965-2019 after a 30-year spin-up with perpetual 1965 climatology (the reference model) and imposed temperature anomalies of 1, 1.5, 2, and 2.5°C for the spin-up. The colour legend for (C) and (D) is indicated in (B).

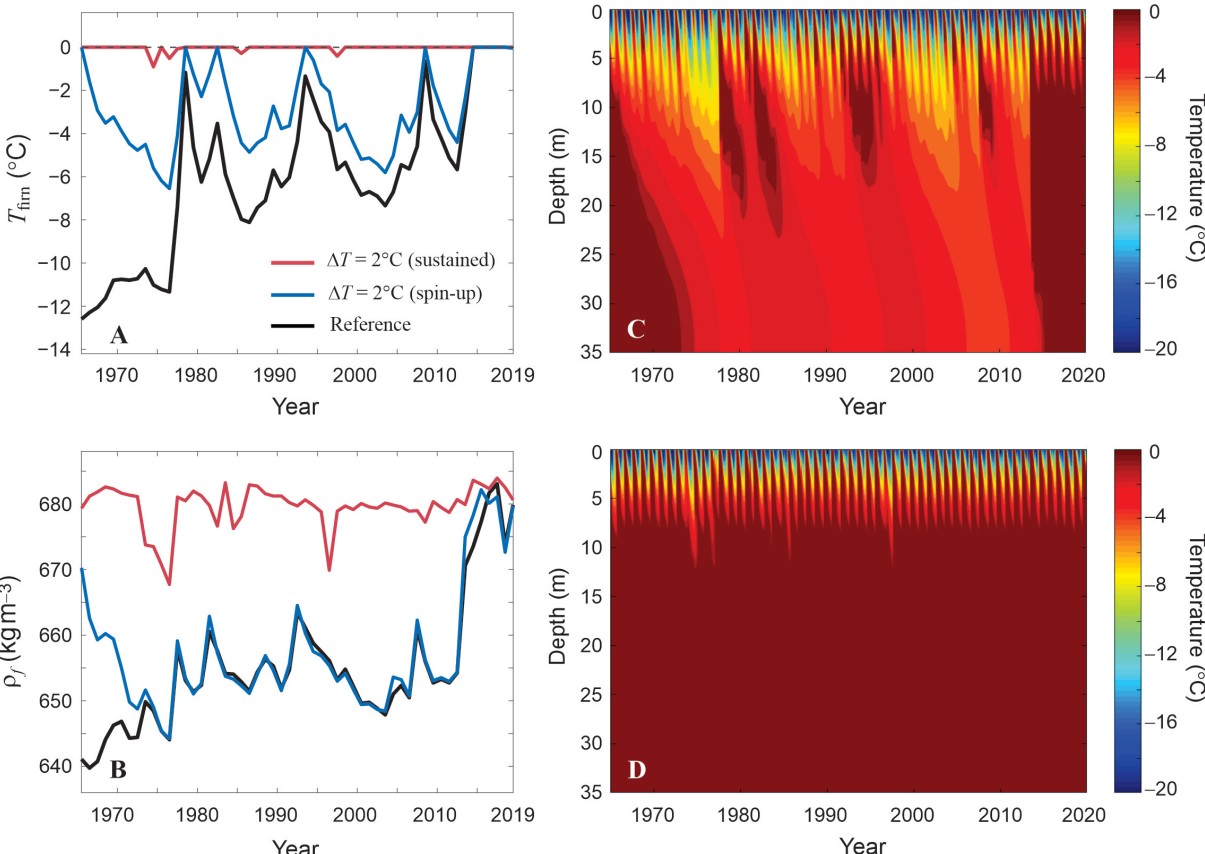


**Figure 8:** Modelled (A) 10-m firn temperature and (B) average firn density for the reference model, for a 2°C temperature
anomaly for the spin-up, and for a sustained temperature anomaly of +2°C. (C, D) Firn temperature evolution for (C) the
warm spin-up, followed by the reference climatology, and (D) sustained 2°C temperature anomalies.