# Peer review of "Evolution of the firn pack of Kaskawulsh Glacier, Yukon: meltwater effects, densification, and the development of a perennial firn aquifer"

_The Cryosphere, 2020_

## Referee Comment (RC1) · Anonymous Referee #1 · 2 Jul 2020

Ochwat and co-authors use firn cores drilled in the accumulation area of Kaskawulsh Glacier, Yukon Territory, Canada, to estimate changes in surface height over the period 2005 to 2018. In the deeper one of the firn core, they could observe a perennial firn aquifer at more than 30 m depth. Based on the cores they state that the surface at the drill site has lowered by ~1.3 m over 2005 to 2018. They emphasize the importance of this result for geodetic glacier mass balance estimates of the glacier and the region.

Ochwat and co-authors make a valuable contribution to the understanding of firn properties in a heavily glacierized region. The area is also considered one of the key regions contributing to global sea level change. The authors address tow main points: (i) the perennial firn aquifer and (ii) surface lowering in the context of geodetic mass balance estimates. Both topics are very interesting scientifically and important in the context

of regional to global glacier change. However, I have the impression that in its current form, the manuscript fails in making sound contributions to either topic.

- The reporting on the discovery of the aquifer is very valuable, but the discussion of the observations remains unspecific, rather general with mostly qualitative comparison to other firn aquifers.

- I believe that the analysis of the firn core, the way it is presented, does not allow retrieval of thinning rates. To do so, additional information is needed, namely evidence of changing density or ice content over time. This evidence is missing, or little used in the argumentation. Consequently, I doubt in the main conclusion of the study.

- Uncertainty analysis generally appears incomplete (see below for details).

**Detailed Remarks**

2.1 Study area: Where is the longer-term average equilibrium line elevation (ELA) on Kaskawulsh glacier? This would be useful to better understand the glaciological situation of the drill site (e.g. 100 m or 1000 m above the ELA?)

Line 111: upper threshold of 917 kg m$^{-3}$: in a firn aquifer, higher densities are physically plausible provided that the water is still in the core segment when weighing. Was this an issue in the context of your study?

Lines 115-116: I do not understand why damage to sample bags affected density measurements, I understand density measurements were carried out in the field, before transporting samples?

Lines 120: What exactly is meant with "human error"?

Lines 145-146: Note that for example Harper et al. (2012) measured a lower density for pure ice (843±36 kg $^{-3}$). Furthermore, you list the wrong study of Machguth et al. (2006 instead of 2016) in the references. Please check references for more errors.

Lines 154-159: Why thinning? I would agree if reference is an ice core without ice

lenses, but it needs to be show that this theoretical reference actually existed at the drill site earlier (2005).

Discussion: The discussion is clearly structure but I perceived the flow of arguments as poor. The text meanders between more general, partly speculative and maybe too qualitative discussion of firn aquifers to the impact on geodetic mass balance estimates. It is not fully clear what the focus of the manuscript is, or what the main message(s) of the manuscript should be.

Lines 265: The statement cited from Christianson et al. (2015) appears incorrect. Already in the 1970s detailed studies of a perennial firn aquifer were carried out in the accumulation area of Abramov glacier (4400 m a.s.l.), Pamir-Alai, present-day Kyrgyzstan. In contrast to other studies, the scientists studied the aquifer in a deep firn pit (up to ∼25 m deep). This allowed continuous monitoring of changes in the water table in relation to, e.g., surface melt intensity. The related studies, however, are mostly published in Russian (Glazirin et al., 1977; Kislov, 1982) and thus not widely known to a broader glaciological audience.

Lines 270-272: Here an estimate of annual accumulation rate is mentioned, based on literature and the authors' own interpretation of the cores. Above (lines 196-198) the authors use a literature value (other sources than here) of 1.76 m w.e. $yr^{-1}$. How do these two numbers relate? Is the implicit assumption made that accumulation rates have remained stable since the 1960s? What is the uncertainty introduced by this assumption?

Line 284 (as well as 184-190): 2 kg $m^{-3}$ appears to be a very low level of overall uncertainty. I assume there must be some potential sources of systematic errors that prevent such a very low uncertainty?

Section 4.3: I think your interpretation stands on weak grounds. There is little evidence presented that accumulation rates from the 1960s are still valid today. As outlined below, the fact that ice lenses exist in the firn does not automatically mean that the

surface lowers. For this to be true, the ice fraction has to change over time. The authors present some evidence of an increase in ice content (lines 288 to 291), but not for the time period represented by the two cores.

Line 322: Surface lowering of 1.3 m: It is confusing to mention this result in cm $yr^{-1}$ in the abstract, not in the results (at least I couldn't find it there) and then again in m $yr^{-1}$ in the discussion.

Lines 322 – 332: I do not understand why there needs to be surface lowering because of the ice lens formation and refreezing? If we knew that there was no or less refrozen water in the firn in 2005, then the surface would have lowered as calculated. However, based on the evidence the authors present, I have the impression we do not know whether ice content has changes 2005-2018. If ice content in the firn would be constant, there would be no surface lowering. Furthermore, the authors make the critical assumption of annual accumulation rate equaling 1.76 m w.e. $yr^{-1}$, leading to the conclusion that the core represents the time period 2005 to 2018 (Lines 196-198). What is the uncertainty of this assumption? Accumulation rates could have changed since the 1960s. Associated uncertainties are neither assessed nor discussed.

Lines 353-355: The fact that an aquifer exist does not mean that the surface has to lower. Evidence is needed that firn properties altered over the time of investigation. If they have not changed (i.e. there was similar ice content earlier, an aquifer existed and accumulation rates remained constant), why should the surface lower?

**References not listed in the manuscript**

Glazyrin G.E., Glazyrina E.L., Kislov B.V. and Pertzinger F.I. (1977) Water level regime in deep firn pits on Abramov glacier [in Russian], volume 45. Gidrometeoizdat

Kislov, B.V. (1982) Formation and regime of the firn-ice stratum of a mountain glacier [in Russian]. Ph.D. thesis, SARNIGMI Tashkent.

---

## Referee Comment (RC2) · Anonymous Referee #2 · 14 Jul 2020

**Review** *"Meltwater Storage in the firn of Kaskawulsh Glacier, Yukon Territory, Canada"* by N. Ochwat.

The authors study the density profile of two firn cores drilled in spring 2018 in the accumulation zone of Kaskawulsh Glacier (Yukon, Canada). These cores are used to calculate local firn density and the impact of meltwater retention and refreezing on surface lowering that must be accounted for to correct geodetic mass balance estimates. The authors obtain an average firn density of $670 \pm 2$ kg m$^{-3}$ in the 36 m deep core, and estimate an average surface lowering of $10 \pm 0.8$ cm per year over the period 2005-2018. The authors also identify a perennial firn aquifer below ~35 m depth.

The paper suffers from major issues including the robustness of the methodology, results and uncertainty estimates, making the conclusions difficult to trust. In addition, some terms used are unclear; the authors sometimes expect a priori knowledge from the readers (e.g. Section 3.3). The reviewer also noted that results reported in the main text and tables are often not matching, and that the conclusions lack of novelty. The paper is mostly descriptive and does not provide novel insight on geodetic mass balance uncertainties compared to previous studies. Therefore, the reviewer deems that the manuscript should be **rejected** in its current form. Below, the authors can find the reviewer's major concerns, listed as General and Point comments.

**General comments**

1. Results are based on "subjective" approximations that may alter the conclusions. For instance, the completeness of the two firn cores section is assessed based on "visual inspection" by three persons. How do the resulting "random" and "human" errors impact the firn density calculated in Eq. 1? In L120, the authors provide a 10-20% uncertainty in estimating the factor f in Eq. 1 (L125-126)? This would lead to a ~100 kg m$^{-3}$ uncertainty in firn density (assuming the 670 kg m$^{-3}$ value reported here), in line with 110 kg m$^{-3}$ estimated in Foy et al. (2011; see L287). However, the authors report uncertainties ranging from 2 to 6 kg m$^{-3}$. Please elaborate. See also Point comment in L137-140.

2. Across the manuscript, the authors report results that are not matching between the main text and tables, making the conclusions hard to trust. For instance, in L18 the authors report an average surface lowering of $10 \pm 0.8$ cm yr$^{-1}$ between 2005-2018. In L356, the authors report $10 \pm 8$ cm yr$^{-1}$ for the same period. In L322, this annual rate is cumulated over the period 2005-2018 to obtain $1.3 \pm 0.8$ m in ~13 years. What uncertainty was used here (0.8 or 8 cm)? Please elaborate. Similar issues can be found across the whole manuscript and are listed in the Point comments.

3. The 13-year period (2005-2008) is estimated using calculated total water content of 23.22 m w.e. at the drilling site and assuming an average accumulation rate of 1.76 m w.e. yr$^{-1}$ (1960s). The authors do not assess the robustness of this estimate given the uncertainty in firn density. Please elaborate.

4. The term "melt-affected firn" is often used in the manuscript but not explained. Is this firn affected by the presence of refrozen meltwater in pore space? What are the associated visual features as stated in L204-205? Perhaps a photo of the cores would help the interpretation. The same holds for "Ice content" in L134, that is sometimes defined as the cumulative thickness of ice layers in the core expressed in m, or as a fraction after being normalized by the length of the firn core (see e.g. L192 and Table 1).

5. The authors sometimes expect "a priori" knowledge from the reader. Section 3.3 on stable isotopes is a good example: how to interpret the summer peaks at -22‰ in Fig. 4? This section is not necessary and the results are not further discussed in the text, except in L244-246 that relates low ion concentrations to active meltwater percolation/motion in firn.

6. The conclusions lack of novelty compared to previous studies that also estimated surface lowering in the region (see L334-339). The paper does not provide a convincing estimate of (local) surface lowering uncertainty for geodetic mass balance measurements, nor estimate

the regional mass change accounting for density correction. In L328-330, the authors claim that density estimated at the two cores are representative of a larger region, which cannot be proved using only two cores as stated in L371-376. The authors should consider combining their core measurements with firn modeling to obtain spatially continuous density profiles and estimate regional mass balance uncertainty due to firn processes.

**Point comments**

L92-94: Are the measurements from the snow pit discussed somewhere in the manuscript or shown in Fig. 2? Please clarify.

L135-136: What does "melt percent" mean? How is this calculated?

L137-140: This is unclear, why should the thickness of ice lenses be divided by a factor two?

L161-164: The authors should provide some references on the methods used to study isotopes.

L184-190: This paragraph includes numerous errors in reporting results. In L186, "571 ± 3 kg m$^{-3}$" is reported in the text while Table 1 lists 518 kg m$^{-3}$ at core 2 between 4-14 m depth. In L187, "608 ± 2 kg m$^{-3}$" is reported while Table 1 lists 618 kg m$^{-3}$ between 4-21 m depth. The authors report an extremely small density uncertainty of 2-3 kg m$^{-3}$ while Figs. 2a and b show much larger uncertainties. In L229-230, the authors state that densities larger than 917 kg m$^{-3}$ are eliminated. However, Fig. 2a shows values of ~1000 kg m$^{-3}$ or larger at 6 and 10 m depth. To the reviewer, it is hard to judge whether these errors are due to negligence or calculation errors. Please elaborate.

L185, 187, 188: For clarity, the authors should better write: "between 4 and 14 m depth" instead of "in the upper 10 m"; "between 4-21 m depth" instead of "in the upper 17 m"; and "between 4-36 m depth" instead of "representing ~32 m". The same holds for L284-286.

L193: 660 kg m$^{-3}$ is actually 1.5% smaller than the firn density of 670 kg m$^{-3}$ reported in L189.

L276: What do the authors mean by "summer melt extent"? Do they mean meltwater production in mm w.e. yr$^{-1}$ as listed in Table 2? Please clarify.

L278: It is hard to assess the robustness of the results in this paragraph. In L278, the authors state that summer 2015 was the warmest in the period 2014-2018, whereas Table 2 shows that it was actually summer 2016 (-1.0ºC in 2016 vs. -1.8ºC in 2015). The same goes for annual mean temperature in 2015-2016 (-9.0ºC in 2016 vs. -9.6ºC in 2015). How to interpret the larger PDD and melt rates in 2015 then? Please clarify.

L284: Again 608 kg m$^{-3}$ is reported in the text whereas 618 kg m$^{-3}$ is listed in Table 1.

L315: What do the authors mean by "certain amount"? Ice layer thickness?

Table 3: What does "1.5-2g" mean in the personal communication of Sass and O'Neel?

---

## Author Comment (AC1) · 14 Aug 2020

**Ochwat and co-authors use firn cores drilled in the accumulation area of Kaskawulsh Glacier, Yukon Territory, Canada, to estimate changes in surface height over the period 2005 to 2018. In the deeper one of the firn core, they could observe a perennial firn aquifer at more than 30 m depth. Based on the cores they state that the surface at the drill site has lowered by ~1.3 m over 2005 to 2018. They emphasize the importance of this result for geodetic glacier mass balance estimates of the glacier and the region. Ochwat and co-authors make a valuable contribution to the understanding of firn properties in a heavily glacierized region. The area is also considered one of the key regions contributing to global sea level change. The authors address tow main points: (i) the perennial firn aquifer and (ii) surface lowering in the context of geodetic mass balance estimates. Both topics are very interesting scientifically and important in the context of regional to global glacier change. However, I have the impression that in its current form, the manuscript fails in making sound contributions to either topic.**

**The reporting on the discovery of the aquifer is very valuable, but the discussion of the observations remains unspecific, rather general with mostly qualitative comparison to other firn aquifers.**

We will revise the discussion to be more specific. The intention of the project was to study the densification of the firn in the region. When we drilled to 32 m we discovered the firn aquifer and realized that it had not been previously reported. We do not know much about the aquifer, but its presence is unequivocal – liquid water dripping from the firn core samples from 32 to 34 m depth – and it is a significant discovery that will be of interest to the *Cryosphere* audience. To our knowledge, this is the first reported firn aquifer in the St. Elias Mountains and the Alaska-Yukon region. We have interesting video footage of the liquid water discharge from the core samples that could be added as a link to the online manuscript. We have also been in touch with Dr. Karl Kreutz and Dr. Seth Campbell of the University of Maine, who have recent unpublished radar data from a site 12 km away that provides evidence for the firn aquifer, across a larger area than we originally reported. We will incorporate their observations into the updated manuscript, together with a discussion of the climatology of the region, which is consistent with the formation of firn aquifers found elsewhere (high accumulation rates, cold winters, moderate summer melt).

**I believe that the analysis of the firn core, the way it is presented, does not allow retrieval of thinning rates. To do so, additional information is needed, namely evidence of changing density or ice content over time. This evidence is missing, or little used in the argumentation. Consequently, I doubt in the main conclusion of the study.**

We will strengthen our explanation of how we retrieved thinning rates and discuss the evidence when arguing for the main conclusion. To determine how densities have changed over time we will make direct comparisons with density data recorded in a 15 m

core taken in 1964 (Grew and Mellor, 1966) in a location close to our field site. Preliminary analysis indicates that the firn was less dense then, and this information will be combined with other unpublished historical data from the region (available from colleagues and historical reports) to quantify how the densification rate has changed over time.

**Uncertainty analysis generally appears incomplete (see below for details).**

We will explain the uncertainty analysis in greater detail and offer more details in the specific responses below. We did complete careful uncertainty analyses for the firn density itself, and there was misunderstanding concerning the uncertainty in point samples vs. average densities, as averaging greatly reduces the uncertainty for random errors, by a factor of $\sqrt{N}$. We will revise the text to be clearer about this. We agree with the reviewers, however, that we did not adequately address the uncertainty in the estimate of average accumulation (hence age) of the core, so we will add this uncertainty analysis to the manuscript.

**Detailed Remarks**

**2.1 Study area: Where is the longer-term average equilibrium line elevation (ELA) on Kaskawulsh glacier? This would be useful to better understand the glaciological situation of the drill site (e.g. 100 m or 1000 m above the ELA?)**

The ELA will be added to this section, as reported in Foy et al. (2011) and Young et al., (2020). In Foy et al. (2011), an ELA of 1958 m a.s.l. is used throughout their study as a long-term average determined by satellite imagery. It will also be noted that the ELA has shifted upwards since that report by almost 300 m. Young et al. (2020) state a mean of 2261 ± 151 m a.s.l. for the years 2013-2019. Our core site is nonetheless well above the ELA at an elevation of ~2640 m a.s.l., within a broad plateau in the main accumulation area of the glacier.

**Line 111: upper threshold of 917 kg m$^{-3}$: in a firn aquifer, higher densities are physically plausible provided that the water is still in the core segment when weighing. Was this an issue in the context of your study?**

The cores drilled in the firn aquifer were drained before being bagged and there was no noticeable liquid water in the sample bags when weighed. That said, this is a good point and our samples could have contained residual liquid water; some may have refrozen due to sub-zero surface temperatures during the drilling. We will note this in this section, and acknowledge in the manuscript that the three outliers from 32-36m depth may have had liquid water in them, thus causing a higher density.

**Lines 115-116: I do not understand why damage to sample bags affected density measurements, I understand density measurements were carried out in the field, before transporting samples?**

Density measurements were carried out in the field for most of the measurements. However, due to the need to leave the glacier at the end of drilling, a few remaining measurements were carried out at the research station within 24 hours of being back. Some of the sample bags were damaged in this transport and were not included in the measurements at the research station. We will clarify this in the text.

**Lines 120: What exactly is meant with "human error"?**

This term will be rephrased to random error. We meant the error associated with core measurements (length, diameter) and the subjective assessment of core completeness. These are all considered to be random rather than systematic sources of error.

**Lines 145-146: Note that for example Harper et al. (2012) measured a lower density for pure ice (843±36 kg/m$^3$). Furthermore, you list the wrong study of Machguth et al. (2006 instead of 2016) in the references. Please check references for more errors.**

Thank you for noticing the typo in the year of the citation. This will be fixed and other references have been double-checked.

We noticed that Harper et al. (2012) determined a lower density. Based on four core sections that had 100% ice in our study, we measured pure ice to have an average density of 907 ± 14 kg/m$^3$ in our data. This is above the reported values of Bezeau et al. (2013) and Machguth et al. (2016), which are in turn higher than the value of Harper et al. (2012). We chose to go with the middle value. In the revised manuscript we will assign an uncertainty of 35 kg/m$^3$ in order to accommodate both ends of the possible spectrum of density for the pure ice sections (i.e., inclusive of both Harper's work and our own data).

**Lines 154-159: Why thinning? I would agree if reference is an ice core without ice lenses, but it needs to be show that this theoretical reference actually existed at the drill site earlier (2005).**

The thinning discussed in the original manuscript is due to annual densification of the accumulation from meltwater percolation and refreezing: a thinning that is not associated with mass loss. The reviewer is therefore correct that it did not indicate densification changes over time. To address changes in densification over time, we will incorporate historical data into the revised manuscript from the original Icefield Ranges Research Reports (as mentioned above), particularly from the 1960s density profiles of Grew and Mellor (1966).

**Discussion: The discussion is clearly structure but I perceived the flow of arguments as poor. The text meanders between more general, partly speculative and maybe too qualitative discussion of firn aquifers to the impact on geodetic mass balance estimates. It is not fully clear what the focus of the manuscript is, or what the main message(s) of the manuscript should be.**

We apologize for this lack of clarity. The main objective of the original manuscript was

to characterize the firn of the upper Kaskawulsh Glacier: a significant ice mass within a major icefield where little or no published data is currently available on firn density or densification rates, meltwater retention, or liquid meltwater storage. The revised paper will be expanded to clarify and reiterate the three main messages: 1) firn density and ice content, 2) changes in densification rate, and 3) the new firn aquifer in this region. The results and discussion will focus around these three points. Number (3) is admittedly a bit of an aside, but it is of great interest and is relevant to meltwater retention and mass balance studies, as well as affecting the glacier thermal and hydrological behavior.

**Lines 265: The statement cited from Christianson et al. (2015) appears incorrect. Already in the 1970s detailed studies of a perennial firn aquifer were carried out in the accumulation area of Abramov glacier (4400 m a.s.l.), Pamir-Alai, present-day Kyrgyzstan. In contrast to other studies, the scientists studied the aquifer in a deep firn pit (up to ~25 m deep). This allowed continuous monitoring of changes in the water table in relation to, e.g., surface melt intensity. The related studies, however, are mostly published in Russian (Glazirin et al., 1977; Kislov, 1982) and thus not widely known to a broader glaciological audience.**

We would like to include these Russian papers for reference in the manuscript. However, despite extensive searching we have not been able to find them and hope that the reviewer can forward them to us so that we can include the data mentioned in the above comment.

**Lines 270-272: Here an estimate of annual accumulation rate is mentioned, based on literature and the authors' own interpretation of the cores. Above (lines 196-198) the authors use a literature value (other sources than here) of 1.76 m w.e. yr⁻¹. How do these two numbers relate? Is the implicit assumption made that accumulation rates have remained stable since the 1960s? What is the uncertainty introduced by this assumption?**

We have obtained a new dataset from 2003-2013 of snow accumulation and density data from the Divide site (12 km from our drill site, similar elevation) on the upper Kaskawulsh Glacier. We will include this data in our estimate of annual accumulation rates in order to provide more supporting evidence for the accumulation rate chosen here. The interannual variability in this data could be used as an estimate of uncertainty. We have three additional lines of evidence for the annual accumulation rate: (i) preserved peaks in the oxygen isotope record, (ii) our own winter accumulation measurements from spring 2018, and (iii) the (much) earlier published data from the IRRP. This will all be taken into account to assess a conservative uncertainty in the annual accumulation rate, which can then be propagated through to the uncertainty in the age of the core.

**Line 284 (as well as 184-190): 2 kg m⁻³ appears to be a very low level of overall uncertainty. I assume there must be some potential sources of systematic errors that prevent such a very low uncertainty?**

We will double check our calculations, but believe this is just confusion regarding the

uncertainty in point samples vs. average values for the core. We use standard error analysis in these calculations, which we are happy to walk through in supplementary material if the reviewers would like to see it. To summarize here, our point samples (10-cm core sections) have significant measurement uncertainty, e.g., $500 \pm 75$ kg m$^{-3}$. Sources of uncertainty are random rather than systematic, to our knowledge. Figuratively speaking, for the case of random errors, averaging of the 10-cm density values for the whole core effectively leads to reductions in uncertainty because random errors cancel out. Based on standard error analysis, one can take the example of a 30-m core with 300 10-cm density values ($N = 300$).  The standard error in the average follows $s_e = \quad /\sqrt{N}$, where in this illustration we can take $\quad = 75$ kg m$^{-3}$.  This gives $s_e = 4$ kg m$^{-3}$. We will nonetheless go through our uncertainty calculations again to ensure that these are accurate.

**Section 4.3: I think your interpretation stands on weak grounds. There is little evidence presented that accumulation rates from the 1960s are still valid today. As outlined below, the fact that ice lenses exist in the firn does not automatically mean that the surface lowers. For this to be true, the ice fraction has to change over time. The authors present some evidence of an increase in ice content (lines 288 to 291), but not for the time period represented by the two cores.**

We will strengthen our interpretations by bringing in more of the recent measurements reported by Foy and others, as well as those made by ourselves in this region over the past ~15 years, to quantify whether accumulation rates have changed over time. The point we were making is that meltwater refreezing increases the density of the firn, thinning the annual accumulation layer without an associated mass loss. This also makes "dry firn" models inappropriate for density estimates that are needed for geodetic mass balance measurements, so even the basic reporting of firn density values and firn ice content are of value. We do understand that firn densification associated with multi-year changes in ice content is also of great interest, so we will include a new analysis of density changes over time in the revised manuscript.

**Line 322: Surface lowering of 1.3 m: It is confusing to mention this result in cm yr$^{-1}$ in the abstract, not in the results (at least I couldn't find it there) and then again in m yr$^{-1}$ in the discussion.**

We will revise the manuscript such that the units are consistent throughout the text, thank you for pointing out this inconsistency. Additionally, we will elaborate on how the 1.3 m surface lowering has been calculated, as well as the uncertainty associated with the value.

**Lines 322 – 332: I do not understand why there needs to be surface lowering because of the ice lens formation and refreezing? If we knew that there was no or less refrozen water in the firn in 2005, then the surface would have lowered as calculated. However, based on the evidence the authors present, I have the impression we do not know whether ice content has changes 2005-2018. If ice content in the firn would be constant, there would be no surface lowering.**

We understand the reviewer's point, and it makes it clear that we did not explain our calculations and objective clearly. In the original manuscript we reported the ice content present in the firn during the period of our study, which we used to determine the extent of surface lowering related to the meltwater percolation and refreezing affects within the firn. This process of densification creates an effective thinning, but not the multi-year changes in this process (i.e. classical firn densification). In the revised manuscript we will address this by using historical density information from the Icefield Ranges Research Reports to provide evidence as to how the firn density and ice content has changed over time.

**Furthermore, the authors make the critical assumption of annual accumulation rate equaling 1.76 m w.e. yr$^{-1}$, leading to the conclusion that the core represents the time period 2005 to 2018 (Lines 196-198). What is the uncertainty of this assumption? Accumulation rates could have changed since the 1960s. Associated uncertainties are neither assessed nor discussed.**

This point is addressed above. We agree that uncertainties need to be assigned and discussed, including error propagation through to the calculation of the age of the core.

**Lines 353-355: The fact that an aquifer exist does not mean that the surface has to lower. Evidence is needed that firn properties altered over the time of investigation. If they have not changed (i.e. there was similar ice content earlier, an aquifer existed and accumulation rates remained constant), why should the surface lower?**

The presence of any kind of liquid water would be a form of firn densification, but in the sense that we were referring to (potentially confusing to readers): surface melting causes surface lowering, and where this meltwater is retained as liquid water or refrozen ice there is a measurable surface lowering that is not accompanied by mass loss. We have undertaken a comprehensive literature search in response to this comment, and found no evidence that the firn aquifer existed in the past. The presence of the new firn aquifer therefore makes it likely that the surface has lowered due to the firn aquifer's presence in the recent past, and we will provide estimates of how much this lowering has been. We will report this in the updated manuscript, and provide supporting information.

**References not listed in the manuscript**
**Glazyrin G.E., Glazyrina E.L., Kislov B.V. and Pertzinger F.I. (1977) Water level regime in deep firn pits on Abramov glacier [in Russian], volume 45. Gidrometeoizdat**

**Kislov, B.V. (1982) Formation and regime of the firn-ice stratum of a mountain glacier [in Russian]. Ph.D. thesis, SARNIGMI Tashkent.**

Please see the previous comments regarding the addition of these references into the manuscript.

---

## Author Comment (AC2) · 14 Aug 2020

**Review "Meltwater Storage in the firn of Kaskawulsh Glacier, Yukon Territory, Canada" by N. Ochwat.**

**The authors study the density profile of two firn cores drilled in spring 2018 in the accumulation zone of Kaskawulsh Glacier (Yukon, Canada). These cores are used to calculate local firn density and the impact of meltwater retention and refreezing on surface lowering that must be accounted for to correct geodetic mass balance estimates. The authors obtain an average firn density of $670 \pm 2$ kg m$^{-3}$ in the 36 m deep core, and estimate an average surface lowering of $10 \pm 0.8$ cm per year over the period 2005-2018. The authors also identify a perennial firn aquifer below ~35 m depth. The paper suffers from major issues including the robustness of the methodology, results and uncertainty estimates, making the conclusions difficult to trust. In addition, some terms used are unclear; the authors sometimes expect a priori knowledge from the readers (e.g. Section 3.3). The reviewer also noted that results reported in the main text and tables are often not matching, and that the conclusions lack of novelty. The paper is mostly descriptive and does not provide novel insight on geodetic mass balance uncertainties compared to previous studies. Therefore, the reviewer deems that the manuscript should be rejected in its current form. Below, the authors can find the reviewer's major concerns, listed as General and Point comments.**

There is quite a bit to unpack here and on some points we provide more detail with the specific responses below.

We will revise the manuscript to clarify the aims, methodology, results, and uncertainties. We apologize for confusion in the values reported – we were inconsistent and unclear in some places, and this will be fixed. We will also provide more supporting evidence indicating how the density of the firn has changed over time, using a range of studies completed since the 1960s, primarily from the Icefield Ranges Research Reports, as well as from previously unpublished field data collected by colleagues. We will revise Section 3.3 in order to provide more a priori knowledge on the stable isotopes, and how these help to inform our inference of annual accumulation rates. We will also be clearer in the uncertainty analysis and in the discussion of the various density values being reported in order to make the relationship between the table and the text more clear.

The main objective of the original manuscript was to characterize the firn of the upper Kaskawulsh Glacier: a significant ice mass within a major icefield where little or no published data is currently available on firn density or densification rates, meltwater retention, or liquid meltwater storage. The revised paper will be expanded and restructured. We will reiterate and clarify the three main messages: 1) firn density and ice content, 2) changes in densification rate, and 3) the new firn aquifer in this region. The results and discussion will focus around these three points. Number (3) is admittedly a bit of an aside, but it is of great interest and is relevant to meltwater retention and mass balance studies, as well as affecting the glacier thermal and hydrological behavior.

**General comments**
**1. Results are based on "subjective" approximations that may alter the conclusions. For instance, the completeness of the two firn cores section is assessed based on "visual inspection" by three persons. How do the resulting "random" and "human" errors impact the firn density calculated in Eq. 1? In L120, the authors provide a 10-20% uncertainty in estimating the factor f in Eq. 1 (L125-126)? This would lead to a ~100 kg m$^{-3}$ uncertainty in firn density (assuming the 670 kg m$^{-3}$ value reported here), in line with 110 kg m$^{-3}$ estimated in Foy et al. (2011; see L287). However, the authors report uncertainties ranging from 2 to 6 kg m$^{-3}$. Please elaborate. See also Point comment in L137-140.**

We will double check our calculations, but believe this is just confusion regarding the uncertainty in point samples vs. average values for the core. We use standard error analysis in these calculations, which we are happy to walk through in supplementary material if the reviewers would like to see it. To summarize here, our point samples (10-cm core sections) have significant measurement uncertainty, as the reviewer notes. There are numerous sources of measurements error, including the subjective assessment of the completeness of each core section. We will refer to these various sources of uncertainty as "random error" in the revised manuscript, as they are all believed to be random (vs. systematic) – sometimes we will measure or estimate a section as being too long or complete, sometimes we will underestimate it. The uncertainty factor of ~15% applies to point samples (10-cm core sections): say, for instance, $500 \pm 75$ kg m$^{-3}$ for a given sample. Conceptually, for the case of random errors, averaging of the 10-cm density values for the whole core leads to marked reductions in uncertainty because random errors cancel out. Based on standard error analysis, the standard error in the average follows $s_e = \ /\sqrt{N}$, where is the uncertainty in each data point and $N$ is the number of samples. Taking the example of a 30-m core with 300 10-cm density values, $N = 300$. Taking $= 75$ kg m$^{-3}$ as an example, $s_e = 4$ kg m$^{-3}$. We will go through our uncertainty calculations again to ensure that these are accurate, but the main point here is the difference between sample uncertainties and the standard error in the mean.

**2. Across the manuscript, the authors report results that are not matching between the main text and tables, making the conclusions hard to trust. For instance, in L18 the authors report an average surface lowering of $10 \pm 0.8$ cm yr$^{-1}$ between 2005-2018. In L356, the authors report $10 \pm 8$ cm yr$^{-1}$ for the same period. In L322, this annual rate is cumulated over the period 2005-2018 to obtain $1.3 \pm 0.8$ m in ~13 years. What uncertainty was used here (0.8 or 8 cm)? Please elaborate. Similar issues can be found across the whole manuscript and are listed in the Point comments.**

We will correct the typos noted in the text, thank you for pointing those out. We also now realize that reporting so many different densities for the calculated background firn and actual firn may have led to some confusion. We report the upper 10 m of firn density to allow comparability to other literature on firn density. We also chose to compare the partial core densities because the length of Core 1 and Core 2 differed – this permits a

direct comparison. We will remove some of these values in order to make the relationship between the values in the table and text clearer, and will be more precise and consistent with the wording we use in the revised manuscript.

**3. The 13-year period (2005-2008) is estimated using calculated total water content of 23.22 m w.e. at the drilling site and assuming an average accumulation rate of 1.76 m w.e. yr$^{-1}$ (1960s). The authors do not assess the robustness of this estimate given the uncertainty in firn density. Please elaborate.**

We agree with the reviewer that we did not adequately address the uncertainty in the estimate of average accumulation (hence age) of the core. We will add this uncertainty analysis to the manuscript. We have obtained a dataset from 2003-2013 of snow accumulation and density data from the Divide site (12 km from our drill site, similar elevation) on the upper Kaskawulsh Glacier. We will include this data in our estimate of annual accumulation rates in order to provide more supporting evidence for the accumulation rate chosen here. The interannual variability in this data could be used as an estimate of uncertainty. We have three additional lines of evidence for the annual accumulation rate: (i) preserved peaks in the oxygen isotope record, (ii) our own winter accumulation measurements from spring 2016, and (iii) the (much) earlier published data from the Icefield Ranges Research Reports. These will all be taken into account to provide an uncertainty in the annual accumulation rate, which can then be propagated through to the uncertainty in the age of the core.

**4. The term "melt-affected firn" is often used in the manuscript but not explained. Is this firn affected by the presence of refrozen meltwater in pore space? What are the associated visual features as stated in L204-205? Perhaps a photo of the cores would help the interpretation. The same holds for "Ice content" in L134, that is sometimes defined as the cumulative thickness of ice layers in the core expressed in m, or as a fraction after being normalized by the length of the firn core (see e.g. L192 and Table 1).**

A definition of "melt-affected firn" will be added to increase clarity in the methods section. "Melt-affected firn" is any firn that displays physical characteristics indicating that there was the presence of liquid water at some point. This can result in ice layers, ice lenses, or can be indicated by the lack of grain boundaries, the presence of air bubbles, texture, and opacity. "Melt-affected firn" can also be identified using stable isotopes and the cation/anions, however, this was not done in the field.

The use of the term "ice content" will be used with more precise wording in the text in order to clarify as to whether or not it is describing cumulative ice layers or the normalized fraction. We will refer to the former as the "total ice content" and the latter as "ice fraction".

**5. The authors sometimes expect "a priori" knowledge from the reader. Section 3.3 on stable isotopes is a good example: how to interpret the summer peaks at -22‰ in Fig. 4? This section is not necessary and the results are not further discussed in the**

**text, except in L244-246 that relates low ion concentrations to active meltwater percolation/motion in firn.**

We agree that more information needs to be included in section 3.3. The following will be added into the manuscript in the paragraph that contains L244-246:
"$\delta^{18}$O records in ice cores are a proxy for paleo-temperature, and are thus often utilized to assist in deriving age scales for ice cores. This method relies on the strong high-latitude temperature modulation of the isotopic composition of precipitation [*Jouzel et al.*, 1997; *Schneider et al.*, 2005]"

Although we realize the amount of melt and percolation rendered the major ion data useless, and the stable isotope data largely washed out, we wish to keep it within the manuscript as these results support our other findings of active meltwater percolation. Additionally, we discuss the sections of the stable isotope record that are preserved, and thus useful in determining an age-depth relationship.

**6. The conclusions lack of novelty compared to previous studies that also estimated surface lowering in the region (see L334-339). The paper does not provide a convincing estimate of (local) surface lowering uncertainty for geodetic mass balance measurements, nor estimate the regional mass change accounting for density correction. In L328-330, the authors claim that density estimated at the two cores are representative of a larger region, which cannot be proved using only two cores as stated in L371-376. The authors should consider combining their core measurements with firn modeling to obtain spatially continuous density profiles and estimate regional mass balance uncertainty due to firn processes.**

There are several good points here, mostly related to points addressed above. Within our objective to characterize the firn density, densification rates, and meltwater retention on the upper Kaskawulsh Glacier, we did not initially set out to quantify changes in densification rates over time – only the annual densification associated with meltwater refreezing, and the resulting firn density profile. We understand that we need to clarify and expand on this to study to include changes in the densification rate as well, as to be more relevant to geodetic mass balance studies. As discussed above, we have reached out to colleagues that have worked in this region for many decades, including previous ice-core work (C. Zdanowicz, K. Kreutz, S. Campbell), and will use this unpublished data, as well as other published data, to quantify how the firn ice content and densification rate has changed over time. We can also develop models of this process, using climate reanalyses as forcing data.

It is difficult to know whether our two cores are representative of the larger accumulation area of this icefield or others in the St. Elias region, but they still provide information where little other recent work on firn properties has been undertaken. We will revise our discussion and not over-extend our claims. Firn modelling is difficult when there is a large amount of meltwater percolation and refreezing – firn densification models do not do well under this condition – which is part of the need for the kind of data that we present.

**Point comments L92-94: Are the measurements from the snow pit discussed somewhere in the manuscript or shown in Fig. 2? Please clarify.**

The snowpit measurements are presented in Figure 2 and are included in the density data and inference about annual snow accumulation. We will edit the caption of the figure in order to clarify that the measurements are part of the density figure.

**L135-136: What does "melt percent" mean? How is this calculated?**

We will explain this more clearly in the manuscript. "Melt percent" has been used in the literature (e.g., Koerner, 1977) to refer to the percent of annual snow accumulation that melts (and refreezes), in the accumulation area of polar environments. At our site we don't use this concept but use "ice content" to refer to the fraction of a core sample that is made up of refrozen meltwater. For clarity, we will refer to this as "ice fraction" in revisions.

**L137-140: This is unclear, why should the thickness of ice lenses be divided by a factor two?**

Ice lenses were partial ice layers, where the ice did not extend horizontally through the whole core section. We assume that, on average, the ice lens occupied 50% of the core; therefore the measured thickness was divided by two. We will clarify this in the text.

**L161-164: The authors should provide some references on the methods used to study isotopes.**

Please kindly refer to our response to comment #5 above.

**L184-190: This paragraph includes numerous errors in reporting results. In L186, "571 ± 3 kg m$^{-3}$" is reported in the text while Table 1 lists 518 kg m$^{-3}$ at core 2 between 4-14 m depth. In L187, "608 ± 2 kg m$^{-3}$" is reported while Table 1 lists 618 kg m$^{-3}$ between 4-21 m depth. The authors report an extremely small density uncertainty of 2-3 kg m$^{-3}$ while Figs. 2a and b show much larger uncertainties. In L229- 230, the authors state that densities larger than 917 kg m$^{-3}$ are eliminated. However, Fig. 2a shows values of ~1000 kg m$^{-3}$ or larger at 6 and 10 m depth. To the reviewer, it is hard to judge whether these errors are due to negligence or calculation errors. Please elaborate.**

The uncertainties of point samples versus average values for the cores are discussed above (please see the response to point #1). Figure 2 shows the point data (10-cm sections). The outliers were removed from the calculations to determine density and background firn density, but we left the outliers in the figure in order to allow the readers to see all of the data used and because the outliers were still within the uncertainty associated with each point. We can eliminate the outliers from the graphs in the revised

manuscript if this is a source of confusion. These are part of the uncertainty in the measurements, but of course these values are not possible so they can be set to the maximum density.

**L185, 187, 188: For clarity, the authors should better write: "between 4 and 14 m depth" instead of "in the upper 10 m"; "between 4-21 m depth" instead of "in the upper 17 m"; and "between 4-36 m depth" instead of "representing ~32 m". The same holds for L284-286.**

Thank you for the suggestion, we will edit the manuscript accordingly. We were attempting to be brief, but it is more important to be clear and therefore use longer wording.

**L193: 660 kg m$^{-3}$ is actually 1.5% smaller than the firn density of 670 kg m$^{-3}$ reported in L189.**

Thank you for noticing this error, we have recalculated and will make sure that the correct percentage is used in the revised paper.

**L276: What do the authors mean by "summer melt extent"? Do they mean meltwater production in mm w.e. yr$^{-1}$ as listed in Table 2? Please clarify.**

This sentence will be reworded so that instead of saying "summer melt extent", it will be clarified as meltwater production in mm w.e./yr, as listed in Table 2.

**L278: It is hard to assess the robustness of the results in this paragraph. In L278, the authors state that summer 2015 was the warmest in the period 2014-2018, whereas Table 2 shows that it was actually summer 2016 (-1.0ºC in 2016 vs. -1.8ºC in 2015). The same goes for annual mean temperature in 2015-2016 (-9.0ºC in 2016 vs. -9.6ºC in 2015). How to interpret the larger PDD and melt rates in 2015 then? Please clarify.**

Thank you for pointing out the unclear text. We will rewrite this. It is that PDD are not the same as average temperatures, and they don't always correlate. PDD refer to the cumulative temperature above 0°C, which can deviate from the average temperature. For instance, days or overnights of −10°C will bring down the average temperature vs. −1°C, but 0 PDD accumulate in each case. Melt is assumed to scale with PDD rather than average temperature, as a proxy for available melt energy. We will revise to say that summer 2015 likely experienced the most melt in recent years, based on higher PDD totals.

**L284: Again 608 kg m$^{-3}$ is reported in the text whereas 618 kg m$^{-3}$ is listed in Table 1.**

Thank you for pointing out this error, it will be fixed.

**L315: What do the authors mean by "certain amount"? Ice layer thickness?**

"Certain amount" will be removed, as it is ambiguous. This will be reworded to include ice layer thickness, spatial extent, and pore space availability, and the reference Machguth et al., (2016) will be added.

**Table 3: What does "1.5-2g" mean in the personal communication of Sass and O'Neel?**

Thank you for noticing the "g" that should not have been there and will be removed.

---

## Author Response (AR1)

Dear Dr. Etienne Berthier,
Thank you for the extension and allowing us the time to implement significant changes into the manuscript. Due to the extent of the changes made we do not have a document that has the "tracked-changes" on it or a point by point description of the changes. We have added several new sections to the manuscript, a surface energy balance model coupled with a firn model, historical data from 1964, data from 2006, and revised the introduction, methods, discussion, and conclusions accordingly. We removed the section on stable isotopes and revised or removed some figures and tables. We have also changed the title to "Evolution of the firn pack of Kaskawulsh Glacier, Yukon: meltwater effects, densification, warming, and the creation of a perennial firn aquifer" to better match the new manuscript. We believe these changes will make the manuscript much stronger.
Thank you,
Naomi Ochwat, on behalf of the coauthors.

Below you will find some description of the edits the reviewers requested and how we responded to them.

**Anonymous Referee #1**

**Ochwat and co-authors use firn cores drilled in the accumulation area of Kaskawulsh Glacier, Yukon Territory, Canada, to estimate changes in surface height over the period 2005 to 2018. In the deeper one of the firn core, they could observe a perennial firn aquifer at more than 30 m depth. Based on the cores they state that the surface at the drill site has lowered by ~1.3 m over 2005 to 2018. They emphasize the importance of this result for geodetic glacier mass balance estimates of the glacier and the region. Ochwat and co-authors make a valuable contribution to the understanding of firn properties in a heavily glacierized region. The area is also considered one of the key regions contributing to global sea level change. The authors address tow main points: (i) the perennial firn aquifer and (ii) surface lowering in the context of geodetic mass balance estimates. Both topics are very interesting scientifically and important in the context of regional to global glacier change. However, I have the impression that in its current form, the manuscript fails in making sound contributions to either topic.**

**The reporting on the discovery of the aquifer is very valuable, but the discussion of the observations remains unspecific, rather general with mostly qualitative comparison to other firn aquifers.**

We have revised the discussion to be more specific. Please review the new section on the surface energy balance and firn modeling (section 3.4 and 4.4). We have also extended the discussion to include this new information. We have also incorporated historical firn temperature data in order to better articulate the thermodynamic processes that have enabled the formation of the firn aquifer. We have also included an ERA climate reanalysis of the region to enhance this discussion (section 3.4 and 4.4).

**I believe that the analysis of the firn core, the way it is presented, does not allow retrieval of thinning rates. To do so, additional information is needed, namely evidence of changing density or ice content over time. This evidence is missing, or little used in the argumentation. Consequently, I doubt in the main conclusion of the study.**

We have strengthened our explanation of how we retrieved thinning rates and discussed the evidence when arguing for the main conclusion. To determine how densities have changed over time we have made direct comparisons with density data recorded in a 15 m core taken in 1964 (Grew and Mellor, 1966) in a location close to our field site as well as data from a 2006 field campaign at the same site. Please refer to section 3.3, 4.3, and 5.1 for this analysis and discussion.

**Uncertainty analysis generally appears incomplete (see below for details).**

We have explained the uncertainty analysis in greater detail and offer more details in the specific responses below. We did complete careful uncertainty analyses for the firn density itself, and there was misunderstanding concerning the uncertainty in point samples vs. average densities, as averaging greatly reduces the uncertainty for random errors, by a factor of $\sqrt{N}$. We have revised the text to be clearer about this. We agree with the reviewers, however, that we did not adequately address the uncertainty in the estimate of average accumulation (hence age) of the core, so this has been added to the uncertainty analysis in the manuscript. Please refer to revised sections 3.1 and 3.2.

**Detailed Remarks**

**2.1 Study area: Where is the longer-term average equilibrium line elevation (ELA) on Kaskawulsh glacier? This would be useful to better understand the glaciological situation of the drill site (e.g. 100 m or 1000 m above the ELA?)**

The ELA has been be added to section 2, as reported in Foy et al. (2011) and Young et al., (2020). In Foy et al. (2011), an ELA of 1958 m a.s.l. is used throughout their study as a long-term average determined by satellite imagery. It will also be noted that the ELA has shifted upwards since that report by almost 300 m. Young et al. (2020) state a mean of 2261 ± 151 m a.s.l. for the years 2013-2019. Our core site is nonetheless well above the ELA at an elevation of ~2640 m a.s.l., within a broad plateau in the main accumulation area of the glacier. [Lines 72-79]

**Line 111: upper threshold of 917 kg m$^{-3}$: in a firn aquifer, higher densities are physically plausible provided that the water is still in the core segment when weighing. Was this an issue in the context of your study?**

We have addressed this by adding the sentence: "Outliers were removed for the background firn density calculations if they were not physically possible (e.g., values

>917 kg/m³ or <300 kg/m³ at depths greater than 4 m). The three outliers from 32-36 m depth may have had residual liquid water in them, thus causing a higher density." [Lines 115 -117]

**Lines 115-116: I do not understand why damage to sample bags affected density measurements, I understand density measurements were carried out in the field, before transporting samples?**

To address this we have added "The Core 2 samples could not be measured for density in the field due to lack of time, so were flown to Kluane Lake Research Station frozen, where the measurements were made within 24 hours of arrival. A random assortment of 125 out of the 196 Core 2 sample bags were damaged during this transport, so were not included in the measurements. This left 71 samples available to use for the density analysis, with at least one sample available per meter except for between 13.29 and 14.95 m. Due to these missing values, only bulk density values are presented for Core 2." [Lines 119-124]

**Lines 120: What exactly is meant with "human error"?**

We have rephrased this to random error. We meant the error associated with core measurements (length, diameter) and the subjective assessment of core completeness. These are all considered to be random rather than systematic sources of error.  [Lines 126-171]

**Lines 145-146: Note that for example Harper et al. (2012) measured a lower density for pure ice (843±36 kg/m³). Furthermore, you list the wrong study of Machguth et al. (2006 instead of 2016) in the references. Please check references for more errors.**

Thank you for noticing the typo in the year of the citation. This was fixed and other references have been double-checked.

We noticed that Harper et al. (2012) determined a lower density. Based on four core sections that had 100% ice in our study, we measured pure ice to have an average density of $907 \pm 14$ kg/m³ in our data. This is above the reported values of Bezeau et al. (2013) and Machguth et al. (2016), which are in turn higher than the value of Harper et al. (2012). We chose to go with the middle value. In the revised manuscript we will assign an uncertainty of 35 kg/m³ in order to accommodate both ends of the possible spectrum of density for the pure ice sections (i.e., inclusive of both Harper's work and our own data).

We added the uncertainty of $\pm$ 35 kg/m³ on line 184.

**Lines 154-159: Why thinning? I would agree if reference is an ice core without ice lenses, but it needs to be show that this theoretical reference actually existed at the drill site earlier (2005).**

The thinning discussed in the original manuscript is due to annual densification of the accumulation from meltwater percolation and refreezing: a thinning that is not associated with mass loss. The reviewer is therefore correct that it did not indicate densification changes over time. To address changes in densification over time, we have incorporated historical data into the revised manuscript from the original Icefield Ranges Research Reports (as mentioned above), particularly from the 1964 density profiles of Grew and Mellor (1966). We have also included additional data from 2006, also from the Divide site. These locations are 12 km apart but are at a similar elevation, slope, and location in the icefield. Though we cannot say that their processes are exactly the same, they do possess similar stratigraphy and density (in 2006 and in 2018 (Kreutz, unpublished stratigraphy data).

**Discussion: The discussion is clearly structure but I perceived the flow of arguments as poor. The text meanders between more general, partly speculative and maybe too qualitative discussion of firn aquifers to the impact on geodetic mass balance estimates. It is not fully clear what the focus of the manuscript is, or what the main message(s) of the manuscript should be.**

We apologize for this lack of clarity. The main objective of the original manuscript was to characterize the firn of the upper Kaskawulsh Glacier: a significant ice mass within a major icefield where little or no published data is currently available on firn density or densification rates, meltwater retention, or liquid meltwater storage. The revised paper has been expanded to clarify and reiterate the three main messages: 1) firn density and ice content, 2) changes in densification rate, and 3) the new firn aquifer in this region. The results and discussion will focus around these three points. Number (3) is admittedly a bit of an aside, but it is of great interest and is relevant to meltwater retention and mass balance studies, as well as affecting the glacier thermal and hydrological behavior. Please refer to Section 3.3, 4.3, and the discussion for this additional analysis and interpretation.

**Lines 265: The statement cited from Christianson et al. (2015) appears incorrect. Already in the 1970s detailed studies of a perennial firn aquifer were carried out in the accumulation area of Abramov glacier (4400 m a.s.l.), Pamir-Alai, present-day Kyrgyzstan. In contrast to other studies, the scientists studied the aquifer in a deep firn pit (up to ~25 m deep). This allowed continuous monitoring of changes in the water table in relation to, e.g., surface melt intensity. The related studies, however, are mostly published in Russian (Glazirin et al., 1977; Kislov, 1982) and thus not widely known to a broader glaciological audience.**

We would like to include these Russian papers for reference in the manuscript. However, despite extensive searching we have not been able to find them and hope that the reviewer can forward them to us so that we can include the data mentioned in the above comment.

**Lines 270-272: Here an estimate of annual accumulation rate is mentioned, based on literature and the authors' own interpretation of the cores. Above (lines 196-198)**

**the authors use a literature value (other sources than here) of 1.76 m w.e. yr$^{-1}$. How do these two numbers relate? Is the implicit assumption made that accumulation rates have remained stable since the 1960s? What is the uncertainty introduced by this assumption?**

We have obtained a new dataset from 2004-2011 of snow accumulation and density data from the Divide site (12 km from our drill site, similar elevation) on the upper Kaskawulsh Glacier. We have included this data in our estimate of annual accumulation rates in order to provide more supporting evidence for the accumulation rate chosen here. We have also included the historical accumulation data present in the IRRP to aid in our snow accumulation estimate. With these additional data, the snow accumulation estimate is 1.8 m w.e. yr$^{-1}$. Please refer to section 3.3 and lines 321-329 for this additional information.

The interannual variability in this data could be used as an estimate of uncertainty. We have three additional lines of evidence for the annual accumulation rate: (i) our own winter accumulation measurements from spring 2018, ii) accumulation data from 2004-2011 and (iii) the (much) earlier published data from the IRRP. This was taken into account to assess a conservative uncertainty in the annual accumulation rate, which can then be propagated through to the uncertainty in the age of the core. Please refer to lines 331-34.

**Line 284 (as well as 184-190): 2 kg m$^{-3}$ appears to be a very low level of overall uncertainty. I assume there must be some potential sources of systematic errors that prevent such a very low uncertainty?**

We have double-checked our calculations. We use standard error analysis in these calculations, which we are happy to walk through in supplementary material if the reviewers would like to see it. To summarize here, our point samples (10-cm core sections) have significant measurement uncertainty, e.g., $500 \pm 75$ kg m$^{-3}$. Sources of uncertainty are random rather than systematic, to our knowledge. Figuratively speaking, for the case of random errors, averaging of the 10-cm density values for the whole core effectively leads to reductions in uncertainty because random errors cancel out. Based on standard error analysis, one can take the example of a 30-m core with 300 10-cm density values ($N = 300$). The standard error in the average follows $s_e = \quad /\sqrt{N}$, where in this illustration we can take $= 75$ kg m$^{-3}$. This gives $s_e = 4$ kg m$^{-3}$. We have nonetheless go through our uncertainty calculations again to ensure that these are accurate.

**Section 4.3: I think your interpretation stands on weak grounds. There is little evidence presented that accumulation rates from the 1960s are still valid today. As outlined below, the fact that ice lenses exist in the firn does not automatically mean that the surface lowers. For this to be true, the ice fraction has to change over time. The authors present some evidence of an increase in ice content (lines 288 to 291), but not for the time period represented by the two cores.**

We have strengthened our interpretations by bringing in more of the recent measurements

reported by Foy and others, as well as those made by ourselves in this region over the past ~15 years, to quantify whether accumulation rates have changed over time. The accumulation rates from the 1960s appear to still be valid today. These rates are similar to the ones measured by Copland and others from 2004-2011, which have been included in the new manuscript (lines 280-301). Though accumulation may have changed since 2011 our measurement of annual accumulation (1.8 m w.e.) is consistent with the variability in the 2004-2011 and IRRP data. The point we were making is that meltwater refreezing increases the density of the firn, thinning the annual accumulation layer without an associated mass loss. This also makes "dry firn" models inappropriate for density estimates that are needed for geodetic mass balance measurements, so even the basic reporting of firn density values and firn ice content are of value. We do understand that firn densification associated with multi-year changes in ice content is also of great interest, so we have included a new analysis of density changes over time in the revised manuscript. Please see the revised discussion section "4.2 Changes in the upper Kaskawulsh Glacier firn" (line 393) for this analysis and interpretation.

**Line 322: Surface lowering of 1.3 m: It is confusing to mention this result in cm yr$^{-1}$ in the abstract, not in the results (at least I couldn't find it there) and then again in m yr$^{-1}$ in the discussion.**

We have revised the manuscript such that the units are consistent throughout the text, thank you for pointing out this inconsistency. See lines 18, 501, 574. We recalculated the surface lowering as well and found it to be 0.73 m. We have elaborated on how this was calculated and the uncertainty associated with the value.

**Lines 322 – 332: I do not understand why there needs to be surface lowering because of the ice lens formation and refreezing? If we knew that there was no or less refrozen water in the firn in 2005, then the surface would have lowered as calculated. However, based on the evidence the authors present, I have the impression we do not know whether ice content has changes 2005-2018. If ice content in the firn would be constant, there would be no surface lowering.**

We understand the reviewer's point, and it makes it clear that we did not explain our calculations and objective clearly. In the original manuscript we reported the ice content present in the firn during the period of our study, which we used to determine the extent of surface lowering related to the meltwater percolation and refreezing affects within the firn. This process of densification creates an effective thinning, but not the multi-year changes in this process (i.e. classical firn densification). In the revised manuscript we have addressed this by using historical density information from the Icefield Ranges Research Reports to provide evidence as to how the firn density and ice content has changed over time. This historical data allows us to compare the density of the firn in 1964, 2006, and 2018. It is apparent that the density has increased and that there has been an increase in meltwater percolation and refreezing. This is verified by a new climate reanalysis melt model (New Section 3.4, 4.4) that indicates that due to warming temperatures more melt has occurred, thus causing more percolation and refreezing. This

occurs to a limit – once the firn is too warm for refreezing the meltwater is likely percolating down and forming a firn aquifer. This too is a new feature as indicated by an additional analysis of snow and firn temperatures from 1964 and 2006 (lines 421-426).

**Furthermore, the authors make the critical assumption of annual accumulation rate equaling 1.76 m w.e. yr$^{-1}$, leading to the conclusion that the core represents the time period 2005 to 2018 (Lines 196-198). What is the uncertainty of this assumption? Accumulation rates could have changed since the 1960s. Associated uncertainties are neither assessed nor discussed.**

This is now discussed in lines 321-334. We elaborate the different accumulation rates and how they have not varied significantly since the 1960s.

**Lines 353-355: The fact that an aquifer exist does not mean that the surface has to lower. Evidence is needed that firn properties altered over the time of investigation. If they have not changed (i.e. there was similar ice content earlier, an aquifer existed and accumulation rates remained constant), why should the surface lower?**

The presence of any kind of liquid water would be a form of firn densification, but in the sense that we were referring to (potentially confusing to readers): surface melting causes surface lowering, and where this meltwater is retained as liquid water or refrozen ice there is a measurable surface lowering that is not accompanied by mass loss. We have undertaken a comprehensive literature search in response to this comment, and found no evidence that the firn aquifer existed in the past. The presence of the new firn aquifer therefore makes it likely that the surface has lowered due to the firn aquifer's presence in the recent past, and we will provide estimates of how much this lowering has been. We have reported this in the updated manuscript, and provided supporting information. This supporting information is a climate reanalysis model, a firn mode, and historical data.

**References not listed in the manuscript**
**Glazyrin G.E., Glazyrina E.L., Kislov B.V. and Pertzinger F.I. (1977) Water level regime in deep firn pits on Abramov glacier [in Russian], volume 45. Gidrometeoizdat**

**Kislov, B.V. (1982) Formation and regime of the firn-ice stratum of a mountain glacier [in Russian]. Ph.D. thesis, SARNIGMI Tashkent.**

Please see the previous comments regarding the addition of these references into the manuscript.

**Anonymous Referee #2**

**Review "Meltwater Storage in the firn of Kaskawulsh Glacier, Yukon Territory, Canada" by N. Ochwat.**

**The authors study the density profile of two firn cores drilled in spring 2018 in the accumulation zone of Kaskawulsh Glacier (Yukon, Canada). These cores are used to calculate local firn density and the impact of meltwater retention and refreezing on surface lowering that must be accounted for to correct geodetic mass balance estimates. The authors obtain an average firn density of $670 \pm 2$ kg m$^{-3}$ in the 36 m deep core, and estimate an average surface lowering of $10 \pm 0.8$ cm per year over the period 2005-2018. The authors also identify a perennial firn aquifer below ~35 m depth. The paper suffers from major issues including the robustness of the methodology, results and uncertainty estimates, making the conclusions difficult to trust. In addition, some terms used are unclear; the authors sometimes expect a priori knowledge from the readers (e.g. Section 3.3). The reviewer also noted that results reported in the main text and tables are often not matching, and that the conclusions lack of novelty. The paper is mostly descriptive and does not provide novel insight on geodetic mass balance uncertainties compared to previous studies. Therefore, the reviewer deems that the manuscript should be rejected in its current form. Below, the authors can find the reviewer's major concerns, listed as General and Point comments.**

There is quite a bit to unpack here, and on some points we provide more detail with the specific responses below.

We have revised the manuscript to clarify the aims, methodology, results, and uncertainties. We apologize for confusion in the values reported – we were inconsistent and unclear in some places, and this was fixed. We have provided more supporting evidence indicating how the density of the firn has changed over time, using a range of studies completed since the 1960s, primarily from the Icefield Ranges Research Reports, as well as from previously unpublished field data collected by colleagues. We have removed the isotope section entirely as it was not fitting in our new manuscript. We have included more details on the uncertainty analysis and in the discussion of the various density values being reported in order to make the relationship between the table and the text more clear.

The main objective of the original manuscript was to characterize the firn of the upper Kaskawulsh Glacier: a significant ice mass within a major icefield where little or no published data is currently available on firn density or densification rates, meltwater retention, or liquid meltwater storage. The revised paper has been expanded and restructured. We have reiterated and clarified the three main messages: 1) firn density and ice content, 2) changes in densification rate, and 3) the new firn aquifer in this region. The results and discussion will focus around these three points. Number (3) is admittedly a bit of an aside, but it is of great interest and is relevant to meltwater retention and mass balance studies, as well as affecting the glacier thermal and hydrological behavior.

**General comments**
**1. Results are based on "subjective" approximations that may alter the conclusions. For instance, the completeness of the two firn cores section is assessed based on "visual inspection" by three persons. How do the resulting "random" and "human" errors impact the firn density calculated in Eq. 1? In L120, the authors provide a 10-20% uncertainty in estimating the factor f in Eq. 1 (L125-126)? This would lead to a ~100 kg m$^{-3}$ uncertainty in firn density (assuming the 670 kg m$^{-3}$ value reported here), in line with 110 kg m$^{-3}$ estimated in Foy et al. (2011; see L287). However, the authors report uncertainties ranging from 2 to 6 kg m$^{-3}$. Please elaborate. See also Point comment in L137-140.**

We have double-checked our calculations. We use standard error analysis in these calculations, and have expanded on this method in sections 3.1 and 3.2. To summarize here, our point samples (10-cm core sections) have significant measurement uncertainty, as the reviewer notes. There are numerous sources of measurements error, including the subjective assessment of the completeness of each core section. We will refer to these various sources of uncertainty as "random error" in the revised manuscript, as they are all believed to be random (vs. systematic) – sometimes we will measure or estimate a section as being too long or complete, sometimes we will underestimate it. The uncertainty factor of ~15% applies to point samples (10-cm core sections): say, for instance, $500 \pm 75$ kg m$^{-3}$ for a given sample. Conceptually, for the case of random errors, averaging of the 10-cm density values for the whole core leads to marked reductions in uncertainty because random errors cancel out. Based on standard error analysis, the standard error in the average follows $s_e = \ /\sqrt{N}$, where is the uncertainty in each data point and $N$ is the number of samples. Taking the example of a 30-m core with 300 10-cm density values, $N = 300$. Taking $= 75$ kg m$^{-3}$ as an example, $s_e = 4$ kg m$^{-3}$. We have gone through our uncertainty calculations again to ensure that these are accurate, but the main point here is the difference between sample uncertainties and the standard error in the mean.

**2. Across the manuscript, the authors report results that are not matching between the main text and tables, making the conclusions hard to trust. For instance, in L18 the authors report an average surface lowering of $10 \pm 0.8$ cm yr$^{-1}$ between 2005-2018. In L356, the authors report $10 \pm 8$ cm yr-1 for the same period. In L322, this annual rate is cumulated over the period 2005-2018 to obtain $1.3 \pm 0.8$ m in ~13 years. What uncertainty was used here (0.8 or 8 cm)? Please elaborate. Similar issues can be found across the whole manuscript and are listed in the Point comments.**

We have corrected the typos noted in the text, thank you for pointing those out. We have edited Table 1 and the manuscript in order to report consistent values and depths throughout the text. We report the upper 10 m of firn density to allow comparability to other literature on firn density. We chose to compare the partial core densities because the length of Core 1 and Core 2 differed – this permits a direct comparison.

**3. The 13-year period (2005-2008) is estimated using calculated total water content of 23.22 m w.e. at the drilling site and assuming an average accumulation rate of 1.76 m w.e. yr$^{-1}$ (1960s). The authors do not assess the robustness of this estimate given the uncertainty in firn density. Please elaborate.**

We have addressed this in a new section, Section 4.3.

**4. The term "melt-affected firn" is often used in the manuscript but not explained. Is this firn affected by the presence of refrozen meltwater in pore space? What are the associated visual features as stated in L204-205? Perhaps a photo of the cores would help the interpretation. The same holds for "Ice content" in L134, that is sometimes defined as the cumulative thickness of ice layers in the core expressed in m, or as a fraction after being normalized by the length of the firn core (see e.g. L192 and Table 1).**

A definition of "melt-affected firn" has been added to increase clarity in the methods section. "Melt-affected firn" is any firn that displays physical characteristics indicating that there was the presence of liquid water at some point. This can result in ice layers, ice lenses, or can be indicated by the lack of grain boundaries, the presence of air bubbles, texture, and opacity. "Melt-affected firn" can also be identified using stable isotopes and the cation/anions, however, this was not done in the field. [Lines 102-104]

The use of the term "ice content" is used with more precise wording in the text in order to clarify as to whether or not it is describing cumulative ice layers or the normalized fraction. We refer to the former as the "total ice content" and the latter as "ice fraction". This has been changed in table 1, lines 176, 184, 227, 266, 267, and 364.

**5. The authors sometimes expect "a priori" knowledge from the reader. Section 3.3 on stable isotopes is a good example: how to interpret the summer peaks at -22‰ in Fig. 4? This section is not necessary and the results are not further discussed in the text, except in L244-246 that relates low ion concentrations to active meltwater percolation/motion in firn.**

We have removed the section on stable isotopes.

**6. The conclusions lack of novelty compared to previous studies that also estimated surface lowering in the region (see L334-339). The paper does not provide a convincing estimate of (local) surface lowering uncertainty for geodetic mass balance measurements, nor estimate the regional mass change accounting for density correction. In L328-330, the authors claim that density estimated at the two cores are representative of a larger region, which cannot be proved using only two cores as stated in L371-376. The authors should consider combining their core measurements with firn modeling to obtain spatially continuous density profiles and**

**estimate regional mass balance uncertainty due to firn processes.**

There are several good points here, mostly related to points addressed above. Within our objective to characterize the firn density, densification rates, and meltwater retention on the upper Kaskawulsh Glacier, we did not initially set out to quantify changes in densification rates over time – only the annual densification associated with meltwater refreezing, and the resulting firn density profile. We have reorganized the manuscript and incorporated new data, historical data from 1964 as well as data from 2006. We have used this data to discuss the changes in density over time and densification rate. We have also included a surface energy balance model and firn model. Additionally, we have expanded on our methodology for the uncertainty and discussed it in greater detail.

It is difficult to know whether our two cores are representative of the larger accumulation area of this icefield or others in the St. Elias region, but they still provide information where little other recent work on firn properties has been undertaken. We have revised our discussion and not over-extended our claims in the revised manuscript.

**Point comments L92-94: Are the measurements from the snow pit discussed somewhere in the manuscript or shown in Fig. 2? Please clarify.**

The snowpit measurements are presented in Figure 2 and are included in the density data and inference about annual snow accumulation. We edited the caption of the figure in order to clarify that the measurements are part of the density figure. "The first meter of data is from the snowpit." Is added to figure caption 2.

**L135-136: What does "melt percent" mean? How is this calculated?**

We have explained this more clearly in the manuscript. "Melt percent" has been used in the literature (e.g., Koerner, 1977) to refer to the percent of annual snow accumulation that melts (and refreezes), in the accumulation area of polar environments. At our site we don't use this concept but use "ice content" to refer to the fraction of a core sample that is made up of refrozen meltwater.

Please see lines 164-171 for this elaboration.

**L137-140: This is unclear, why should the thickness of ice lenses be divided by a factor two?**

Ice lenses were partial ice layers, where the ice did not extend horizontally through the whole core section. We assume that, on average, the ice lens occupied 50% of the core; therefore the measured thickness was divided by two. We have added "In core samples that had ice lenses, ice lens diameter, on average, occupied 50% of the core sample; therefore their thickness was divided by two before being summed." [Lines 168-171]

**L161-164: The authors should provide some references on the methods used to study isotopes.**

We have removed the section on isotopes.

**L184-190: This paragraph includes numerous errors in reporting results. In L186, "571 ± 3 kg m$^{-3}$" is reported in the text while Table 1 lists 518 kg m$^{-3}$ at core 2 between 4-14 m depth. In L187, "608 ± 2 kg m$^{-3}$" is reported while Table 1 lists 618 kg m$^{-3}$ between 4-21 m depth. The authors report an extremely small density uncertainty of 2-3 kg m$^{-3}$ while Figs. 2a and b show much larger uncertainties. In L229- 230, the authors state that densities larger than 917 kg m$^{-3}$ are eliminated. However, Fig. 2a shows values of ~1000 kg m$^{-3}$ or larger at 6 and 10 m depth. To the reviewer, it is hard to judge whether these errors are due to negligence or calculation errors. Please elaborate.**

The uncertainties of point samples versus average values for the cores are discussed above (please see the response to point #1). A more detailed explanation of the error analysis has been included in the revised paper (Line 128-178).

Figure 2 shows the point data (10-cm sections). The outliers have been removed from Figure 2 in order to be clearer, because they were removed from the calculations to determine density and background firn density.

Line 187 referred to the average of core 1 and core 2 of the density of firn at both locations – this was not reported in the table. A row has been added for the average to help clarify the confusion. This is reported in both sentences in line 187 and 372.

**L185, 187, 188: For clarity, the authors should better write: "between 4 and 14 m depth" instead of "in the upper 10 m"; "between 4-21 m depth" instead of "in the upper 17 m"; and "between 4-36 m depth" instead of "representing ~32 m". The same holds for L284-286.**

We have edited the manuscript throughout in order to be clearer as to the depths we are referring to when discussing the firn. We have also reworded it in the way you suggested.

**L193: 660 kg m$^{-3}$ is actually 1.5% smaller than the firn density of 670 kg m$^{-3}$ reported in L189.**

We have significantly altered the manuscript and no longer have this sentence in it.

**L276: What do the authors mean by "summer melt extent"? Do they mean meltwater production in mm w.e. yr$^{-1}$ as listed in Table 2? Please clarify.**

This sentence will be reworded so that instead of saying "summer melt extent", it will be clarified as meltwater production in m w.e./yr, as listed in Table 2 (which is now table 1).

We also refer to the "last summer surface" and clarify what that means in the text.

**L278: It is hard to assess the robustness of the results in this paragraph. In L278, the authors state that summer 2015 was the warmest in the period 2014-2018, whereas Table 2 shows that it was actually summer 2016 (-1.0ºC in 2016 vs. -1.8ºC in 2015). The same goes for annual mean temperature in 2015-2016 (-9.0ºC in 2016 vs. -9.6ºC in 2015). How to interpret the larger PDD and melt rates in 2015 then? Please clarify.**

This has been removed entirely and replaced with a surface energy balance model please refer to sections 3.4 and 4.4.

**L284: Again 608 kg m$^{-3}$ is reported in the text whereas 618 kg m$^{-3}$ is listed in Table 1.**

All of the density measurements have been recalculated.

We have addressed this by adding the following lines [440-450]:

"Research in Greenland proposes that ice-layer formation and the presence of firn aquifers may delay surface run-off due to the water storage characteristics of firn (eg. pore space availability, water at interstitial grain boundaries, etc) (Fountain and Walder, 1989; Scheider, 1999). If ice layers become too extensive or thick, they can form an 'ice slab,' a thick impermeable barrier that leads to enhanced surface runoff (MacFerrin et al., 2019). The thickness of ice layers that prevents percolation varies and depends on the local climate and conditions of the firn. For example, in Greenland 12-cm thick ice layers were still permeable (Samimi et al., 2020) whereas Bell et al., (2008) reports a 1-2 cm ice layer prevented percolation at the Devon Ice Cap, Canada."

**Table 3: What does "1.5-2g" mean in the personal communication of Sass and O'Neel?**

Thank you for noticing the "g" that should not have been there and was removed.

---

## Referee Report (RR1)

**General Remarks**

The study by Ochwat et al. has been modified substantially. I would like to thank the authors for making a thorough effort in revising their study. Generally, I believe the study has improved and has gained in focus. The inclusion of firn modelling supports the main message that the aquifer is likely a new phenomenon (but also raises some questions, see below under detailed remarks).

My main concern is that the authors compare firn cores that are as much as ~10 km apart (the 1964 Divide core and the 2018 Kaskawulsh core, the distance between the 1964 and 2006 Divide cores is much smaller but unknown), while not commenting on potential issues of such a comparison. I am concerned with this point based on my own research experience: I was involved in a study quantifying firn changes from 1973 to 2018 on Abramov glacier, Kyrgyzstan, Central Asia (the paper will be published in Journal of Glaciology within the next couple of weeks). When we visited the glacier in winter 2018, we thought that we knew the exact location of the historic 1973 firn profiles and drilled at that site. When back from the field, we received further historic documents which showed us that we had drilled about 250 m away from the historic site. We visited the glacier again in summer 2018 and drilled at the historic location. Together with extended GPR measurements, this provided us with the possibility to quantify short-scale variability of accumulation rates and firn properties on the relatively large mountain glacier (24 km$^2$). Although both drill sites look very similar (they both seem so be located in the same flat plain), the variability is large. Over a distance of 250 m, mean annual accumulation varies by a factor of almost 1.7. If we had compared the historic 1973 profile to our core drilled 250 m away from the historic site, our conclusions with respect to firn changes (accumulation, firn ice content, and more) would have been dramatically wrong. I am aware that Abramov glacier probably shows more small-scale variability as the accumulation area of the much larger Kaskawulsh. But even on the Greenland ice sheet, I would be cautious when comparing cores that are ~10 km apart. Hence, I would like to ask the authors to at least mention such potential issues and to thoroughly evaluate and discuss to what degree drill sites, conditions at the drill sites and the cores are comparable.

**Detailed Remarks**

*Lines 315-319:* I do not agree to these statements. The fact that the firn was at 0 °C in 2006 (0 to 10 m depth) and cold between ~1 m and ~15 m in 1964 does not proof firn warming. To reliably quantify firn temperature changes, temperatures below the depth of zero annual amplitude (roughly 10 to 15 m depth) need to be compared. There is a high risk that the differences discussed here and shown in Fig. 5 are just the result of weather conditions and not a climatological signal. This risk is amplified (i) by comparing average temperatures for only the top 10 m and (ii) by the 1964 core showing 0 °C at ~15 m depth, potentially indicating that the firn was temperate already then. At the minimum it needs to be acknowledged that the data have to be used with care when quantifying firn temperature changes. I also suggest evaluating and discussing potential uncertainties.

*Lines 351 – 352:* "Meltwater which does not ..." I do not understand the statement, does the model also simulate lateral drainage? If yes, consider updating the model description. If not, please remove the statements as this would not be a result of your modelling efforts.

*Line 375:* "These Characteristics …" a bit confusing, the previous sentence describes the current situation, not the original situation.

*Lines 379 – 380:* As mentioned above, I do not think the data fully support this conclusion.

*Lines 388 -394:* This could be placed in the introduction or description of data and methods. In my opinion, these are general statements based on the literature and do not fully fit here in the discussion.

*Line 397:* "… affects continue …", something is wrong here.

*Line 399:* I do not understand these comments. Sorge's Law has been derived from the study of dry firn at Eismitte, roughly at 3050 m near the centre of the Greenland ice sheet (*Sorge,* 1935). It is intended to reflect certain basic characteristics of dry firn under a constant climate. It was not intended to be valid for firn which experiences substantial melt under a warming climate. To my understanding, this is also how the law is formulated and the term "Sorge's Law" coined in *Bader* (1954).

*Lines 437 – 438:* My apologies for not making the Central Asia glacier studies available that document a firn aquifer on a mountain glacier already in the 1970s. Please find the studies by *Glazirin et al.* (1977) and *Kislov* (1982) available for download under this link: https://drive.switch.ch/index.php/s/51wrYzVb9r4XRSh

*Lines 489-490:* "This is a conservative estimate …" I do not fully understand what is meant here.

*Figure 1:* I could not find a clear reference to the IRRP A Site in the text. I suggest adding a clear reference to this site in the text (what, when and who measured there) or remove the site from the figure.

*Figure 6:* Which location is modelled? While a location is clearly indicated in, e.g., Figure 5, there is no information here and also in the text where Figure 6 gets referenced. On line 233 it says that "the study site" is modelled, furthermore it is stated that model forcing ERA data are compared to the "Divide" meteorological observations. However, the study refers to a rather large area with locations at different elevations. Hence the question what exactly is modelled?

*Figure 6:* Related to my comment above: It is not fully clear what is modelled and whether the model output can be compared to the Divide, the Kaskawulsh field data or both? Nevertheless, I note that the modelled firn temperatures at 10 and 20 m depth in the 1960s are around -12 °C while the 1964 core drilled at Divide indicates a temperate firn regime (0 °C at ~15 m depth). I consider this a substantial disagreement between measurements and model.

*Figure 7:* It looks like there are white areas at the top of Figure 7a (data gaps or variations in surface elevation?) which do not appear in Figure 7b. Why the difference?

*References:* Now the Machguth et al. citations are fully confused. There is a 2016 paper in Nature Climate Change and a 2006 paper in Geophysical Research Letters. You have now created a combined citation of both papers :-) Please correct this citation but also check all the other references for correctness.

**References not listed in the manuscript**

Glazyrin G.E., Glazyrina E.L., Kislov B.V. and Pertzinger F.I. (1977) Water level regime in deep firn pits on Abramov glacier [in Russian], volume 45. Gidrometeoizdat

Kislov, B.V. (1982) Formation and regime of the firn-ice stratum of a mountain glacier [in Russian]. *Ph.D. thesis,* SANIGMI Tashkent.

Sorge, E. (1935), *Glaziologische Untersuchungen in Eismitte, Wissenschaftliche Ergebnisse der deutschen Grönland-Expedition Alfred Wegener 1929 und 1930/31,* in: K. Wegener, im Auftrag der Notgemeinschaft der Deutschen Wissenschaft (Ed.), Band III, Glaziologie.

---

## Referee Report (RR2)

[referee-annotated manuscript omitted]

---

## Author Response (AR2)

Dear Dr. Etienne Berthier,

Thank you for the extension once again and allowing us the time to implement significant changes into the manuscript. We have included a track-changes document and a point by point description of the changes (below). We have added significant new material regarding the firn modelling and addressed comments about the spatial variability – among other things. We have also added supplemental information in order to go into greater detail on the modelling. We have uploaded a video supplement but it is still waiting to be given a DOI. Lastly, we have changed the title to "Evolution of the firn pack of Kaskawulsh Glacier, Yukon: meltwater effects, densification, and the development of a perennial firn aquifer" to better match the new manuscript. We believe these changes will make the manuscript much stronger.

Thank you,
Naomi Ochwat, on behalf of the coauthors.

Below you will find the description of the edits the reviewers requested and how we responded to them.

**Reviewer 1**

**General Remarks**
The study by Ochwat et al. has been modified substantially. I would like to thank the authors for making a thorough effort in revising their study. Generally, I believe the study has improved and has gained in focus. The inclusion of firn modelling supports the main message that the aquifer is likely a new phenomenon (but also raises some questions, see below under detailed remarks).

My main concern is that the authors compare firn cores that are as much as ~10 km apart (the 1964 Divide core and the 2018 Kaskawulsh core, the distance between the 1964 and 2006 Divide cores is much smaller but unknown), while not commenting on potential issues of such a comparison. I am concerned with this point based on my own research experience: I was involved in a study quantifying firn changes from 1973 to 2018 on Abramov glacier, Kyrgyzstan, Central Asia (the paper will be published in Journal of Glaciology within the next couple of weeks). When we visited the glacier in winter 2018, we thought that we knew the exact location of the historic 1973 firn profiles and drilled at that site. When back from the field, we received further historic documents which showed us that we had drilled about 250 m away from the historic site. We visited the glacier again in summer 2018 and drilled at the historic location. Together with extended GPR measurements, this provided us with the possibility to quantify short-scale variability of accumulation rates and firn properties on the relatively large mountain glacier (24 km2). Although both drill sites look very similar (they both seem so be located in the same flat plain), the variability is large. Over a distance of 250 m, mean annual accumulation varies by a factor of almost 1.7. If we had compared the historic 1973 profile to our core drilled 250 m away from the historic site, our conclusions with respect to firn changes (accumulation, firn ice content, and more) would have been dramatically wrong. I am

aware that Abramov glacier probably shows more small-scale variability as the accumulation area of the much larger Kaskawulsh. But even on the Greenland ice sheet, I would be cautious when comparing cores that are ~10 km apart. Hence, I would like to ask the authors to at least mention such potential issues and to thoroughly evaluate and discuss to what degree drill sites, conditions at the drill sites and the cores are comparable.

This is an interesting point, and we respect these concerns. We feel comfortable with the general climatic homogeneity of the plateau/divide region of the Kaskawulsh Glacier, given what we know about it, but agree that this requires appropriate qualification and caveats. Our own two cores, about 1 m apart, have considerable stratigraphic variability, despite similar winter snow depths and average firn densities, so we appreciate the complexity of spatial variability with both accumulation rates and melt-affected firn.

Our comfort comes partly from the relatively flat and uniform character of the plateau. There are nunataks, but the elevation of the glacier ranges by less than 200 m (2550 ± 100 m) over an area of more than 63 km$^2$, and available snow accumulation data from our site, the Copland site, and the IRRP A 1960s core sites are all consistent, with average values in the range of 1.6 to 1.8 m w.e. yr$^{-1}$. The latter number, 1.8 m w.e., is close to what we measured for the 2017-2018 winter accumulation at our core site and based on isotopic data from within the core (three consecutive winter (negative) peaks that were not washed out). This is also the average value found from 7 years of measurements at the Copland site, about 12 km away. So we agree that there can be strong horizontal variations in snow accumulation, and have seen this, but the available data support that accumulation rates are similar where it has been measured on the upper Kaskawulsh. The value we use, 1.8 m w.e., is partially informed by data at the core site, as well as the regional data. We do include some uncertainty analysis associated with this value, and have now added some discussion concerning potential issues of spatial variation, L. 89-94

**Detailed Remarks**
*Lines 315-319:* I do not agree to these statements. The fact that the firn was at 0 °C in 2006 (0 to 10 m depth) and cold between ~1 m and ~15 m in 1964 does not proof firn warming. To reliably quantify firn temperature changes, temperatures below the depth of zero annual amplitude (roughly 10 to 15 m depth) need to be compared. There is a high risk that the differences discussed here and shown in Fig. 5 are just the result of weather conditions and not a climatological signal. This risk is amplified (i) by comparing average temperatures for only the top 10 m and (ii) by the 1964 core showing 0 °C at ~15 m depth, potentially indicating that the firn was temperate already then. At the minimum it needs to be acknowledged that the data have to be used with care when quantifying firn temperature changes. I also suggest evaluating and discussing potential uncertainties.

Thanks for raising this, we totally agree. Temperatures and densities in the seasonal snowpack and near-surface firn cannot be compared from one year to another, as they are recording weather rather than long-term trends. Whether it was a cold winter or warm spring, etc. We have now removed this from the results and discussion, including Figure 5. Depths greater than 10 m are safer, as these are below the annual air temperature wave

(via thermal diffusion), but the 'textbook' understanding that 10-m temperatures reflect mean annual conditions also does not apply where there is extensive meltwater refreezing (latent heat release), and where this can vary a lot from year to year. This is well illustrated in the Figures in the new Supplemental Information, and also by the observation of deep temperate firn at this site, despite mean annual air temperatures of about –11°C.

The second point, that Grew and Mellor (1966) reported temperate firn at 15-m depth in 1964, is particularly important, and we apologize that we missed this. We divided the task of revising the manuscript and the person charged with the modelling effort (SM) was aware of the Grew and Mellor core from the Icefield Ranges Research Program reports, but not of this particular CRREL technical paper, or the borehole temperature measurements. We therefore did not present a consistent story of the firn warming, and had not realized that the initial conditions in the firn model were inconsistent with this observation. An embarrassing error on our part. We now acknowledge this discrepancy, and perform sensitivity tests on the initial conditions in the model to try to better understand the model disagreement. We still present a similar 'reference model' for the 55-year firn temperature evolution, which predicts significant firn warming and the recent development of the firn aquifer, but are much more cautious in our interpretation. This reference model uses the bias-adjusted ERA climatology and the firn model parameters as calibrated at DYE-2 in Greenland (Samimi et al., 2020), but is not locally calibrated or validated, as the reviewer notes, so we cannot be too confident in either this model or the climatological forcing.

That said, we model strong fluctuations in the 10- and 20-m temperatures through the period 1965-2019, and even in the case where the model is forced to produce temperate deep firn in the spin-up simulation (Figures 7, 8, and S4), the firn refreezes in the 1970s and experiences decadal-scale warming trends, not returning to temperate conditions until ~2015. We discuss these deep-firn temperature trends from the model, but are now more careful in arguing for firn warming at this site, making clear that this is speculative and based on a model result. The inference that firn has warmed at this site is now removed from the title, abstract, and conclusions.

*Lines 351-352:* Meltwater which does not ..." I do not understand the statement, does the model also simulate lateral drainage? If yes, consider updating the model description. If not, please remove the statements as this would not be a result of your modelling efforts.

This is correct, thank you – we rewrote this sentence, Lines 381-382. In the model, water that percolates to the base of the firn column (35 m in the model re-runs for the revisions) is assumed to leave the system, so we refer to this as 'runoff' or 'drainage', but we don't explicitly model lateral runoff. See Line 381-383 and Lines 589-591.

*Lines 375:* These Characteristics ..." a bit confusing, the previous sentence describes the current situation, not the original situation.

This sentence has been deleted because we appreciate we cannot say for certain that all of the characteristics listed in the previous sentence have changed since 1964.

*Lines 379-380:* As mentioned above, I do not think the data fully support this conclusion.

The density results support this conclusion, but we agree, not the temperature data. We have removed the last part of the sentence "and temperature". This section has been extensively rewritten, Lines 449-462. We still discuss the possibility of firn warming, but note that this is just a model result and we don't have confidence in it.

*Lines 388-394:* This could be placed in the introduction or description of data and methods. In my opinion, these are general statements based on the literature and do not fully fit here in the discussion.

This is a good comment; it's true, this was formulated as more of a literature review. We reworded and shortened this discussion throughout this paragraph, now Lines 464-474. We retain some discussion of the processes as part of the interpretation of our density structure.

*Line 397:* "... affects continue ...", something is wrong here.

This has been addressed in the rewritten text, Line 469.

*Lines 399:* I do not understand these comments. Sorge's Law has been derived from the study of dry firn at Eismitte, roughly at 3050 m near the centre of the Greenland ice sheet (*Sorge, 1935*). It is intended to reflect certain basic characteristics of dry firn under a constant climate. It was not intended to be valid for firn which experiences substantial melt under a warming climate. To my understanding, this is also how the law is formulated and the term "Sorge's Law" coined in *Bader* (1954).

The discussion concerning these two points has been reformulated: "However, with increasing meltwater percolation and refreezing effects, higher densities are common in the upper portions of the firn, as observed in our cores," Lines 469-470.  We use the discussion to explain that the firn on Kaskawulsh does not follow Sorge's Law for dry densification, though perhaps this is too obvious. We shortened and simplified this section, but still make some comments on the complex density structure caused by the meltwater refreezing.

*Lines 437-438:* My apologies for not making the Central Asia glacier studies available that document a firn aquifer on a mountain glacier already in the 1970s. Please find the studies by *Glazirin et al.* (1977) and *Kislov* (1982) available for download under this link: https://drive.switch.ch/index.php/s/51wrYzVb9r4XRSh

This sentence "Apart from these studies, there have been no other published reports of PFAs on mountain glaciers (Christianson et al, 2015)" has been removed.  We have also added a few sentences about other PFAs found, one on Abramov glacier in Russia by Glazyrin et al., 1977 (translated by Eduard Khachatrian) and another on Austfonna Ice Cap

(Zagorodnov et al., 2006), Lines 531-535. Unfortunately, we were not able to translate the PhD thesis.

*Lines 489-490:* This is a conservative estimate ...” I do not fully understand what is meant here.

This has been reworded to "this estimate of thinning is likely low" on Line 603.

*Figure 1:* I could not find a clear reference to the IRRP A Site in the text. I suggest adding a clear reference to this site in the text (what, when and who measured there) or remove the site from the figure.

Thanks for pointing out this omission. This is the IRRP site for the earlier snow-pit work and the 15-m borehole. In the IRRP there are several sites that range in alphabetical name (A, B, etc). An additional sentence has been added to the caption "IRRP A site is the site of the 1964 firn core (Grew and Mellor, 1966)." We also rewrote the study area section to better explain this, Lines 87-91.

*Figure 6:* Which location is modelled? While a location is clearly indicated in, e.g., Figure 5, there is no information here and also in the text where Figure 6 gets referenced. On line 233 it says that "the study site" is modelled, furthermore it is stated that model forcing ERA data are compared to the "Divide" meteorological observations. However, the study refers to a rather large area with locations at different elevations. Hence the question what exactly is modelled?

This is intended to be a model of the core site, at an elevation of 2640 m in the upper Kaskawulsh accumulation area, but in truth there is nothing in the model that is specific to that point (60.78°N, 139.63°W), other than the elevation for the bias-adjustments in the meteorological forcing. Rather, the model should be interpreted as representative of general conditions (the glaciological and climatological setting) of the upper plateau and Kaskawulsh-Hubbard divide region. The ERA5 reanalysis uses 0.25° grid calls, ca. 28 km, so our climate forcing is a general regional representation, not specific to a particular point. We discuss this now, Lines 262-269.

*Figure 6:* Related to my comment above: It is not fully clear what is modelled and whether the model output can be compared to the Divide, the Kaskawulsh field data or both? Nevertheless, I note that the modelled firn temperatures at 10 and 20 m depth in the 1960s are around -12 °C while the 1964 core drilled at Divide indicates a temperate firn regime (0 °C at ~15 m depth). I consider this a substantial disagreement between measurements and model.

Agreed that this is a substantial disagreement, which we now discuss directly, and as discussed above. But in terms of the modelling, these sites (IRRP A, the Copland weather station, our core site, the Copland camp) are all within the same ERA5 grid cell, so our meteorological forcing cannot be compared with AWS data or something that is specific to

a point.  It is at best a representation of the regional climatology. It is our assumption that similar elevations within the accumulation area of the broad plateau region will experience similar meteorological conditions. We explicitly state this assumption now, Lines 262-269.

*Figure 7:* It looks like there are white areas at the top of Figure 7a (data gaps or variations in surface elevation?) which do not appear in Figure 7b. Why the difference?

Apologies, these were 'off-scale' (below -20°C) temperatures that saturated in the contour plot and were rendered white. Corrected in the revised Figure 6.

*References:* Now the Machguth et al. citations are fully confused. There is a 2016 paper in Nature Climate Change and a 2006 paper in Geophysical Research Letters. You have now created a combined citation of both papers :-) Please correct this citation but also check all the other references for correctness.

Apologies for the confusion. We have corrected the references and double-checked everything again.

**References not listed in the manuscript**
Glazyrin G.E., Glazyrina E.L., Kislov B.V. and Pertzinger F.I. (1977) Water level regime in deep firn pits on Abramov glacier [in Russian], volume 45. Gidrometeoizdat

Kislov, B.V. (1982) Formation and regime of the firn-ice stratum of a mountain glacier [in Russian]. *Ph.D.thesis,* SANIGMI Tashkent.

Sorge, E. (1935). Glaziologische Unterzuchungen in Eismitte. Wissenschaftliche Ergebnisse der Deutchen Gronland-Expedition Alfred-Wegener 1929 und 1930-1931, 3, 270. in: K. Wegener, im Auftrag der Notgemeinschaft der Deutschen Wissenschaft (Ed.), Band III, Glaziologie, 1935*.*

We have added Glazyrin et al. (1977) and Sorge (1935), but we could not find a translator for the PhD thesis.

**Reviewer 3**

**Suggestions for revision or reasons for rejection (will be published if the paper is accepted for final publication)**
The manuscript describes firn evolution at a high elevation site on Kaskawulsh Glacier, St.

Elias Mountains, Yukon. This is a highly relevant topic and the presented results contribute to an improved understanding of ongoing trends in firn density, temperature and potential development of firn aquifers in mountainous environments. Generally, I find that the observational datasets are well described and interpreted. Also, the comparison with model output is valuable. Still, I have some moderate to major concerns that I would like to see addressed, primarily related to the lack of model calibration and the interpretation of surface lowering. If it would be an option to perform some additional model experiments, I would highly recommend to perform some additional runs to calibrate melt rates e.g. by minimizing the misfit between modelled and observed subsurface temperatures. Right now, discrepancies between the model and observations are hardly discussed and the lack of model calibration makes it difficult to draw strong conclusions on trends in firn conditions. My specific comments are listed below.

Specific comments

L39-41: This needs to be reformulated. The phrase "If the surface continues to melt" should be removed, since refreezing will happen directly when melt water enters cold snow/firn. 'Warming firn' does not necessarily lead to more refreezing, rather the opposite.

Reworded as suggested, Line 40.

L42: A firn aquifer only forms if the water does not directly run off through moulins/crevasses.

Agreed; we added a short comment to note this, Lines 42-43.

L42-44: It would be good to consistently use "perennial firn aquifer" rather than "firn aquifer", since here long-term (multi-annual) storage of water in firn is meant.

Thank you, revised to reflect this suggestion and now "PFA" is defined and used throughout.

L51-52: Another useful reference for the Canadian Arctic is Noël et al. (2018; https://doi.org/10.1029/2017JF004304), and for Svalbard references to Van Pelt et al. (2019; https://doi.org/10.5194/tc-13-2259-2019) and Noël et al. (2020; https://doi.org/10.1038/s41467-020-18356-1) could be relevant to add.
L55: Here Machguth et al. (2016; https://doi.org/10.1038/nclimate2899) could be cited.

We have added these references as suggested.

L69: I suppose this refers to air temperature? Please clarify.

Yes, clarified as suggested Line 81.

L107-117: This part fits better at the start of the next section (3.2).

Rearranged as suggested, Lines 132-144.

L137: I suppose L_unc should be dL (or dL in the equation should be L_unc).

Thank you, yes, revised to *dL*, Line 156.

L186-190: "Surface lowering associated with refreezing" is confusing. Surface melting leads to thinning and gravitational settling of the snow as well, but refreezing just adds mass to the existing vertical column and does not (directly) cause any thinning. See also my later comment.

Yes this is true, that is what we intended to convey. Revised to " There is surface lowering associated with melting but without associated mass loss, due to subsurface refreezing", Line 205. Also revised to clarify this in Lines 211-212.

Section 3.4: It appears that no calibration of the model has been done, presumably because there were no melt observations to compare to (?). This currently makes it very hard to trust the model output, especially since there appear to be major biases in modelled subsurface temperatures, which may indicate an underestimation of melt rates. See also my later comment.

This is a fair comment, and we agree – lacking local energy balance, melt, or firn observations to constrain the modelling, it should not be interpreted as a rendering of reality. Rather, the modelling has been added to explore potential scenarios for the long-term firn evolution at the core site, and to examine questions of whether the PFA may be a recent or long-term feature.  We agree that since the model is not observationally constrained, much more sensitivity analysis is needed here, and we should not be presenting just one model scenario as 'truth'.  We have now added a range of model sensitivity experiments to a) perturbations in the climate forcing, b) sensitivity to model parameters, and c) sensitivity to the spin-up (initial) conditions. Most of this has been added to a new Supplementary Information section, but we also added new Figures 7 and 8 and a discussion of model sensitivity in the main text. The 'reference model', using the ERA climatology, and with model parameters from a calibration at DYE-2 in southern Greenland, is still presented as our 'default' scenario, but the model sensitivity experiments show that this reference-model firn evolution is not well-constrained. Significant biases in the climatology or different assumptions about irreducible water content, in particular, can lead to either permanently temperate or permanently polythermal firn at this site (permanent referring to the simulation period, 1965-2019). This is presented and discussed now, and we feel that the only robust conclusion is that the climatological and glaciological setting of the upper Kaskawulsh render it very close to the tipping point between polythermal and temperate firn.

It is hard to know whether there are major biases in the subsurface firn temperatures, as there are limited data. But the reviewer is correct, the only historical data we do have, from

borehole air temperatures in a 15-m firn core collected in July, 1964, indicate temperate conditions at 15-m depth at that time (Grew and Mellor, 1966). Spot measurements of borehole air temperatures during the summer melt season are not strongly reliable, when meltwater can enter the borehole and mixing occurs with the surface air (e.g., wind-pumping, convection). The methods used for the 1964 borehole temperature measurements are not well documented in Grew and Mellow (1966), but temperature appears to have been logged at 1-m intervals over a short time (< 1 day), without evidence of capping the borehole or installing a thermistor string for an extended period (e.g., as discussed in Zagorodnov et al., 2006). However, taking these data at face value, there is certainly a chance that firn has been temperate since the 1960s (or longer) at this site. We discuss this now, and model experiments explore the changes in model settings that are necessary to produce temperate firn at this time (Figures 7 and 8). It requires: (i) a warming of at least 1.9°C in the ERA climate forcing, (ii) an increase of more than 36 W m$^{-2}$ in the ERA incoming longwave radiation, (ii) an increase of at least 62 W m$^{-2}$ in the ERA incoming shortwave radiation, or (iv) snow albedo values of less than 0.65 (vs. our estimate of 0.78, as a JJA mean value), or (v) some combination of the above. On their own, these represent strong anomalies or significant deviations from the reference model, particularly as the ERA temperature and radiation data have been bias-adjusted against regional observations, but a combination of these biases may certainly be possible. Low values of capillary water retention, e.g. irreducible water contents of less than 1.2% by volume, also facilitate deeper meltwater infiltration and firn warming to greater depths, but are not enough on their own to bring the firn to a temperate state. In short, we cannot rule out that the model underestimates melt extent and/or meltwater infiltration, and therefore has a cold bias in the firn temperatures. We acknowledge this now and present some of the model experiments discussed above. We discuss this further below in response to the reviewer's questions about model spin-up assumptions.

Section 3.4: Most likely the subsurface model also simulates density evolution, but this is not mentioned here and no graphs of it are shown in the results. It would be an important validation of the model results if it could be shown that simulated density evolution matches the observed densities well.

True, the firn model does simulate densification and the stratigraphic evolution of ice layers, though this has not been calibrated or constrained in the DYE-2, Greenland work as well as the coupled thermal and hydrological parameters have been evaluated, via thermistor measurements and TDR probes in firn pits (Samimi et al., 2020). We have added the modelled densities and densification rates to Table 2 and Figure 5. The increase since the 1960s is consistent with the observations, with average firn density (from 4- to 35-m depth) increasing from ~640 to ~680 kg m$^{-3}$ from 1965 to 2019. The modelled firn density in 2018, the time of acquisition of the firn core, was 682 kg m$^{-3}$, compared with the observed value of 698 kg m$^{-3}$. This is now discussed, Lines 403-404.

Section 3.4: I am missing a description of how the model and in particular the subsurface conditions were initialized, i.e. if some spin up has been done.

Apologies, this should have been explained. This has been added to the manuscript and Supplemental Information in some detail now, with Figures 7 and 8 also illustrating the sensitivity to model spin-up assumptions. We start with linear temperature and density profiles and then do a 30-year spin-up with perpetual 1965 climatology, i.e., running the energy balance and firn model through the 1965 annual evolution 30 times. This develops the 35-m temperature, density, and ice-layer stratigraphy that is used for the initial conditions. There is no memory of the initial linear profiles, but the spun-up firn conditions certainly influence the 1965-2019 evolution, so this is another important source of uncertainty in the model simulations (Figures 7B and 7D; Supplemental Figure S3). That said, even when forced to temperate initial conditions in model experiments, the firn refreezes in the model in the 1970s and remains polythermal until the last decade (Figure 8). This may be due to a systematic underestimate in the melt rates or meltwater infiltration in the model, as discussed above, but it provides some support for the model-based inference of decadal-scale firn warming and the recent development of the PFA.

L249: "heat advection from melt water percolation" seems odd, since melt water typically is at 0 degrees C and if it encounters cold snow it will just refreeze thereby releasing heat. This is the only way in which heat is "advected", but that is probably not meant.

This is what we meant, but we did not explain this properly. We allow for rain that can be above 0°C to add to the meltwater, in principle, and consider this a source of heat advection if water a bit above 0°C percolates into sub-zero firn. First the water cools to 0°C, warming the surrounding firn (the heat advection), then it will refreeze and release latent heat. The latter is by far dominant and we have revised the text to focus just on refreezing here, Line 277.

L251-252: The symbols k_h and k_w are mixed.

Apologies, corrected, Lines 280-281.

L284-293: It remains unclear how melt-affected and not melt-affected firn are distinguished. I think this is an important aspect, because if there is indeed non melt-affected firn with a firn aquifer below, that probably implies that fast deep melt water percolation through piping is an important process here. I would like to see additional discussion on this in the manuscript.

The following sentence has been added Lines 112-114 "Melt-affected firn is distinguished by ice layers, ice lenses, or can be indicated by the lack of grain boundaries, the presence of air bubbles, and opacity." See also Lines 324-326 for added discussion on this in the results. Additionally, a photo has been added to the supplementary information, Figure S5.

L306-313: What is the density difference between 1964 and 2018 when considering the mean density between the LSS and 15m depth?

We have added a sentence on Lines 344-245: " The difference between the average densities from the upper 7 m in the 1964 and 2018 core is 33 kg m$^{-3}$, which is an increase of ~7%. " and we have elaborated on this section including information from the modelling results.

L318-319: This is comparing snapshots of subsurface temperature to extract trends. Since subsurface temperatures may vary strongly from year to year, care should be taken to determine long-term trends based on only two snapshots in time.

You are completely correct, we have removed this section of the manuscript, eliminating the temperature data from both the Divide site and the IRRP. The data cannot accurately be compared, as you note, as it is related to the recent weather and not long-term changes.

L344-345: These kind of statements about melt trends are hard to defend without any calibration or validation of melt estimates against observations.

This is fair, we have qualified this now to note that this is just based on the ERA climatology, and we lack in situ observational constraints. See Lines 361-362.

L375-376: "due to increased presence of ice layers": Is there any information on ice content in the 1964 core?

There is not any information on the ice content in the 1964 core, but Grew and Mellor (1964) discuss meltwater percolation and refreezing, and present stratigraphic plots that display some ice lenses, but with relatively low densities and no mention of ice layers. There is a figure in the report that displays ice lenses but only in the first several meters of the snow accumulation and not the firn (Grew and Mellor 1966, Figure 2).

L381-383: The modelled subsurface temperature trends may be reasonable, but the absolute values are quite a bit off when comparing temperatures in Fig. 5 and Fig. 7. This discrepancy is important to discuss in much more detail in the manuscript, especially since many of the conclusions for example on when firn became temperate and when the PFA may have formed are based on the modelled temperature evolution.

Agreed, now revised as discussed above and examined in some depth in the supplementary material. See also new Figures 7 and 8.

L382-383: "The ERA5 climate analysis": This is rather an analysis of snow model output. Please reformulate.

Thank you yes, this is from the energy balance and firn model as forced by ERA. Revised in the revised discussion of results, Lines 455-462.

L388: "increases" and "effect"

This section has been revised and typos have been corrected, Lines 464-474.

L391-394: I would suggest to reformulate this. It is unclear what this "first stage of densification" is. In my view there are two processes that affect densification 1) gravitational settling (which will go faster at higher temperatures) and 2) refreezing. Refreezing will increase subsurface temperatures, which in turn may increase the densification rate by gravitational settling/packing. That is a completely different sequence of processes than described in Line 391-394.

We have revised this section, Lines 464-474. There is also the process of the snow grains being rounded due to warming temperatures, which impacts the settling. We rewrote the line on 'first stage' to " Melting rounds snow grains and increases the rate of the first stage of densification ", Line 465.

L397: "effects"

Thank you, this section has been revised and typos have been corrected.

L411-412: "The firn model predicted the development of wet, temperate conditions in the deep firn following the 2013 melt season, although it took two years to fully develop (Figure 7)." But the observations reveal that the firn was already temperate in 2006. This should be acknowledged.

This was not really the cases – it was just a deep snowpit in 2006, extending to 7 m, and we have no knowledge about the firn temperatures below this.  We have included more information on the modelling but have removed the temperature data from 2006 and the discussion of this data.

L440: "Kuipers Munneke et al. (2014)"

Thank you for catching this. We have gone through the manuscript and edited the reference throughout.

L445-446: Kuipers Munneke et al. (2014) indicate what accumulation and melt conditions favour the development of firn aquifers. So in addition to the accumulation comparison it would be good to also compare melt rates with rates observed in southeast Greenland.

Good suggestion. We have added this discussion, Lines 548-551. We include melt estimates from southeast Greenland according to firn aquifer studies in the area by Miege et al (2016) and Miller et al. (2020). Indeed, both the accumulation and melt regimes are very similar to those in southeast Greenland.

L453: Temperature of the firn will not have a major impact on the perennial firn aquifer. Typically once a perennial firn aquifer has formed the firn above it is temperate and the

winter cold wave does not penetrate deep enough to case any refreezing. A factor that is important though is how easily the water can runoff via moulins and crevasses.

The first part is true in a warming climate, which we are currently experiencing, so we agree. However, sustained cool conditions can refreeze the deep firn from above, although on a diffusive (decadal) time scale vs. the potentially rapid work of meltwater infiltration and latent heat release.  e.g., if the ~10-m temperatures cool off due to reduced meltwater and cooler air temperatures, these eventually cool and refreeze the underlying firn. There are examples in Figure 8 and in Figures S3 and S4. We revised the discussion in the main text and now discuss the deep firn temperatures and the possibilities of cooling in the Supplementary Material. Drainage in crevasses or moulins is discussed on Line 42-43 – very true that this is the best possible way to drain the firn aquifer in a flat-lying area like this. We don't have radar data or other evidence of such features, and the presence of the water table that we drilled into suggests a lack of such features.

L454-455: Internal accumulation commonly refers to the amount of refreezing below the last summer surface, which is probably not what is meant here. See for example Cogley et al. (2011; https://wgms.ch/downloads/Cogley_etal_2011.pdf).

Agreed, thank you for catching this. Actually, reading the definition in the link you provided verifies that this is what we meant. However, through this it was brought to our attention that we included the wrong Schneider reference; we mean Schneider and Jansson 2004, not Schneider 1999. To clarify, we separated the sentences, Lines 560-561.

L469-470: Ice layers in snow and firn happen in any accumulation zone that experiences some melt, which is the case for the vast majority of glaciers on Earth. Hence, the presence of ice layers in firn is not something special. Please reformulate.

Thank you, agreed and rephrased, Lines 575-578.

L474-475: This is an important notion. I understand that there may not be any melt observations to make use of, but I would instead strongly suggest to perform new modelling experiments where one or more parameters affecting the modelled melt rates are calibrated such that a best match between modelled and observed subsurface temperature is achieved. Right now, it seems that modelled melt rates are underestimated, which would result in too little water percolation and refreezing in snow and firn, thereby explaining the currently underestimated subsurface temperatures. With a calibrated model, confidence in modelled melt rates and firn conditions would considerably increase!

We have added sensitivity experiments that increase melt rates to produce temperate firn, although we are hesitant to trust this model scenario more than the 'reference model'. We have added these results now though, Figures 7 and 8, discussed on Lines 416-445.  Of course it is true – they imply much more melt, drainage/ablation, and denser firn.

L477: "with most of the meltwater refreezing": It is unclear if this is still the case. The

subsurface temperatures reveal that firn was already temperate in 2006 implying that already then some melt water did not refreeze.

We revised this discussion and added some additional numbers, Lines 587-591:
"Within the model, 96% of total meltwater refreezes over the 55-year simulation, but this is reduced to 86% for the period represented by the firn core, 2005-2017. The remaining 27% drains to the deep firn through this period, where it is either retained within the PFA or it may drain from the system. A total of ~1.3 m w.e. 'runs off' through the period 2005-2017. In the model, this drains through the bottom layer and leaves the system; in reality, this water may drain through lateral transport in the PFA or at the ice-firn interface."

Over the full period, some meltwater infiltrated to depth in warm summers with high melt, but it does not escape the system (drain to 35 m depth) until after 2017, when the deep firn becomes temperate.  As noted above, we don't know that this was the case in 2006 – only that the seasonal snow and upper ~2 m of firn were temperate, with no knowledge of deep firn conditions.  It is possible though, per Figures 7 and 8 and the discussion in the results.

L484: It would be nice to have an additional figure showing the modelled density evolution.

We have added this figure, please see Figure 5E.

L487: "0.73+/-0.23 m". If this is calculated from Eq 6 then, if I am correct, this is not the actual surface lowering, but rather the surface lowering relative to a snow/firn pack that would not experience any melting. I do not really see why this is relevant here. For me, the interesting thing to know would be how much additional refrozen mass sits in the firn column in 2018 compared to 1964, because that is a mass term that is missed by geodetic mass balance observations.

Yes, this number refers to surface lowering due to the internal refreezing (not taking into account potential mass loss due to drainage into and out of the firn aquifer). We believe that it is useful to include this number to understand the impact that refreezing has on the snowpack.

The changes in near-surface density between 1964 and 2018 that we report provide an indication of the changes in mass over time. We agree that determining the change in mass of the snowpack as a whole would be ideal, but we lack the data to know how to properly extrapolate our local data on a regional basis, so we do not do this as it would require too many assumptions and suggest a level of confidence in the accuracy of the data that is unrealistic on a regional basis.

L487-488: "to have experienced a minimum of 0.73 ±0.23 m of surface lowering due to internal refreezing": Refreezing does not lower the surface, melting does and gravitational settling of snow/firn. Please clarify and rephrase.

Apologies, rephrased.

L487-497: I am missing a bit the point here. Surface elevation changes are the effect of long-term trends in melt and accumulation. How much of the melt water refreezes does not (directly) affect surface elevation or thinning. I would rather expect a discussion here on the impact of increased densification on geodetic mass balance estimates. Geodetic mass balance observations will just consider surface height changes and not any mass changes that result from an increasing density of firn.

As stated in the reply to Line 487, we unfortunately lack the data to provide a meaningful discussion of this point in the paper.

L496: "liquid water retention processes cause the surface to lower". This is not correct. Refreezing just adds mass to the existing firn column, which leads to densification, but not to thinning! Higher firn temperatures after refreezing do speed up the compaction (gravitational settling) process though.

Thank you, we have rephrased this, Lines 607-610.

Figure 1: It could be good to include coordinate axes.

Done.

Figure 6c: In addition to Fig. 7 also Fig 6c confirms that the modelled subsurface temperatures are much colder than observed (Fig. 5).

We have removed this figure, but we do discuss the discrepancy at length in the manuscript and supplementary material.

---

## Author Response (AR3)

Dear Dr. Etienne Berthier,

Thank you for the opportunity to respond to the reviewers and tidy up the manuscript. We have attended to the comments point-by-point and we added the correct DOIs for the videos. We believe these changes will make the manuscript much stronger. We look forward to your response.

Thank you, Naomi Ochwat, on behalf of the coauthors.

Below you will find the description of the edits the reviewers requested and how we responded to them.

**Referee # 1**

General Remarks

The study by Ochwat et al. has again been modified substantially. Once again, I would like to thank the authors for making a thorough effort in revising their study. I think my comments were addressed. The study has gained a much stronger focus on the modelling and the description of the modelling approach as well as the results has further matured.

I have few comments on details and one suggestion which is more general and concerns surface lowering. Assuming that thickening from accumulation and thinning due to ice flow are in balance, the surface at the plateau would remain at stable elevations. However, melt has increased over the recent years likely leading to an imbalance and surface lowering. The terms of the imbalance seem to have three components that all cause surface lowering but their impact on mass balance differs: (i) melt which runs off (actual mass loss; amount poorly known because runoff is difficult to observe), (ii) melt that refreezes (apparent mass loss; subject to uncertainties but better known than the other terms) and (iii) melt that makes it into the PFA (likely contributing to mass loss but this is not known with certainty; this term is not well constrained and overlaps with (i)). Finally, there is a potential fourth component, namely accelerated compaction of warming firn (apparent mass loss, difficult to quantify as there is substantial uncertainty to what degree the firn warmed). While I think the authors have now outlined these different components well, I still suggest distinguishing their influence on geodetic mass balance more clearly in the discussion, in particular in Section 5.3.

Thank you for this good summary. This is a nice way to summarize the mass balance influences – we agree, (ii) is best-known here, from the firn core, while (i), (iii), and (iv) are all potentially significant but poorly understood. Indeed, one of the contributions of this study is having identified that (i), (iii), and potentially (iv) are active processes at this site, and need further work to constrain. We have rewritten

the first paragraph of section 5.3 to introduce these different components, the factors that influence them, and indicate how they are typically interrelated. We then outline their likely importance in the following paragraphs, using much of the text as previously written. We appreciate the suggestion and additional clarity here.

Detailed Remarks

Line 420: I suggest removing "perpetual".

We removed this, thank you.

Line 558: "... has played ..." Why past tense? It still plays an important role.

Yes, apologies. We fixed the tense, line 582.

Lines 586 – 587: This sentence is unclear. At the drill site, summer melting is the precondition to both refreezing and mass loss (through runoff). I think what is meant it that both refreezing and runoff will result in surface lowering (but only one of them is associated with mass loss).

Yes, that is what we mean. We reworded this and believe this is clear now, lines 625-627. Thank you.

Line 588: I do not understand " ... remaining 27%". Both 100 - 96 and 100 - 86 (percentages from previous sentence) do not yield 27 %.

Apologies, this was our confusion around the percentage of refreezing and runoff for total melt vs. the net surface melt. For clarity, 96% and 86% of total melt refreezes. But expressed as a percentage of the net surface melt (i.e., not including recycled meltwater), this is 91% and 73% - hence the 27% runoff, for the period 2005-2017 (see Table 2 for the numbers behind these values). As a percentage of the total melt, the runoff from 2005-2017 is 14%, as the reviewer notes. However, we decided to revise this to discuss everything in terms of the net melt, which represents the effective meltwater that is available at the surface and infiltrates the snow and firn (where it can refreeze or drain – that which drains can then either be stored as liquid water in the pore space of the deep, temperate firn, or it can drain from the system – runoff and mass loss). Thank you for catching this. We have corrected this and now more clearly discuss it in lines 627-632.

Lines 612 – 619: In its current form, the study thoroughly discusses uncertainties in the measurements and in modelling. The statements here in the conclusion, however, do not reflect the considerable uncertainties anymore. No need for another detailed discussion, somewhat more cautious wording or a brief remark on uncertainties might suffice.

This is a fair comment; we don't wish to lose the uncertainties. We added a sentence to the conclusion to help address this: "Though not observationally constrained and therefore uncertain, the modelling results suggest the likelihood of significant increases in melting and refreezing since the 1960s at this site, driving decadal-scale increases in firn temperature, ice content, and density" on lines 642-644.

**Referee # 3**

Comments exported from the annotated .pdf file.

What about rainfall?

We neglect rainfall in this study, as we don't know the amount of summer rainfall at the site and are not sure how reliable ERA5 estimates of this may be. Summer temperatures are cold (mean JJA temperature of –2.4°C; Figure 5a and Table 2) and over the course of about 10 field seasons working near the site in July (LC, at the Copland Divide camp), we have witnesses numerous summer snowfall events but never a rain event. We therefore think that summer rainfall is rare, but it is likely to occur on occasion, which would be in addition to the melt totals and would also add heat to the snow/firn. This would be a good future consideration to follow up on this work, as summer rain events may become more frequent in future years and decades. We now briefly discuss this in the main text (lines 272-276).

These numbers do not add up to 0.38 m w.e.

Apologies, we mixed the observed vs. modelled ice content, whereas the other numbers quoted are from the model. We revised to report just the model numbers here, for internal consistency. Now revised in the abstract, line 18. This is also refreshed in the main text, lines 390-396, with a more explicit discussion of the amount of liquid meltwater retained in the firn (in the model).

Please add some references for these statements.

References added (Sommerfeld and LaChapelle, 1970; Cuffey & Paterson), lines 41-42.

It would be relevant to mention the recent melt water retention intercomparison study in Vandecrux et al. (2020; https://doi.org/10.5194/tc-14-3785-2020).

Reference added as suggested, line 51.

It could be good to mention that this requires cold (deep) firn.

Revised as suggested, line 52.

The modelling part of this study could be introduced here too.

Added as suggested, lines 62-63.

and

Revised as suggested, line 122.

and

Revised as suggested, line 124.

I am a bit confused here. Figure 2B does present the density profile and I do not see a gap between 13 and 15 m or elsewhere in the record.

Apologies, we reworded this, lines 124-126. Enough samples were intact to estimate the bulk density and plot the density stratigraphy, but because of the greater uncertainty with Core 2, we focus most of the analysis on Core 1 (now stated).

Note: How is this done? It is not described in the Supplementary information. Are elevation gradients used?

Sorry to be unclear here, this was done using the elevation difference of the ERA5 grid cell (2522 m) vs that at our site (2640 m), $\Delta z$ = 118 m, and (i) assuming a temperature lapse rate of $-5°C$ km$^{-1}$, as is typical of a glacier surface, (ii) a constant relative humidity, which then informs calculation of specific humidity for the lapsed temperature of the study site, and (iii) air pressure adjustment following $dP = -\rho g \Delta z$, with air density $\rho$ based on the ERA5 temperature. We have added this discussion to the supplement.

There is no mentioning in this section of precipitation in the form of snow accumulation and rainfall, which both are relevant parameters for modelling snow conditions. How are they accounted for?

Sorry to omit this as well – we did not give snow accumulation much attention, but we use the ERA5 total precipitation, updated monthly in the model and assuming that accumulation = precipitation at this site. As noted above, we believe that rainfall is rare at this site, and can be neglected, although future work should test this assumption as occasional summer rainfalls may occur. ERA precipitation totals are low compared to snowpack and firn-core observations of accumulation at the site, so we scale ERA precipitation inputs by a factor of 1.6, giving an average ($\pm$ 1$\sigma$) of 1.83 $\pm$ 0.32 m w.e. yr$^{-1}$. We added a short discussion in the main text, lines 271-276.

Doesn't ERA5 come with hourly resolution?

Yes, true, though we bridged with ERA20th century from 1965-1978 (available every 3 hours) and we also prefer not to work with the large dataset that attends the hourly data, so we chose to work with daily meteorological forcing. Our simple model of diurnal cycles (based on daily Tmax and Tmin, along with daily mean and maximum shortwave radiation) is expedient and computationally efficient in terms of the memory demands of the numerical experiments. While not discussed in the manuscript, we are working towards distributed modelling at a large scale (e.g., St. Elias range; Greenland Ice Sheet), so we are developing melt and firn models that are pragmatic and feasible at that scale. We also have an eye to future projections, where climate model outputs are seldom available with hourly precision, so we are developing and calibrating our methods with daily meteorological forcing.

Is heat advection due to accumulation also considered?

No, we neglect this and note that in line 283.

It would have been 'safer' to repeat multiple years rather than one year for intialization.

This is true, although we cover it by including a number of sensitivity experiments around this assumption of a perpetual 1965 climatology. The year 1965 had representative mean annual and mean summer temperatures compared to the long-term means, within 1-$\sigma$, and our results are not strongly sensitive to using this year for the model spin-up, but we agree that it would be preferable to use historical forcing for the model spin-up, e.g. the period 1935 to 1965. Now acknowledged, lines 293-295. We could add this experiment if the Editor thinks this to be important, but the range of sensitivity experiments around the initial conditions covers a much larger span of conditions than the interannual variability at the site, so we believe that we have adequately addressed this sensitivity and source of uncertainty.

In fact we considered going back to ERA20c to do exactly this, a spin-up based on the historical climatology, but in looking into this we see that ERA5 has now been extended back to 1950, with a preliminary release of reanalyses for the period 1950 to 1978 in ~February 2021, since the time of our last revision. This offers an opportunity to avoid the splice between ERA20c and ERA5 in our study, and to extend the simulation back to 1950. We have done this, but we suggest not to introduce this to the revisions at this stage, as it would constitute more of a major revision (changes to methods, figures, tables, discussion) and I suspect the appetite for another round of major review is limited after such extensive and helpful work by the reviewers and Editor to date. The back-extension to 1950 using purely ERA5 forcing, with a different spin-up strategy, slightly changes values for things like the decadal trends, etc., but does not change the essential results or conclusions.

For the interest of the Editor and the reviewers, here are the results for the extension back to 1950 with ERA5 forcing now and a spin-up strategy that repeats 1950s climatology for 30 years (3*10 years), before launching into the simulation from 1950 to 2019. Figure R3-1 plots contours of daily firn temperature evolution through this 70-year period and Figure R3-2 plots time series of the main variables of interest in our analysis: summer temperatures, modelled melt, firn temperatures, firn density, and firn ice content. Results are qualitatively unchanged from those shown in Figures 5 and 6 of the manuscript. These new results are arguably cleaner, given that the meteorological forcing is all from ERA5. The results also indicate an interesting warming of the firn to ~7 m depth in 1959, which could be tempting to relate to the early 1960s inference of temperate conditions at this depth. This would be over-interpreting the model though, given the uncertainties within it. Overall, the modelling remains consistent to the results in the manuscript, with reconstructions of decadal-scale firn warming and the likelihood that deep, temperate firn and the PFA are recent features, having developed since 2013. The modelled decadal-scale increases in firn density and ice content are also robust.

[Figure]

**Figure R3-1**. Modelled firn temperature to 35 m depth for the years 1950 to 2019, using ERA5 forcing and the reference model parameters.

[Figure]

**Figure R3-2**. Modelled evolution of meteorological and firn conditions from 1950 to 2019 under extended ERA5 forcing. (a) summer (JJA) air and snow surface temperature. (b) Net annual surface melt and the annual melting minus refreezing, which represents drainage to the deep firn. Where negative, this is deeper meltwater that refreezes in the following calendar year. (c) Mean annual snow surface and firn temperatures at 10, 20, and 35 m depth. (d) Maximum depth of the annual thawing and wetting fronts. (e) Average firn density, and (f) Firn ice content.

The lack of consistency between the density profiles of the two cores is quite striking (Figure 2AB). It could be worth highlighting this more, or could this be mainly the effect of problems with Core 2? Similar strong variability in firn stratigraphy over very short distances has been found in Marchenko et al. (2016, http://dx.doi.org/10.1017/jog.2016.118).

This is also true, and while bulk densities and ice contents are similar (Table 1), the stratigraphies are truly not correlated. We added a sentence on lines 347-350 to

discuss this in a bit more detail. It is not likely related to the compromised samples, as the firn ice-layer stratigraphy was conducted in the field, on the full core samples and prior to sawing them into 10-cm sections.

This is quite significant given the relatively modest trend in temperature. Is this maybe related to e.g. an increased frequency of warm spells or less summer snowfall?

It is indeed a large trend in surface melting, partly driven by the strong ($\sim$3$\sigma$) melt years in 2013 and 2017. Overall, the last decade really pulled up the linear trend, and a linear fit might be misleading, though it is statistically significant. These two summers were also anomalously warm (the warmest summers in the 55-year records, at –0.7°C and –0.9°C, or $\sim$ +2$\sigma$ above normal). While not as exceptional as the melt totals these summers, this relates to the nonlinearity of temperature and net energy effects on melting, particularly at temperatures close to 0°C. We don't examine the effects of summer snowfall within the model – it is treated as a random variable in the albedo model (as described in detail in Marshall and Miller, 2020), but its impact is modest as most summer precipitation events are snow events throughout the period, and the site has remained within the accumulation zone of the glacier, with relatively high values of albedo.

It could be added that the trend roughly matches the observed density trend.

Now noted, line 419-420.

Temperature anomalies may also reflect differences in height above the surface the data represent (what atmospheric level do the ERA data represent?). This matters since temperature gradients near the surface are usually strong.

It is true. Within ERA5 and ERA20c, we took the surface-level data, which represents 2-m temperature and dew-point temperature. Surface-level wind speeds in ERA are at 10 m. The AWS temperature data which we use for the bias-adjustment (0.6°C) is also at $\sim$2 m. We added a note the ERA data heights above the surface in the methods section, line 261.

Selected
Revised as suggested, line 432.

obtained
Revised as suggested, line 437. Thanks for catching this.

Was 1965 a relatively cold year compared to the surrounding years?

We discuss this in more detail in the Supplementary Information, but no, the mean summer (JJA) and annual air temperatures in 1965 were almost identical to the 55-year average: –2.5°C and –10.8°C, compared with the 1965-2019 averages of –2.4 ± 0.8°C and –10.7 ± 0.9°C.  Incoming solar radiation in JJA 1965 averaged 304 W m$^{-2}$, compared with a normal of 298 ± 9 W m$^{-2}$. Net energy and melt were a bit lower than the long-term average, due to lower incoming longwave radiation, 240 vs. 255 W m$^{-2}$ (it seems to have been a clear-sky summer), but overall, 1965 was very representative of the longer-term climatology, particularly for the 1960s through 1990s. We add a short note to this effect here, lines 452-459.

I think it is good that this is statement is now included.

Also this part is a valuable addition.

Thanks for these comments. We are happier now as well, with the clear and direct discussion of uncertainties and limitations. There is a strong case for further study of this interesting site, given how much we don't know about it.

Arguably the 'reference model' could have been set to the one that matches the observations best, but the current approach works well for me too.

We did consider this too, but it is truly hard to force the model to satisfy this observation of intermediate-depth temperate firn in the 1960s – we have to push the climate forcing pretty strongly away from the ERA5 reanalyses, and no combination of model parameters on their own (admitting to some structural uncertainties in our model) were able to produce this. We find the modelling 'best guess' to be a more helpful reference model, from which we can assess how the model forcing, parameters, or physics need to be pushed to match the available observations.  This informs flaws in the model, or alternatively raises questions about the validity of the borehole air temperature observation – we feel that both are in question, and hope that this is reflected in the discussion.

This is an important notion, indeed very slight changes in climate have a major impact on the state of firn!

Agreed, we were (are) surprised at how delicately balanced this site, essentially at the transition point from polythermal to temperate conditions.

It could be noted that with densities between 800-900 between 30-40 m there is not much room for a PFA to extend much deeper, since the pore-close of density is in the same range.

Added as suggested, lines 548-550.

This may be good to rephrase, the main problem for geodetic mass balance observations is the unknown density (change), which makes it hard to interpret mass change associated with surface lowering.

Thanks for this suggestion, we rewrote this, lines 596-597.

Again, what about rainfall? In case it is not accounted for, it should be acknowledged as a source of uncertainty (assuming there are occassional rainfall events at the site).

Agreed – we don't account for rainfall but now acknowledge this in the manuscript (lines 272-276) and also discuss that this could be an important source of uncertainty in a warming climate, which should be considered in follow-up studies at this site. Discussion added on lines 615-620.

See my earlier comment, refreezing does not lower the surface, but rather increases the density (which is the troublesome part for geodetic mass balance estimation, which assumes constant density in time). I understand what is meant here, but the way it is formulated is not correct.

Sorry that we continue to struggle with wording this clearly. Revised, lines 656-659.

Does this include rainfall or not?

No, rainfall is neglected and is assumed to be negligible – now directly discussed in the text, per the comment above. We also add a note to this effect in the Table, line 874.

The term 'net melt' term is slightly confusing. How to distinguish between freeze-thaw cycles and refreezing at greater depths? With 'melt', 'refreeze' and 'drainage' there is a nice set of variables that is mass-conserving, so there is no need to have another term.

Yes and no. We are also struggling to define this. We agree on melt, refreeze, and drainage – these are all well-defined and refer to the full snow/firn column. And yes, they conserve the overall system mass and energy. Net melt is something important though, which we evidently need a different name for. We thought about 'surface melt', but all of the melting is surface melt. Net melt is the actual surface ablation: the mass loss and drawdown of the surface due to melting (though not ablation in the proper sense, which also includes sublimation, wind scour, etc). Surface ablation or mass loss of the surface layer due to melting is less than the total melting, due to meltwater that is retained in the near-surface (irreducible water content) and undergoes successive freeze-thaw cycles. The same water molecules are melted many, many times by this process, as there are diurnal freeze-thaw cycles throughout the summer at this site. This consumes a significant fraction of the net energy that is available for melting – a portion of this net energy is directed to 'recycled' water/ice, essentially. As a result, the 'net melt' is less than the total melt: about 44% of total melt at this site. (It is closer to 90% at a temperate glacier site where we have studied this process, Samimi and Marshall, 2017). This is important because this is the actual amount of mass that is transferred to deeper snow and firn, as meltwater infiltration. We are sticking with the term 'net melt', for lack of a better idea here, and by analogy with e.g., total vs. net income. But we appreciate that it is not very clear, so we have attempted to better explain this, lines 869-874.

We removed this, thank you.

---

## Author Response (AR4)

Dear Dr. Etienne Berthier,

Thank you for correcting the manuscript. We appreciate your time working with the manuscript and us. We have addressed the comments you provided below.
Thank you,

Naomi Ochwat, on behalf of all of the coauthors.

**L217. For clarity, maybe refer to it as the "IRRP A site" as you do in Figure 1.**

We have made this edit, thank you.

**L274. "We do not have"**

We have made this edit, thank you.

**L276. What about "and while working at the Copland" ?**

We have fixed this sentence, thank you.

**L288. Put the minus sign as exponent for conductivity**

We have fixed this, thank you. We also have realized this was a formatting problem throughout the text and have revised accordingly.

**L298. Maybe replace "normal" by "close to climatological values"**

We have replaced the word normal with "close to standard climatological values."

**L426. I find the formulation weird. A weather station is punctual/local not regional by definition. Or do you refer to a network of stations? Or did I misunderstood?**

Thank you for catching that, "local" is a much better word to use here to describe the weather station. We have changed it accordingly.

**L454. Can authors double check this sentence? It was not clear to me.**

We have fixed the sentence, now line 457.

**L505-506. I wonder whether this should not be moved to (or first mentioned in) the "results" section when liquid water is mentioned, i.e. L339 (section about core stratigraphy)**

We have added an additional sentence on line 340 to include the supplemental videos.